# The partial Bondi gauge:
# Further enlarging the asymptotic structure of gravity

Marc Geiller[1] & Céline Zwikel[2]

[1]*ENS de Lyon, CNRS, Laboratoire de Physique, F-69342 Lyon, France*
[2]*Perimeter Institute for Theoretical Physics,*
*31 Caroline Street North, Waterloo, Ontario, Canada N2L 2Y5*

**Abstract**

We present a detailed analysis of gravity in a partial Bondi gauge, where only the three conditions $g_{rr} = 0 = g_{rA}$ are fixed. We relax in particular the so-called determinant condition on the transverse metric, which is only assumed to admit a polyhomogeneous radial expansion. This is sufficient in order to build the solution space, which here includes a cosmological constant, time-dependent sources in the boundary metric, logarithmic branches, and an extra trace mode at subleading order in the transverse metric. The evolution equations are studied using the Newman–Penrose formalism in terms of covariant functionals identified from the Weyl scalars, and we build the explicit dictionary between this formalism and the tensorial Einstein equations. This provides in particular a new derivation of the (A)dS mass loss formula. We then study the holographic renormalisation of the symplectic potential, and the transformation laws under residual asymptotic symmetries. The advantage of the partial Bondi gauge is that it allows to contrast and treat in a unified manner the Bondi–Sachs and Newman–Unti gauges, which can each be reached upon imposing a further specific gauge condition. The differential determinant condition leads to the Λ-BMSW gauge, while a differential condition on $g_{ur}$ leads to a generalized Newman–Unti gauge. This latter gives access to a new asymptotic symmetry which acts on the asymptotic shear and further extends the Λ-BMSW group by an extra abelian radial translation. This generalizes results which we have recently obtained in three dimensions.

# 1 Introduction

This article aims at reassessing and relaxing some of the hypothesis used in standard analysis of the asymptotic structure of gravity in Bondi gauge. We start by presenting some general motivations, but the reader familiar with the topic can jump to section 1.2 for a summary of the new results.

## 1.1 Motivations

The formalization and the study of the asymptotic structure of general relativity, pioneered by Bondi, van der Burg, Metzner, and Sachs [1–7], followed shortly after by Newman, Penrose, and Unti [8–12], played a central role in the understanding of gravitational radiation and of its non-linear nature (see also [13] for a historical account of the contributions of Robinson and Trautman). It was recognized early on by these authors that the asymptotic symmetry group of asymptotically flat four-dimensional spacetimes is infinite-dimensional [3, 5, 14]. Recently, these asymptotic symmetries have received tremendous attention following the unraveling of their connection with memory effects [15–20] and the soft graviton theorems controlling the infrared behavior of scattering amplitudes [21, 22]. These connections, which were initially spelled out in gravity in [23–25] (but also extended to other massless theories [26]), are the manifestation of unexpectedly rich mathematical structures attached to boundaries of gravitational systems. The investigation of these boundary structures and of the associated physics has unfolded along several directions, which have all required some relaxation and extension of the boundary conditions being considered.

In four-dimensional asymptotically flat spacetimes, the historical BMS symmetry group was extended to include local Lorentz transformations [27–31], and later on arbitrary smooth diffeomorphisms of the boundary two-sphere [32–34]. This was then shown to be connected to a new, subleading soft graviton theorem [35, 36], and in turn to a new type of memory effect [37]. This also brought the perspective of establishing a duality between gravity in four-dimensional asymptotically flat spacetimes and a two-dimensional conformal field theory living on the boundary celestial sphere [38–42], which then developed into the program of celestial holography [43–48]. It was shown in this context that the scattering amplitudes satisfy an infinite tower of soft theorems, which are in turn controlled by a higher spin symmetry [49–54]. Extending the known relationship between the spin-0 (mass) and spin-1 (angular momentum) charges, the leading and subleading soft theorems respectively, and the supertranslations and superrotations respectively, it was recently shown in [55] that there is indeed a spin-2 charge [56], related to a new type of asymptotic symmetries, whose asymptotic evolution equation is equivalent to the sub-subleading soft graviton theorem [57–62]. This hierarchy presumably extends to an infinite tower, and the properties of the charges and associated asymptotic symmetries are under active investigation.

The analysis of the asymptotic structure of general relativity has also been extended to spacetimes which are asymptotically (A)dS$_4$ [63–69], where part of the motivations and intriguing questions pertain to the very nature, when $\Lambda \neq 0$, of gravitational radiation, mass, angular momentum, and news [70–85]. Even if there is no notion of $S$-matrix in this context, there are meaningful asymptotic symmetries and (yet poorly understood) memory effects. This has in particular led to the introduction of the $\Lambda$-BMS group of asymptotic symmetries [67–69].

Concomitant with these developments studying enlargements of the asymptotic structure and the associated physics, there as been work focusing on the technical tools used to describe asymptotic charges [86–89], and to discuss subtle issues related to non-integrability and/or non-conservation sourced by flux [90–97], symplectic renormalization [98, 99], and corner ambiguities [100–104].

Three-dimensional gravity has also played an important role in the understanding of the asymptotic structure of gravity, of asymptotic symmetries, and of their relationship with holography. Since the seminal work of Brown and Henneaux on AdS$_3$ [105], a variety of boundary conditions and gauges have been introduced [91, 106–122]. Recently, we have presented an analysis of three-dimensional gravity in a so-called Bondi–Weyl gauge, which enables in particular to access Weyl rescaling of the boundary metric [123]. This analysis has furthermore been performed with so-called leaky boundary conditions, which are a relaxation of the variational principle due to the presence of sources for the boundary metric (i.e. the three components of the two-dimensional boundary metric on $\mathcal{I}^+$ are completely free). This is analogous to what happens in four spacetime dimensions, where the leakiness is however related to the presence of non-trivial news, and therefore to the fact that the system is open. In this three-dimensional Bondi–Weyl gauge, one finds integrable charges consistently with the fact that there are no local degrees of freedom, but these charges are however not conserved because of the arbitrary time-dependency of the sources for the boundary metric. More importantly, we have shown that the asymptotic symmetry algebra, in addition to the[1] $\Lambda$-BMS$_3$ and Weyl sectors, contains an extra abelian sector corresponding to radial translations. Together with the Weyl charges these translations form an Heisenberg algebra. The study requires a careful analysis of issues related to integrability and corner ambiguities, but eventually leads to an extension of the $\Lambda$-BMS$_3$ group.

In four-dimensional asymptotically flat spacetimes, the Weyl charges and the role of boundary Weyl transformations were discussed extensively in [28, 124, 125]. Part of the motivations for the present work is to investigate whether a gauge similar to the one used in [123] exists in four-dimensional asymptotically locally (A)dS spacetimes, and whether the BMSW group of asymptotic symmetries with cosmological constant can be enhanced with an extra radial translation. In short, we are investigating alternative gauge choices for the analysis of asymptotic symmetries, and asking how much extra structure (such as the time-dependency and sources in the boundary metric) can be included in the solution space. Interestingly, this brings out the connection with many other motivations and open questions, which are most clearly understood by summarizing our construction and the ensuing results.

## 1.2  Summary of the results

We list by category the various points which are touched upon in this work. It should be noted that these points are to a large extent independent. For example, the introduction of logarithmic terms in the solution space is unrelated to the appearance of the radial translation symmetry in the Newman–Unti gauge, and neither are related to the sources for the boundary metric.

---

[1]More precisely, the algebra associated to $\Lambda$-BMS$_3$ is $\mathfrak{bms}_3$ when $\Lambda = 0$, and the double Virasoro algebra $\mathfrak{vir} \oplus \mathfrak{vir}$ when $\Lambda \neq 0$.

## Gauge choices

This work begins with Bondi coordinates and the Bondi form (2.1) of the line element, where only the three conditions $g_{rr} = 0 = g_{rA}$ are imposed. This is what we call the *partial Bondi gauge*. Usually one assumes from the onset a full gauge fixing by supplying a fourth condition, either chosen as the Bondi–Sachs (BS) determinant condition $\det(g_{AB}) = r^4 \det(q^\circ_{AB})$ [3], or as the Newman–Unti condition $g_{ur} = -1$ [9]. Here $g_{AB}$ is the transverse metric on the two-sphere, and $q^\circ_{AB}$ is the metric of a fixed round sphere. In [28] the algebraic BS condition was traded for the differential condition $\partial_r \big( \det(g_{AB})/r^4 \big) = 0$ in order to allow for an arbitrary two-sphere metric $q_{AB}$ and boundary Weyl transformations. Following this idea, it is natural to introduce the differential NU gauge condition $\partial_r g_{ur} = 0$, which as we will recall also enables to access Weyl transformations.

The BS and NU gauges[2] differ by the choice of a radial coordinate [126–129]. This latter is an areal distance in BS gauge, while it is the affine parameter of the null congruence in NU gauge. Another major difference between the two gauges is the fact that in NU gauge the first subleading term in the transverse metric, denoted $C_{AB}$, is allowed to have a non-vanishing trace $C$ [127–129]. The asymptotic shear is the trace-free part $C^{\mathrm{TF}}_{AB}$, which differs in NU gauge from $C_{AB}$ by the possible presence of a trace. This trace is set to vanish in BS gauge by the determinant condition. In addition, and for related reasons, the NU gauge actually possesses an extra asymptotic symmetry compared to the BS gauge (see table 1 below). This is a radial translation appearing at order $\mathcal{O}(r^0)$ in the radial part of the asymptotic Killing vector. This symmetry can a priori be used to set the trace $C$ to vanish, which is equivalent to a choice of origin for the affine parameter $r$. One should however first investigate whether there is a non-trivial charge associated to the radial translation, since this would mean that the symmetry is physical and not pure gauge. This is indeed what we have observed in the three-dimensional case[3] [123], where the radial translations also act on a subleading term in the circle metric, and possess a non-vanishing charge which forms a Heisenberg algebra with the Weyl charges. The study of the charges in the four-dimensional case is devoted to future work, but here we set the stage for this investigation. Even if this symmetry turns out to be pure gauge in general relativity and associated with a trivial flux-balance law, there are suggestions that it could become physical and associated with a memory effect in modified theories of gravity like Brans–Dicke [130–133]. It would also be interesting to study these charges in the presence of matter, as in Einstein–Maxwell theory [134, 135].

Interestingly, it is actually possible to perform the whole analysis of the asymptotic structure (i.e. the solution space, the potential, the symmetries, and the transformation laws) without having to choose between the BS and NU gauge until the very end. It is therefore possible, as the title suggests, to completely characterize the partial Bondi gauge. This has the advantage of treating the NU gauge very closely to the BS gauge. At the end of the day the two gauges can be reached by imposing different conditions on the traces of the subleading tensors in the transverse metric. So far, studies of the NU gauge have relied on the Newman–Penrose formalism to build the solution

---

[2]From now on we refer to the differential versions of these gauge conditions.

[3]What we have called the Bondi–Weyl gauge in [123] is actually a differential Newman–Unti gauge. In this reference the three-dimensional determinant condition is indeed relaxed, and the boundary source $\beta_0$ is turned on.

space [127, 128, 136, 137], while the BS gauge solves the tensorial Einstein equations using the Bondi hierarchy. Here we build the solution space in the partial Bondi gauge using the tensorial equations, and then switch to the Newman–Penrose formalism to study the evolution equations. We therefore treat both the BS and NU gauges in one go using the two approaches (Einstein and Newman–Penrose).

| gauge | condition | radial coordinate | symmetry group |
|---|---|---|---|
| algebraic BS | $\det(g_{AB}) = r^4 \det(q^{\circ}_{AB})$ | areal distance | $(3 \cdot \infty)$-dimensional |
| differential BS | $\partial_r \big( \det(g_{AB})/r^4 \big) = 0$ | areal distance | $(4 \cdot \infty)$-dimensional |
| algebraic NU | $g_{ur} = -1$ | affine parameter | $(4 \cdot \infty)$-dimensional |
| differential NU | $\partial_r g_{ur} = 0$ | affine parameter | $(5 \cdot \infty)$-dimensional |

Table 1: In the algebraic BS gauge the asymptotic symmetry group contains one supertranslation and two superrotations. In the differential BS gauge this is extended to include a Weyl rescaling. In the algebraic NU gauge however we have an extra radial translation only, while the use of the differential NU gauge also allows for Weyl transformations, and therefore leads to a 5-dimensional symmetry algebra given by (1.1)

## Logarithmic terms and weakening of peeling

It has been known for a long time that the solution space may contain logarithmic terms in $r$. These can appear in two ways. First, they can appear when solving the Einstein equations, as in $g_{uA}$ at $\mathcal{O}(r^{-3})$ [138]. Second, they can actually be introduced by hand from the onset when writing the radial expansion for $g_{AB}$, which may be chosen as polyhomogeneous [139]. Although these terms have slightly different origins (a consequence of the equations in the former case, and a choice in the latter), they turn out to be related via the evolution equations [139].

Here we include from the onset logarithmic terms in the expansion (2.3) of the angular metric. Our goal is to see how they affect the solution space, and to derive their evolution equations. We find, consistently with the Fefferman–Graham theorem, that all the logarithmic terms vanish when $\Lambda \neq 0$ [140]. When $\Lambda = 0$ we identify the evolution equations relating the logarithmic terms, and also explain how the presence of these terms modifies the peeling behavior of the Weyl tensor by producing overleading non-smooth terms in $\Psi_0$ and $\Psi_1$. Such violations of the peeling and arguments in favor of logarithmic terms have been discussed in the literature on numerous occasions [126, 138, 141–147], but there is so far no agreement as to the type of realistic physical situations in which this would occur. Under some technical assumptions it has indeed been shown that compact sources with no incoming radiation preserve a smooth peeling [148–150]. Nonetheless, we include at minor cost logarithmic terms in our solution space for the sake of understanding the modifications which they induce. In particular, we find that part of the symmetry arguments used in [124] in order to single out the covariant functionals and the asymptotic equations of motion actually break down in the presence of logarithmic terms. It will be interesting in future work to study symplectic renormalization and the construction of the charges in the polyhomogeneous case. More importantly, it is still unclear how the logarithmic terms in the solution space are related to the logarithmic terms (in $u$) appearing in the soft theorems [151–153].

## Enlarged solution space

In addition to the trace terms arising from the use of the partial Bondi gauge and to the logarithmic terms discussed above, our solution space is built with a non-trivial time-dependency for the boundary metric, and also includes the "boundary sources" $\beta_0$ and $U_0^A$ which appear as integration constants at leading order in the resolution of the $(rr)$ and $(rA)$ Einstein equations. The boundary metric on $\mathcal{I}^+$ is parametrized by the time-dependent data $(\beta_0, U_0^A, q_{AB})$. While $q_{AB}$ is allowed, since the work of Barnich–Troessaert [27, 28, 30], to be different from the fixed round sphere metric $q_{AB}^\circ$, the additional fields $\beta_0$ and $U_0^A$ are typically set to zero by hand (the exception being [67] in four dimensions and [91, 123, 154, 155] in three dimensions). Here we include them in the construction of the solution space and the symplectic potential because in NU gauge $\beta_0 \neq 0$ is required in order to have access to the BMSW group (and not just the subgroup where Weyl is related to the superrotations). For convenience we will often refer to $\beta_0$ and $U_0^A$ as the "boundary sources", and therefore leave $q_{AB}$ out of this denomination. We will also sometimes need to set the sources $\beta_0$ and $U_0^A$ to zero in order to have simpler expressions to display, but we will never freeze $q_{AB}$. This also justifies why we call $\beta_0$ and $U_0^A$ the boundary sources (although they are unrelated to any source of physical radiation).

Allowing for time-dependency of the boundary metric when $\Lambda \neq 0$ is necessary in order to attempt discussing radiation in (A)dS (although this remains an ambiguous concept). One of the Einstein equations indeed implies that when $\Lambda \neq 0$ the shear $C_{AB}^{\mathrm{TF}}$ is related to the time evolution of the boundary metric and to the boundary source $U_0^A$. If both quantities are excluded from the solution space, the setup trivializes immensely since $C_{AB}^{\mathrm{TF}}$ is forced to vanish on-shell. Including this time-dependency and $\Lambda \neq 0$ also enables us to reproduce results of [67] concerning the mass loss and the symplectic structure in (A)dS. In particular, we identify the canonical partner to $q_{AB}$ in terms of quantities computed from the Weyl scalars, namely a combination of the radiation $\Psi_4$ and the covariant spin-2 tensors. The solution space presented in section 2.2 is, in summary, an extension of that in [67] including logarithmic terms (which survive the flat limit) and the trace $C$ (which survives when going to the NU gauge). We hope that this will also later help study aspects of sourced holography in Robinson–Trautman spacetimes [156–160].

The full structure of the solution space is summarized in table 2. After having determined the radial expansion of the functions entering the partial Bondi gauge line element (2.1), we study the evolution equations which are implied by the $(uu)$, $(uA)$ and $(AB)$ Einstein equations. While this is tractable to find the evolution of e.g. some logarithmic terms, it quickly becomes too heavy to find the evolution of the mass and angular momentum. For this reason we then turn to the Newman–Penrose formalism in order to translate the partial Bondi gauge solution space and find more compact expressions for the evolution equations.

## Newman–Penrose formalism, Weyl scalars, and evolution equations

The Newman–Penrose (NP) formalism has been used by many authors to study BMS asymptotic symmetries and charges, and also to discuss the Newman–Unti gauge [82, 127, 128, 136, 161, 162]. Here, after having studied the solution space in the partial Bondi gauge using the tensorial form

of the Einstein equations, we turn to the NP formalism to study the evolution equations when the boundary sources $\beta_0$ and $U_0^A$ are set to zero. This also includes the case $\Lambda \neq 0$. We start by computing the Weyl scalars, which enables us to identify and generalize (to the partial Bondi gauge) the so-called covariant functionals. These are the covariant spin-2, angular momentum, complex mass, energy current, and radiation. We show (at least in the flat limit and without boundary sources) that the radiation and the energy current can be rewritten in terms of a generalized news, which is in turn introduced as a shear in (2.34). We also identify the violations of smooth peeling induced by the presence of the logarithmic terms. In addition, we explain how these terms break the covariance properties used in [124] to single out the covariant functionals and the asymptotic Einstein equations.

We show explicitly how the tensorial evolution equations are neatly repackaged in compact evolution equations for the covariant functionals. In the case $\Lambda \neq 0$ we obtain in particular the mass loss formula (2.76) which was recently derived in [67], and identify which combination of the Weyl scalars is sourcing it. In the case $\Lambda = 0$ we also study the NP evolution equations for the logarithmic terms. Finally, we explain how the radial dependency of the NP dyad for the angular metric affects the Weyl scalars and introduces a mixing in helicities [44, 54]. This will have important consequences when studying the subleading spin-$s$ NP charges.

## Symplectic potential

Assuming the absence of logarithmic terms, we compute the divergent and finite pieces of both the temporal and radial parts of the Einstein–Hilbert symplectic potential. This is done with an arbitrary cosmological constant and in the presence of the time-dependent sources for the boundary metric. The flat limit is also well-defined. The resulting expressions are quite lengthy, but we perform some useful consistency checks on them by setting $\beta_0$ and $U_0^A$ to zero. First, the divergent pieces in the radial part are shown, consistently, to be the sum of total variations and total derivatives, which therefore allows to implement symplectic renormalization. Then, in the BS gauge we rewrite the finite piece of the radial potential in (A)dS in the form (3.8), which generalizes the result of [67] to the case of non-trivial Weyl sector with $\delta\sqrt{q} \neq 0$. This also properly identifies the canonical momentum to $q_{AB}$ as the combination of the radiation tensor and the covariant spin-2 which appear in the (A)dS mass loss formula.

Although the potential obtained in NU gauge is very lengthy in the general case (with the enlarged solution space) and differs from that in BS gauge by several terms, we identify the simplest setup in which the BS and NU potentials already differ. This is when we consider the sector of the solution space where $(\Lambda, \beta_0, U_0^A, \delta\sqrt{q}) = 0$ but $q_{AB} \neq q_{AB}^\circ$. In this case, the potential (4.3) shows that the trace $C$ is conjugated to the 2d Ricci $R[q]$ and also shifts the momentum to $q_{AB}$. This sets the stage for the study of the charges in NU gauge which we will perform in forthcoming work.

## Symmetries and transformation laws

The partial Bondi gauge (2.1) together with the polyhomogeneous expansion (2.3) are enough to build the solution space and therefore determine the radial expansion of the metric. Once

this radial expansion is determined, one can find the asymptotic Killing vectors and derive the ensuing transformation laws. By computing these transformation laws we show that some "covariant functionals" derived from the Weyl scalars do not actually transform homogeneously under the subgroup $\mathrm{Diff}(S^2) \ltimes \mathrm{Weyl}$ because of the logarithmic terms[4].

In the partial Bondi gauge, the radial part $\xi^r$ of the vector field is determined a priori by free functions at all order in a polyhomogeneous expansion. In order to obtain the explicit field-dependency of these free functions, and therefore also see how many functions actually remain free and therefore parametrize the symmetries, one needs to complete the gauge to BS or NU. Since picking such a gauge amounts to a choice of radial coordinate, this is indeed what determines the coefficients in the expansion of $\xi^r$. We show that $\xi^r$ contains a single free function in BS gauge, appearing at order $\mathcal{O}(r)$ and parametrizing the Weyl transformations. In NU gauge however there is an additional free function at order $\mathcal{O}(r^0)$ parametrizing the radial translations. This function acts on the subleading term $C_{AB}$ in the angular metric, and in particular shifts its trace $C$. This explains the relationship between this extra mode in NU gauge and the extra radial symmetry [129]. In order to use the radial symmetry to fix $C$ (say, to zero), one must ensure that the symmetry is not large (i.e. physical). This will be answered by the computation of the charges.

Finally, by going to an appropriate basis for the asymptotic Killing vector fields (this is often called a choice of slicing [91, 95, 96, 123]) we show that their bracket in NU gauge forms an algebra (instead of an algebroid) given by

$$\big(\mathrm{diff}(\mathcal{I}^+) \leftplus \mathbb{R}_h\big) \leftplus \mathbb{R}_k. \tag{1.1}$$

Here $\mathrm{diff}(\mathcal{I}^+)$ stands for the diffeomorphisms $\mathrm{diff}(S^2)$ of the two-sphere, generated by superrotations, together with the temporal diffeomorphisms $\mathrm{diff}(\mathbb{R})$ generated by supertranslations (which is a slight abuse of language since here we cover also the case $\Lambda \neq 0$). The two abelian sectors $\mathbb{R}_h$ and $\mathbb{R}_k$ correspond respectively to the Weyl rescalings and the radial translations. Moreover, these five symmetry components have an arbitrary time-dependency. This algebra of vector fields is precisely the four-dimensional generalization of the algebra found in the three-dimensional Bondi–Weyl gauge [123]. This is the observation suggesting that, as in the three-dimensional case, the radial translation $k$ and the trace $C$ might be associated to a non-vanishing charge.

## 1.3   Outline

This article is divided into three parts. Section 2, the first and main part, is devoted to the construction and analysis of the partial Bondi gauge. This is where we build the general solution space, study the evolution equations, the asymptotic symmetries, and part of the symplectic potential. Then, all these results can be further specified to either the BS or the NU gauge. Section 3 contains some extra details about the BS gauge, and in particular the computation of the symplectic potential in the cases $\Lambda = 0$ and $\Lambda \neq 0$. We also explain there how the reduction to the BS gauge determines $\xi^r$. Section 4 also gives details about the NU gauge, and in particular about the role of the extra symmetry generator and the trace $C$. Our notations are gathered in appendix A.

---

[4]We still adopt the terminology "covariant functionals" for convenience.

As a word of caution, we should say that many of the calculations presented here (such as the solution space starting at subleading order and the Weyl scalars) are extremely lengthy and have been performed with Mathematica. We are happy to provide details about our code if necessary.

## 2 The partial Bondi gauge

This main section is devoted to the study of the partial Bondi gauge. After defining this partial gauge, we construct the associated solution space and use the Newman–Penrose formalism to study the evolution equations. We then study the residual symmetries and the associated transformation laws before analysing the symplectic potential.

### 2.1 Partial gauge fixing

We consider 4-dimensional coordinates $x^\mu = (u, r, x^A)$, where $u$ is a time coordinate labelling null geodesics, $r$ is a parameter along these geodesics (whose precise geometrical meaning will be discussed later on), and $x^A = (\theta, \phi)$ are angular coordinates in the transverse directions. The requirements that the normal vector $\partial_\mu u$ be null and that the angular coordinates be constant along the null rays translate into the conditions $g^{\mu\nu}(\partial_\mu u)(\partial_\nu u) = 0 = g^{\mu\nu}(\partial_\mu u)(\partial_\nu x^A)$, which in turn imply the three Bondi gauge fixing conditions $g^{uu} = 0 = g^{uA}$, or equivalently $g_{rr} = 0 = g_{rA}$. Line elements satisfying these three conditions parametrize what we call the partial Bondi gauge, and are of the form [1–4]

$$\mathrm{d}s^2 = e^{2\beta}\frac{V}{r}\mathrm{d}u^2 - 2e^{2\beta}\mathrm{d}u\,\mathrm{d}r + \gamma_{AB}(\mathrm{d}x^A - U^A\mathrm{d}u)(\mathrm{d}x^B - U^B\mathrm{d}u). \tag{2.1}$$

At this stage $V(u, r, x^A)$, $\beta(u, r, x^A)$, and $U^A(u, r, x^B)$ are four unspecified functions of the four spacetime coordinates, and the angular metric $g_{AB} = \gamma_{AB}$ is also unspecified. We denote the determinant of this latter by $\gamma := \det\gamma_{AB}$, and its associated covariant derivative by $\mathcal{D}_A$. In matrix form the metric and its inverse are given by

$$g_{\mu\nu} = \begin{pmatrix} e^{2\beta}\dfrac{V}{r} + \gamma_{AB}U^A U^B & -e^{2\beta} & -\gamma_{AB}U^B \\ -e^{2\beta} & 0 & 0 \\ -\gamma_{AB}U^B & 0 & \gamma_{AB} \end{pmatrix}, \quad g^{\mu\nu} = \begin{pmatrix} 0 & -e^{-2\beta} & 0 \\ -e^{-2\beta} & -e^{-2\beta}\dfrac{V}{r} & -e^{-2\beta}U^A \\ 0 & -e^{-2\beta}U^A & \gamma^{AB} \end{pmatrix}, \tag{2.2}$$

and we can already note that $\sqrt{-g} = e^{2\beta}\sqrt{\gamma}$. The Christoffel connection coefficients for this metric are given in appendix B.

The three Bondi gauge fixing conditions leading to the form (2.1) of the metric are what we choose to call the partial Bondi gauge. As we are about to see in details, after choosing a radial expansion for the angular metric this partial gauge is enough to build a solution space and to study the evolution equations as well as the residual symmetries. After doing so we will explain how to further reduce the partial gauge to a genuine gauge fixing. So far the gauge is only partial because there remains the freedom of further fixing a fourth condition. The partial gauge is indeed preserved under arbitrary $x^\mu$-dependent redefinitions $r \mapsto \tilde{r}(x)$ of the radial coordinate, under which we get $\mathrm{d}u\,\mathrm{d}r \mapsto (\partial\tilde{r}/\partial x^\mu)\,\mathrm{d}u\,\mathrm{d}x^\mu$. In this sense we can therefore understand the reduction of the partial

gauge to a "true" gauge as a choice of radial coordinate. As already mentioned in the introduction, the two choices discussed in the literature correspond to $i$) the Bondi–Sachs (BS) gauge, reached with the so-called determinant condition, and in which $r$ is the areal distance, and $ii$) the Newman–Unti (NU) gauge, reached with a condition on $\beta$, and in which $r$ is the affine parameter for the null vector $\partial_\mu u$. These two gauges are discussed respectively in sections 3 and 4, but we will also comment on the differences between them on numerous instances below.

We note that the partial Bondi gauge, which was also given this name in [163], played there an important role in the non-covariant $3 + 1$ Hamiltonian analysis with null hypersurfaces.

## 2.2  Solution space

We now study the Einstein equations $\mathrm{E}_{\mu\nu} := G_{\mu\nu} + \Lambda g_{\mu\nu} = 0$ in order to define the solution space in the partial Bondi gauge. This resolution follows closely [28, 67]. As usual, the equations in this gauge conveniently split into four hypersurface equations (the $(rr)$, $(rA)$, and $(ur)$ equations), five constraint and/or evolution equations (the trace-free $(AB)$, the $(uu)$, and the $(uA)$ equations), and one trivial equation (the trace of the $(AB)$ equation). In summary, the four hypersurface equations determine the radial expansion of the four free functions $V$, $\beta$, and $U^A$, and the remaining non-trivial equations contain constraints on the free data as well as evolution equations in retarded time $u$. While progressing through this hierarchy the equations get more and more involved and lengthy. For this reason, after presenting their tensorial form using the Einstein equations $\mathrm{E}_{\mu\nu} = 0$, we will also derive the evolution equations in terms of the Weyl scalars using the much more compact Newman–Penrose formalism.

In order to solve the Einstein equations and define the solution space, we need to choose a fall-off and a radial expansion for the angular metric. We choose this to be of the general form

$$\gamma_{AB} = r^2 q_{AB} + r C_{AB} + D_{AB} + \frac{1}{r}\Big( E_{lAB}\ln r + E_{AB}\Big)$$
$$+ \frac{1}{r^2}\Big( F_{l^2AB}(\ln r)^2 + F_{lAB}\ln r + F_{AB}\Big) + \mathcal{O}(r^{-3}), \qquad (2.3)$$

and therefore relax analyticity by allowing at this stage the presence of logarithmic terms[5]. This relaxation is still compatible with conformal compactification. As is well-known, in the case $\Lambda = 0$ the solution space contains a logarithmic branch which appears in the solution for $U^A$ at order $\mathcal{O}(r^{-3})$ [4]. When this logarithmic branch is not removed by hand (as is usually done), the stability of the solution space under the action of non-trivial residual symmetries necessarily requires logarithmic terms in the angular metric, as we explain in section 2.6. This can also be seen from the evolution equations, which necessarily require logarithmic terms in the transverse metric (2.3) if one wants to keep a non-trivial logarithmic term e.g. in $U^A$ at order $\mathcal{O}(r^{-3})$ [139]. This is part of the motivation for considering the general expansion (2.3). One can alternatively motivate the presence of these logarithmic terms by the will to be general and have a more relaxed solution space [138, 139, 164]. The modifications of the peeling due to these logarithmic terms will be discuss is section 2.4.1. We

---

[5]By abuse of notation we allow $\mathcal{O}(r^n)$ to contain logarithmic terms.

will also see throughout this section how the various logarithmic terms disappear when $\Lambda \neq 0$, in agreement with the Fefferman–Graham theorem [140, 165, 166].

Let us also point out at this stage that all the tensors appearing in (2.3) are arbitrary functions of $(u, x^A)$. Their traces are in particular unconstrained at this stage in the partial Bondi gauge. The traces are constrained in the BS gauge by the determinant condition (starting with $C_{AB}$), and in the NU gauge by the $(rr)$ Einstein equation and the requirement that $\partial_r \beta = 0$ (starting with $D_{AB}$ and thereby allowing for the trace $C$ to be non-vanishing).

At the end of the day, after determining the solution space the fall-offs of the components of the spacetime metric turn out to be

$$g_{uu} = \mathcal{O}(r^2), \qquad g_{ur} = \mathcal{O}(r^0), \qquad g_{uA} = \mathcal{O}(r^2), \qquad g_{AB} = \mathcal{O}(r^2). \qquad (2.4)$$

The last condition is standard and comes from the leading behavior of (2.3), while the first three are the boundary conditions used in BS gauge in [67]. These are relaxed with respect to e.g. [28, 125, 139, 167] due to the presence of the cosmological constant and/or the boundary sources. It is important to point out at this stage that these relaxed fall-offs are not choices, but rather consequences of the Einstein equations in the presence of a cosmological constant. Indeed, the goal of this section is to simply start from the partial Bondi gauge (2.1), together with (2.3), and to let the Einstein equations determine the solution space without imposing any additional restrictions (e.g. on the time dependency of $q_{AB}$, on the boundary metric, or on the logarithmic branches). The resulting solution space then admits the fall-offs (2.4). This is the logic we have followed in [123].

We recall that our conventions and notations are gathered in appendix A. In particular, we denote by $D_A$ the covariant derivative associated with $q_{AB}$. The angular indices are lowered and raised with this leading order metric $q_{AB}$, appart from when we write $\gamma^{AB}$, which is the true inverse to $\gamma_{AB}$.

## Equation $(rr)$

We start by solving the $(rr)$ Einstein equation, which determines the radial expansion of the function $\beta$. This equation does not depend on $\Lambda$ since $g_{rr} = 0$ in the partial Bondi gauge. Explicitly, we have

$$\mathrm{E}_{rr} = (2\partial_r \beta - \partial_r)\partial_r \ln \sqrt{\gamma} + \frac{1}{4}\partial_r \gamma^{AB} \partial_r \gamma_{AB}. \qquad (2.5)$$

The equation $\mathrm{E}_{rr} = 0$ is solved by the expansion

$$\beta = \beta_0 + \frac{\beta_1}{r} + \frac{\beta_2}{r^2} + \frac{1}{r^3}\big(\beta_{l3} \ln r + \beta_3\big) + \mathcal{O}(r^{-4}), \qquad (2.6)$$

with

$$\mathrm{E}_{rr} = \mathcal{O}(r^{-4}) \qquad \Rightarrow \qquad \beta_1 = 0, \qquad (2.7a)$$

$$\mathrm{E}_{rr} = \mathcal{O}(r^{-5}) \qquad \Rightarrow \qquad \beta_2 = \frac{1}{32}\big([CC] - 4D\big), \qquad (2.7b)$$

$$\mathrm{E}_{rr} = \mathcal{O}(r^{-6}) \qquad \Rightarrow \qquad \begin{cases} \beta_{l3} = -\dfrac{1}{4}E_l, \\ 48\,\beta_3 = -12E + 8[CD] + C^3 - C\left(D + \dfrac{11}{4}[CC]\right) + 6E_l, \end{cases} \qquad (2.7c)$$

where $[CD] = C^{AB}D_{AB}$ and $D = q^{AB}D_{AB}$ (we refer the reader to appendix A for a summary of the notations). As expected, since (2.5) is first order in $r$ the solution features an integration "constant" $\beta_0(u, x^A)$. This function parametrizes part of the induced boundary metric (2.26).

The solutions for the various coefficients in the expansion quickly become very lengthy, but luckily to compute e.g. the symplectic potential we only need to go all the way down to $\beta_3$. The other coefficients can be determined analytically using the results of appendix D on the expansion of the inverse angular metric and its determinant. In order to derive the evolution equations in the case $\Lambda \neq 0$ we need in particular to study the subleading structure of the solution space and to access $\beta_4$. These subleading terms are also used to illustrate later on the disappearance of the logarithmic branches in the case $\Lambda \neq 0$. This subleading structure of the solution space is reported in appendix F.

Let us now take a moment to explain the different interpretation of the $(rr)$ Einstein equation in the BS and NU gauges.

- In BS gauge the tensors entering the expansion (2.3) of the angular metric have their trace determined by the determinant gauge condition as in (3.2). The $(rr)$ Einstein equation then determines the expansion of $\beta$ in terms of various contractions and traces of the tensors appearing in the expansion of the angular metric.

- In NU gauge we have $\beta = \beta_0$ fixed by the gauge choice, so all the lower orders in $\beta$ vanish. Starting with (2.7b) this determines the traces of the tensors appearing in the expansion of the angular metric. Importantly, one should however notice that in NU gauge the trace $C$ is free. In section 4.2 below we will write the explicit form of the expansion of the angular metric which satisfies the trace conditions imposed by the $E_{rr}$ Einstein equation in NU gauge.

We therefore see the tradeoff between the BS and NU gauges. In both cases there are constraints on the traces of the tensors appearing in the expansion of the angular metric. In BS gauge these constraints are imposed by the determinant gauge condition, while in NU gauge these constraints are determined on-shell by the $E_{rr}$ Einstein equation. The important point is that, at the end of the day, the NU gauge has one more degree of freedom contained in the trace $C$.

## Equations $(rA)$

Once the $(rr)$ equation has been solved to determine the expansion (2.6) of $\beta$, we turn to the $(rA)$ equations to determine the expansion of $U^A$. Once again these two equations do not depend on $\Lambda$ since $g_{rA} = 0$. We have

$$E_{rA} = \frac{1}{2\sqrt{\gamma}}\partial_r\left(\sqrt{\gamma}\,e^{-2\beta}\gamma_{AB}\partial_r U^B\right) + (\partial_A\beta - \partial_A)\partial_r\ln\sqrt{\gamma} - \partial_A\partial_r\beta - \frac{1}{2}\gamma_{AC}\mathcal{D}_B\partial_r\gamma^{BC}. \quad (2.8)$$

Together with the above solution for $\beta$, this equation is solved by an expansion of the form

$$U^A = U_0^A + \frac{U_1^A}{r} + \frac{U_2^A}{r^2} + \frac{1}{r^3}\left(U_{l3}^A\ln r + U_3^A\right) + \mathcal{O}(r^{-4}), \quad (2.9)$$

with

$$\mathrm{E}_{rA} = \mathcal{O}(r^{-2}) \qquad \Rightarrow \qquad U_1^A = 2e^{2\beta_0}\partial^A\beta_0, \tag{2.10a}$$

$$\mathrm{E}_{rA} = \mathcal{O}(r^{-3}) \qquad \Rightarrow \qquad U_2^A = -\frac{1}{2}e^{2\beta_0}\left(2C^{AB}\partial_B\beta_0 + D_BC^{AB} - \partial^A C\right), \tag{2.10b}$$

$$\mathrm{E}_{rA} = \mathcal{O}(r^{-4}) \qquad \Rightarrow \qquad U_{l3}^A = -\frac{2}{3}e^{2\beta_0}D_B\mathfrak{L}^{AB}, \tag{2.10c}$$

where we have introduced the trace-free quantity

$$\mathfrak{L}_{AB} := D_{AB}^{\mathrm{TF}} - \frac{1}{4}CC_{AB}^{\mathrm{TF}}. \tag{2.11}$$

This new quantity, and later on similar objects denoted with `mathfrak` fonts, denotes an object which will be subject to constraints or evolution equations obtained from the $(AB)$ Einstein equations. We already see that such constraints are related to the fate of the logarithmic terms in the expansion.

Since the equation for $U^A$ is second order, we get as expected two integration "constants", which are the vectors $U_0^A(u, x^B)$ and $U_3^A(u, x^B)$. The former is part of the data parametrizing the induced boundary metric, while the latter is related to the angular momentum aspect (whose definition is not set in stone at this stage since it comes from a free integration constant).

## Equation $(ur)$

To study the $(ur)$ Einstein equation, it is useful to rewrite it in a more manageable form (although as we will see this can lead to subtleties). Using $R_{rr} = 0 = R_{rA}$ gives $R = 2g^{ur}R_{ur} + \gamma^{AB}R_{AB}$. Together with the convenient fact that $g^{ur} = (g_{ur})^{-1}$ in Bondi gauge, this then shows that $\mathrm{E}_{ur} = 0$ is equivalent to $\gamma^{AB}R_{AB} = 2\Lambda$. We have

$$\gamma^{AB}R_{AB} = R[\gamma] + e^{-2\beta}\partial_r\left(\frac{V}{r}\partial_r\ln\sqrt{\gamma} + \mathcal{D}_AU^A\right) + e^{-2\beta}\frac{V}{2r}\left(\frac{1}{\gamma}\det(\partial_r\gamma_{AB}) - \frac{1}{2}(\partial_r\gamma^{AB})(\partial_r\gamma_{AB})\right)$$
$$- e^{-2\beta}\left(\frac{1}{2}e^{-2\beta}\gamma_{AB}\partial_rU^A\partial_rU^B - \left(2\partial_u\ln\sqrt{\gamma} + 2\partial_u + 2\mathcal{D}_AU^A + U^A\partial_A\right)\partial_r\ln\sqrt{\gamma}\right)$$
$$- 2\partial_A\left(\gamma^{AB}\partial_B\beta\right) - 2\gamma^{AB}(\partial_A\beta)\partial_B(\ln\sqrt{\gamma} + \beta), \tag{2.12}$$

so the $(ur)$ equation rewritten in this form is first order in $\partial_r V$. As expected it will lead to a single integration constant, which is the Bondi mass aspect $M(u, x^A)$. The equation $\gamma^{AB}R_{AB} = 2\Lambda$ is solved by the expansion

$$V = r^3V_{+3} + r^2V_{+2} + rV_{+1} + V_{l0}\ln r + 2M + \mathcal{O}(r^{-1}), \tag{2.13}$$

with

$$\gamma^{AB}R_{AB} - 2\Lambda = \mathcal{O}(r^{-1}) \qquad \Rightarrow \qquad V_{+3} = \frac{\Lambda}{3}e^{2\beta_0}, \tag{2.14a}$$

$$\gamma^{AB}R_{AB} - 2\Lambda = \mathcal{O}(r^{-2}) \qquad \Rightarrow \qquad V_{+2} = \frac{\Lambda}{6}e^{2\beta_0}C - \partial_u\ln\sqrt{q} - D_AU_0^A, \tag{2.14b}$$

$$\gamma^{AB}R_{AB} - 2\Lambda = \mathcal{O}(r^{-3}) \qquad \Rightarrow \qquad V_{+1}, \tag{2.14c}$$

$$\gamma^{AB}R_{AB} - 2\Lambda = \mathcal{O}(r^{-4}) \qquad \Rightarrow \qquad V_{l0} = -\frac{\Lambda}{6}e^{2\beta_0}[C\mathfrak{L}] + \frac{\Lambda}{3}e^{2\beta_0}E_l, \tag{2.14d}$$

and

$$V_{+1} = -\frac{1}{2}e^{2\beta_0}\left(R[q] + \frac{\Lambda}{2}\left(\frac{3}{4}[CC] - \frac{C^2}{3} - D\right) + 4D_A\partial^A\beta_0 + 8(\partial^A\beta_0)(\partial_A\beta_0)\right)$$
$$-\frac{1}{4}\left(2\partial_u + \partial_u\ln\sqrt{q} + D_AU_0^A + 2U_0^A\partial_A\right)C. \tag{2.15}$$

Notice that here again, as for the previous set of Einstein equations, we see the quantity $\mathfrak{L}_{AB}$ appearing in the logarithmic term.

We comment in appendix G on an alternative route for determining the expansion in $V$ using the $(ur)$ equation. This seemingly treats the logarithmic terms in a different way, but at the end of the day, as it should, the solution space is unambiguously defined once the $(AB)$ Einstein equations are solved. This is because the two routes for determining $V$, which involve two different rewritings of the initial equation $\mathrm{E}_{ur} = 0$, differ by constraints coming from the remaining $(AB)$ Einstein equations. We now give the form of these constraints.

## Equations ($AB$)

The four Einstein equations discussed above have now determined the radial expansion of the free functions $(\beta, V, U^A)$. However, we still have three sets of equations to solve, namely $(AB)$, $(uu)$, and $(uA)$. These equations are going set constraints and temporal evolution equations on the data.

To study the angular equations $\mathrm{E}_{AB} = 0$ it is convenient to split them into trace-free and pure trace parts given by[6]

$$\mathrm{E}_{AB}^{\mathrm{TF}} = G_{AB} - \frac{1}{2}(\gamma^{CD}G_{CD})\gamma_{AB} = 0, \qquad\qquad \gamma^{AB}\mathrm{E}_{AB} = \gamma^{AB}G_{AB} + 2\Lambda = 0. \tag{2.16}$$

Let us now consider the Bianchi identity $\nabla_\mu G_\alpha^\mu = 0$ rewritten as $2\partial_\mu(\sqrt{-g}\,G_\alpha^\mu) + \sqrt{-g}\,G_{\mu\nu}\partial_\alpha g^{\mu\nu} = 0$, for $\alpha = r$. Using the fact that the non-trivial equations in $\mathrm{E}_{\mu r} = G_{\mu r} + \Lambda g_{\mu r} = 0$ have already been solved in the previous steps, we can further write the Bianchi identity as $G_{AB}\partial_r\gamma^{AB} \simeq \Lambda\partial_r\ln\gamma$ (the wiggly equality meaning that we have already used some EOMs). Using this it is then immediate to see that the trace-free part of (2.16), when satisfied, implies the trace part. We can therefore focus on the trace-free part. We find that its expansion starts as

$$\mathrm{E}_{AB}^{\mathrm{TF}}\big|_{\mathcal{O}(r)} = -e^{-2\beta_0}\mathfrak{B}_{AB}^\Lambda, \tag{2.17a}$$

$$\mathrm{E}_{AB}^{\mathrm{TF}}\big|_{\mathcal{O}(r^0)} = -\frac{1}{4}e^{-2\beta_0}\left(C\mathfrak{B}_{AB}^\Lambda + 2[C\mathfrak{B}^\Lambda]q_{AB}\right) - \frac{\Lambda}{3}\mathfrak{L}_{AB}, \tag{2.17b}$$

$$\mathrm{E}_{AB}^{\mathrm{TF}}\big|_{\mathcal{O}(r^{-1})} = \frac{1}{16}e^{-2\beta_0}\left(([CC] - 4D)\mathfrak{B}_{AB}^\Lambda - 16[\mathfrak{L}\mathfrak{B}^\Lambda]q_{AB} - 8C_{AC}^{\mathrm{TF}}C_{DB}^{\mathrm{TF}}\mathfrak{B}_\Lambda^{CD}\right)$$
$$- e^{-2\beta_0}\left(\partial_u\mathfrak{L}_{AB} + U_0^CD_C\mathfrak{L}_{AB} + \mathfrak{L}_{(AC}D_{B)}U_0^C\right) - \frac{\Lambda}{2}E_{lAB}^{\mathrm{TF}}, \tag{2.17c}$$

---

[6]Note that here the superscript in $\mathrm{E}_{AB}^{\mathrm{TF}}$ refers to the trace-free part in $\gamma_{AB}$, and *not* in $q_{AB}$. The various terms in the radial expansion of this trace-free Einstein equation will therefore not be trace-free in $q_{AB}$.

where we have introduced the objects

$$\mathfrak{B}_{AB}^{\Lambda} := \frac{\Lambda}{3} e^{2\beta_0} C_{AB}^{\mathrm{TF}} + \mathfrak{B}_{AB}^0, \tag{2.18a}$$

$$\mathfrak{B}_{AB}^0 := \left(\partial_u \ln \sqrt{q} - \partial_u\right) q_{AB} - \left(D_{(A} U_{B)}^0\right)^{\mathrm{TF}}. \tag{2.18b}$$

Just like $\mathfrak{L}_{AB}$ which has appeared previously and resurfaces here, these tensors are trace-free. We give them an interpretation in terms of a shear in section 2.3 below.

Let us take a moment to discuss the meaning of these equations before continuing. At the leading order, (2.17a) imposes that $\mathfrak{B}_{AB}^{\Lambda} = 0$. In the case $\Lambda = 0$, the requirement that $\mathfrak{B}_{AB}^0 = 0$ is simply relating the time evolution of the leading sphere metric $q_{AB}$ to the time evolution of its conformal factor, twisted by the presence of the non-vanishing boundary sources $U_0^A$. In the case $\Lambda \neq 0$, this equation is the statement that the shear $C_{AB}^{\mathrm{TF}}$ is not an independent free data, but is related to the boundary source $U_0^A$ and the time evolution of the boundary metric. This also shows, as is well-known, that there cannot be any asymptotic shear in the case $\Lambda \neq 0$ when the boundary conditions are chosen such that $U_0^A$ and the time evolution of the boundary metric vanish. At the next order, on-shell of $\mathfrak{B}_{AB}^{\Lambda} = 0$ equation (2.17b) imposes that $\Lambda \mathfrak{L}_{AB} = 0$. This is trivial in the case $\Lambda = 0$, but enforces the constraint $\mathfrak{L}_{AB} = 0$ in the case $\Lambda \neq 0$. This is the condition which removes the logarithmic branch in (2.9) in (A)dS. Finally, on-shell of $\mathfrak{B}_{AB}^{\Lambda} = 0$ only the last line of (2.17c) survives. In the case $\Lambda \neq 0$, the requirement that $\mathfrak{L}_{AB} = 0$ tells us that $E_{lAB}^{\mathrm{TF}}$ has to vanish as well. In the case $\Lambda = 0$ we obtain however an equation relating the time evolution of $\mathfrak{L}_{AB}$ to itself and to the boundary source $U_0^A$. This equation, which has appeared in [67] in BS gauge, is the generalization with boundary sources of the evolution equation in [138]. As expected, we see how the $(AB)$ Einstein equations constrain the logarithmic terms to vanish in the case $\Lambda \neq 0$. This is summarized in section 2.5 below.

From now on we could in principle impose everywhere the on-shell constraints $\mathfrak{B}_{AB}^{\Lambda} = 0 = \Lambda \mathfrak{L}_{AB}$ which have been obtained from (2.17a) and (2.17b), in order to display lighter equations. We refrain from doing so in most of this section (unless otherwise indicated) in order to really display the full structure of the $(AB)$, $(uu)$, and $(uA)$ Einstein equations which are obtained from the solution space defined above.

We now continue with the study of the Einstein equations $\mathrm{E}_{AB}^{\mathrm{TF}}$ at order $r^{-2}$. At this stage the equations start to get too lengthy to be displayed. The explicit expressions for these equations are however not needed, since we are more interested in their structure and their meaning. We are indeed going to recover these equations in compact form using the Newman–Penrose formalism in section 2.4, and here we are simply interested in identifying where the evolution equations appear. Let us therefore set $\beta_0 = 0 = U_0^A$ at this stage since it will not impact the conclusions we want to draw. For the next order we then find

$$\mathrm{E}_{AB}^{\mathrm{TF}}\big|_{\mathcal{O}(r^{-2})}^{\beta_0=0=U_0} = \frac{2\Lambda}{3} F_{l2\,AB}^{\mathrm{TF}} (\ln r)^2 - \mathrm{EOM}(E_{lAB}) \ln r - \frac{2}{3} \mathrm{EOM}(E_{AB}). \tag{2.19}$$

The general form of this equation off-shell of the constraints and with $\Lambda \neq 0$ is given in appendix I. For simplicity, let us first analyse the logarithmic term (I.2) on-shell of the constraints. This is

$$\mathrm{EOM}(E_{lAB}) = \left(2\partial_u + \partial_u \ln \sqrt{q}\right) E_{lAB}^{\mathrm{TF}} + \frac{1}{2}\left(D_{(A} U_{B)}^{l3}\right)^{\mathrm{TF}} - \frac{1}{2} E_l \mathfrak{B}_{AB}^0 + \frac{\Lambda}{3}\left(5 F_{l2\,AB}^{\mathrm{TF}} - 2 F_{lAB}^{\mathrm{TF}}\right). \tag{2.20}$$

With this equation we see the same mechanism as above, differentiating between vanishing and non-vanishing $\Lambda$. For $\Lambda \neq 0$ the squared logarithmic term sets $F_{l^2 AB}^{\mathrm{TF}} = 0$. On-shell of the previous equations (2.17) the logarithmic term then relates $F_{lAB}^{\mathrm{TF}}$ to $E_l \mathfrak{B}_{AB}^0$. The trace $E_l$ is however vanishing when the partial Bondi gauge is further reduced to BS or NU (either by the determinant condition in BS gauge, or by the vanishing of (2.7c) in NU gauge). This then implies the vanishing of $F_{lAB}^{\mathrm{TF}}$. In summary, these Einstein equations lead once again to constraints removing the logarithmic terms in the (A)dS case. In the case $\Lambda = 0$ however we obtain an evolution equation for $E_{lAB}^{\mathrm{TF}}$.

Going further, we now study $\mathrm{EOM}(E_{AB})$, whose general expression is given in appendix I. First let us display here this equation in the case $\Lambda = 0$, and furthermore go on-shell of the equations $\mathfrak{B}_{AB}^0 = 0$ and (2.20). We find

$$\mathrm{EOM}(E_{AB})\big|_{\Lambda=0} = \partial_u \mathcal{E}_{AB} - \frac{1}{2}\big(D_{(A}\mathcal{P}_{B)}\big)^{\mathrm{TF}} - \frac{3}{2}C_{AB}^{\mathrm{TF}}\mathcal{M}\big|_{\Lambda=0} + \frac{3}{2}q_{AC}\epsilon^{CD}C_{DB}^{\mathrm{TF}}\widetilde{\mathcal{M}}$$
$$+ \frac{1}{2}\left(\mathcal{E}_{AB} - 3E_{lAB}^{\mathrm{TF}} + \frac{1}{2}C\mathfrak{L}_{AB}\right)\partial_u \ln\sqrt{q} + \frac{1}{2}\left(\partial_u C + \frac{3}{2}R[q]\right)\mathfrak{L}_{AB}. \quad (2.21)$$

Here we have introduced the so-called covariant functionals. These are the spin-2 $\mathcal{E}_{AB}$ defined below in (2.39), the covariant mass $\mathcal{M}$ defined in (2.41) and such that in the absence of boundary sources $\mathcal{M}\big|_{\Lambda=0} := M + \frac{1}{16}\big(\partial_u + \partial_u \ln\sqrt{q}\big)\big(4D - [CC]\big)$, the covariant dual mass $\widetilde{\mathcal{M}}$ defined in (2.42), and the covariant angular momentum $\mathcal{P}_A$ defined in (2.40). We identify these quantities in section 2.4 as components of the Weyl tensor. The main thing to notice at this point is that $\mathrm{E}_{AB}^{\mathrm{TF}}\big|_{\mathcal{O}(r^{-2})}$ is the Einstein equation which constrains the time evolution of the spin-2.

In the case $\Lambda \neq 0$ however, one can see on (I.1) that $\mathrm{EOM}(E_{AB})\big|_{\Lambda \neq 0}$ is instead an algebraic equation which determines $F_{AB}^{\mathrm{TF}}$ in terms of $\partial_u E_{AB}^{\mathrm{TF}}$. Therefore, unlike in the flat case, the spin-2 is unconstrained and a completely free data in (A)dS.

At this point one can note, given the lengthy general expression which we have moved to appendix I, that the tensorial approach in which we compute the components of the Einstein equations by brute force is perhaps not the most appropriate one. We will indeed recover this evolution equation in a much more compact form using the Weyl scalars and the Newman–Penrose formalism below in section 2.4. This is also the efficient way of accessing the evolution equations for the higher spin charges, rather than continuing with the expansion of $\mathrm{E}_{AB}^{\mathrm{TF}}$.

Finally, as a cross-check we can verify that the trace part (2.16) does indeed contain redundant information. Its expansion is

$$\gamma^{AB}\mathrm{E}_{AB} = -\frac{1}{2r^2}e^{-2\beta_0}[C\mathfrak{B}^\Lambda] - \frac{1}{2r^3}\left(\frac{\Lambda}{3}[C\mathfrak{L}] + e^{-2\beta_0}\big(2[D\mathfrak{B}^\Lambda] - C[C\mathfrak{B}^\Lambda]\big)\right) + \mathcal{O}(r^{-4}), \quad (2.22)$$

which is clearly vanishing when the constraints (2.17) are enforced.

We are now left with three Einstein equations involving the retarded time $u$. On-shell of the constraints found above from the $(AB)$ equations, these remaining Einstein equations give evolution equations in $u$, in particular for the mass and angular momentum aspects.

### Equation ($uu$)

This is the Einstein equation which gives the time evolution of the mass aspect. The expression is however too lengthy to be displayed here. Fortunately, we will be able below to write a compact

form of this evolution equation using the covariant functionals in the Newman–Penrose formalism. For the time being, let us simply mention where the evolution equation appears. For this, we can restrict ourselves for simplicity to the case $\beta_0 = 0 = U_0^A$. We then find

$$\mathrm{E}_{uu}\big|_{\mathcal{O}(r^0)}^{\beta_0=0=U_0} = -\frac{1}{4}[\mathfrak{B}^0\mathfrak{B}^\Lambda], \tag{2.23a}$$

$$\mathrm{E}_{uu}\big|_{\mathcal{O}(r^{-1})}^{\beta_0=0=U_0} = -\frac{\Lambda}{6}[\mathfrak{L}\mathfrak{B}^0] + \frac{1}{2}[N\mathfrak{B}^\Lambda] - \frac{1}{2}D^AD^B\mathfrak{B}_{AB}^\Lambda - \frac{1}{4}[C\mathfrak{B}^\Lambda]\partial_u\ln\sqrt{q}, \tag{2.23b}$$

$$\mathrm{E}_{uu}\big|_{\mathcal{O}(r^{-2})}^{\beta_0=0=U_0} = -\frac{\Lambda}{2}\left(D_A U_{l3}^A + \frac{1}{3}[N\mathfrak{L}] - \frac{1}{6}[C\mathfrak{L}]\partial_u\ln\sqrt{q} + \frac{1}{3}[C^{\mathrm{TF}}\partial_u\mathfrak{L}] - \frac{1}{2}[E_l\mathfrak{B}^0]\right)\ln r$$
$$+ \partial_u\mathcal{M} + (\dots). \tag{2.23c}$$

This shows, as announced, that on-shell of the previous Einstein equations, the $(uu)$ equation controls the time evolution of the mass aspect $M$, or equivalently of the covariant mass $\mathcal{M}$.

## Equation $(uA)$

We finally turn to the remaining two Einstein equations, which contain the time evolution of the angular momentum aspect. For generality, let us go once again off-shell of the $(AB)$ equations. We then find

$$\mathrm{E}_{uA}\big|_{\mathcal{O}(r)} = -e^{-2\beta_0}\mathfrak{B}_{AB}^\Lambda U_0^B, \tag{2.24a}$$

$$\mathrm{E}_{uA}\big|_{\mathcal{O}(r^0)} = \frac{\Lambda}{3}\mathfrak{L}_{AB}U_0^B + \frac{1}{4}e^{-2\beta_0}\left(C\mathfrak{B}_{AB}^\Lambda U_0^B + 3[C\mathfrak{B}^\Lambda]U_A^0\right) - \frac{1}{2}D^B\mathfrak{B}_{AB}^\Lambda + 2\mathfrak{B}_{AB}^\Lambda\partial^B\beta_0. \tag{2.24b}$$

At the next order the equation gets too lengthy to be displayed, so let us consider the case of vanishing sources $\beta_0 = 0 = U_0^A$, again without loss of generality concerning the general structure of the equations. We then find

$$\mathrm{E}_{uA}\big|_{\mathcal{O}(r^{-1})}^{\beta_0=0=U_0} = \frac{1}{4}\left(2\Lambda U_A^{l3} + C_{\mathrm{TF}}^{BC}D_A\mathfrak{B}_{BC}^\Lambda + \mathfrak{B}_{AB}^\Lambda D_C C_{\mathrm{TF}}^{BC} - \mathfrak{B}_\Lambda^{BC}D_B C_{CA}^{\mathrm{TF}} - 4C_{CA}^{\mathrm{TF}}D_B\mathfrak{B}_\Lambda^{BC}\right), \tag{2.25a}$$

$$\mathrm{E}_{uA}\big|_{\mathcal{O}(r^{-2})}^{\beta_0=0=U_0} = \frac{1}{2}\left[\Lambda\left(\frac{1}{2}D^B E_{lBA}^{\mathrm{TF}} + \frac{1}{6}D_A[C\mathfrak{L}] - U_{l3}^B\mathfrak{B}_{AB}^\Lambda\right) - 3\left(\partial_u U_A^{l3} + U_A^{l3}\partial_u\ln\sqrt{q}\right)\right]\ln r$$
$$+ \partial_u\mathcal{P}_A + (\dots). \tag{2.25b}$$

The equations (2.24a), (2.24b), and (2.25a) are trivial. Indeed, on-shell we have on the one-hand that $\mathfrak{B}_{AB}^\Lambda = 0$, and on the other hand that $\Lambda U_A^{l3} = 0 = \Lambda\mathfrak{L}_{AB}$ (either because $\Lambda = 0$ in the flat case, or because $\mathfrak{L}_{AB} = 0$ when $\Lambda \neq 0$).

The key point to observe here is that equation (2.25b) gives the time evolution of the (covariant) angular momentum aspect $\mathcal{P}_A$, as well as an evolution constraint on the logarithmic term (2.10c). Once again, we postpone the investigation of these equations in compact form to section 2.4, where we will recover them using the NP formalism.

## 2.3 Geometry

We now comment on the geometrical meaning of some of the quantities encountered above. First, we compute the induced boundary metric to find [67]

$$ds^2\big|_{\mathcal{I}^+} = \lim_{r\to\infty}\left(\frac{ds^2}{r^2}\right) = \frac{\Lambda}{3}e^{4\beta_0}du^2 + q_{AB}\left(dx^A - U_0^A du\right)\left(dx^B - U_0^B du\right). \tag{2.26}$$

This shows as expected that the integration "constants" $\beta_0$ and $U_0^A$, together with $q_{AB}$, parametrize the boundary metric, and that this latter becomes degenerate in the flat limit.

Let us now introduce vectors whose shear will be related to the constraints derived above, and which will be used in the Newman–Penrose formalism. First, we consider the two null vectors

$$\ell^\mu\partial_\mu := \partial_r, \qquad n^\mu\partial_\mu := e^{-2\beta}\left(\partial_u + \frac{V}{2r}\partial_r + U^A\partial_A\right), \tag{2.27}$$

such that $\ell^\mu n_\mu = -1$. The vector $\ell$ is outgoing and tangent to the null geodesics intersecting $\mathcal{I}^+$ along 2-spheres at constant $u$. The vector $n$ is ingoing, tangent to $\mathcal{I}^+$, transverse to $\ell$, and forms with this latter the binormal $(\ell, n)$ orthogonal to the spacelike slices tangent to the 2-spheres. Its radial expansion is given in (C.7). In addition, let us consider the vectors

$$v^\mu\partial_\mu := e^{2\beta}n - \frac{V}{2r}\ell = \partial_u + U^A\partial_A, \qquad w^\mu\partial_\mu := e^{2\beta}n + \frac{V}{2r}\ell = \left(\partial_u + \frac{V}{r}\partial_r + U^A\partial_A\right), \tag{2.28}$$

which satisfy

$$v^\mu w_\mu = 0, \qquad v^2 = -w^2 = e^{2\beta}\frac{V}{r}, \qquad \ell^\mu v_\mu = \ell^\mu w_\mu = -e^{2\beta}, \qquad n^\mu v_\mu = -n^\mu w_\mu = \frac{V}{2r}. \tag{2.29}$$

We are interested in the 2-dimensional asymptotic shear of these vectors. For this, we define the extrinsic curvature

$$K_{\mu\nu}(X) = \frac{1}{2}h_\mu^\alpha h_\nu^\beta \pounds_X g_{\alpha\beta}, \qquad h_\mu^\alpha = \delta_\mu^\alpha + n^\alpha\ell_\mu + \ell^\alpha n_\mu, \qquad h_\mu^\alpha n^\mu = h_\mu^\alpha \ell^\mu = 0, \tag{2.30}$$

which gives

$$K_{AB}(\ell) = \frac{1}{2}\pounds_\ell g_{AB} = \frac{1}{2}\partial_r\gamma_{AB}, \tag{2.31a}$$

$$K_{AB}(n) = \frac{1}{2}\pounds_n g_{AB} = \frac{1}{2}e^{-2\beta}\left(\partial_u\gamma_{AB} + \gamma_{(AC}\mathcal{D}_{B)}U^C + \frac{V}{2r}\partial_r\gamma_{AB}\right), \tag{2.31b}$$

$$K_{AB}(v) = \frac{1}{2}\pounds_v g_{AB} = \frac{1}{2}\left(\partial_u\gamma_{AB} + \gamma_{(AC}\mathcal{D}_{B)}U^C\right), \tag{2.31c}$$

$$K_{AB}(w) = \frac{1}{2}\pounds_w g_{AB} = \frac{1}{2}\left(\partial_u\gamma_{AB} + \gamma_{(AC}\mathcal{D}_{B)}U^C + \frac{V}{r}\partial_r\gamma_{AB}\right). \tag{2.31d}$$

Taking the trace-free parts in the transverse metric $\gamma_{AB}$ then leads to the shears

$$S_{AB}(\ell) = -\frac{1}{2}C_{AB}^{\mathrm{TF}} + \mathcal{O}(r^{-1}), \tag{2.32a}$$

$$S_{AB}(n) = -\frac{1}{4}e^{-2\beta_0}\big(\mathfrak{B}_{AB}^{\Lambda} + \mathfrak{B}_{AB}^{0}\big)r^2 + rS_{AB}^{+1}(n) + \mathcal{O}(r^0), \tag{2.32b}$$

$$S_{AB}(v) = -\frac{1}{2}\mathfrak{B}_{AB}^{0}r^2 + \mathcal{O}(r), \tag{2.32c}$$

$$S_{AB}(w) = -\frac{1}{2}\mathfrak{B}_{AB}^{\Lambda}r^2 + \mathcal{O}(r). \tag{2.32d}$$

This shows as expected that $C_{AB}^{\mathrm{TF}}$ is the asymptotic shear of the null congruence, and also gives an interpretation of the quantity $\mathfrak{B}_{AB}^{\Lambda}$ as a shear which is set to vanish by the $(AB)$ Einstein equations.

It is interesting to note that the term of order $\mathcal{O}(r)$ in the shear of $n$ is

$$S_{AB}^{+1}(n) = \frac{1}{2}e^{-2\beta_0}\left[\left(\partial_u + \pounds_{U_0} + \frac{1}{2}V_{+2}\right)C_{AB}^{\mathrm{TF}} - \frac{1}{2}C\mathfrak{B}_{AB}^{\Lambda} + \big(D_{(A}U_{B)}^{1}\big)^{\mathrm{TF}}\right]$$
$$- \frac{\Lambda}{24}\big(4\mathfrak{L}_{AB} + [CC^{\mathrm{TF}}]q_{AB}\big). \tag{2.33}$$

From this we recover the fact that in the flat case, on-shell of $\mathfrak{B}_{AB}^{0} = 0$, without boundary sources and without time-dependency in $\sqrt{q}$, the leading order of $S_{AB}(n)$ is the news[7] $N_{AB} = \partial_u C_{AB}^{\mathrm{TF}}$ [17]. In the general case, this suggests to introduce a generalization of the news, which up to a numerical factor and on-shell of $\mathfrak{B}_{AB}^{\Lambda} = 0 = \Lambda\mathfrak{L}_{AB}$ is defined from the shear of $n$ as

$$\mathbf{N}_{AB} := e^{-2\beta_0}\left[\left(\partial_u + \pounds_{U_0} - \frac{1}{2}\big(\partial_u\ln\sqrt{q} + D_A U_0^A\big)\right)C_{AB}^{\mathrm{TF}} + \big(D_{(A}U_{B)}^{1}\big)^{\mathrm{TF}}\right]$$
$$+ \frac{\Lambda}{12}\big(CC_{AB}^{\mathrm{TF}} - [CC^{\mathrm{TF}}]q_{AB}\big). \tag{2.34}$$

In the flat case, this is the generalization of the news in the presence of a free time-dependent boundary metric. In (A)dS this interpretation is not immediate since one should replace the shear $C_{AB}^{\mathrm{TF}}$ using $\mathfrak{B}_{AB}^{\Lambda} = 0$. We will see below that this generalized news appears, as expected, in the dual mass and the energy current.

We can now use the properties of the null vector $\ell$ to discuss the different roles played by the radial coordinate in the NU and BS gauges. First, we compute the parallel transport of $\ell$ along itself is, which is

$$\nabla_\ell \ell = (2\partial_r\beta)\partial_r. \tag{2.35}$$

Since $\ell^\mu = \mathrm{d}x^\mu/\mathrm{d}r$, the fact that in the NU gauge $\nabla_\ell\ell \overset{\mathrm{NU}}{=} 0$ (by virtue of the gauge condition $\partial_r\beta = 0$) implies that in this case $r$ is the affine parameter for $\ell$. Of course, $r$ is however defined only up to an affine transformation $r \mapsto ar + b$ corresponding to a choice of scaling and origin. It is precisely this choice of origin which is typically used to set $C = 0$ in the NU gauge (see e.g.

---

[7]We thank Ali Seraj for bringing this to our attention, and for suggesting to study the shear of $n$.

[127]). We will see below however that there is a symmetry generator associated with this choice of origin in NU gauge. This therefore raises the question of whether this symmetry could be large and associated to a non-vanishing charge, as in the three-dimensional case [123].

Finally, in order to understand the role of the radial coordinate in BS gauge, we compute the expansion of $\ell$ to find

$$\Theta(\ell) = g^{\mu\nu} K_{\mu\nu}(\ell) = \partial_r \ln \sqrt{\gamma} = \frac{r^4}{2\gamma} \partial_r \left( \frac{\gamma}{r^4} \right) + \frac{2}{r}. \tag{2.36}$$

This expansion is the rate of growth of the cross-sectional area of the geodesic congruence as one moves along the parameter $r$. When the determinant condition is imposed in order to reach the BS gauge, this area is proportional to $r^2$ as in flat spacetime, and $r$ is called the areal distance.

## 2.4 Newman–Penrose formalism

In section 2.2 we have obtained the solution space, but we are still missing part of the evolution equations (although we have identified where they are encoded in the tensorial Einstein equations $E_{\mu\nu} = 0$). An efficient and elegant way of obtaining these evolution equations is to use the Newman–Penrose (NP) spin coefficient formalism [8, 9, 168, 169], where they are encoded in the Bianchi identities. To setup the calculation, let us introduce the null frame fields

$$e_1 := \ell, \qquad e_2 := n, \qquad e_3 := \hat{m} = \sqrt{\frac{\gamma_{\theta\theta}}{2\gamma}} \left( \frac{\sqrt{\gamma} + i\gamma_{\theta\phi}}{\gamma_{\theta\theta}} \partial_\theta - i\partial_\phi \right), \qquad e_4 := \hat{\bar{m}}. \tag{2.37}$$

These are such that $\ell^\mu n_\mu = -1 = -\hat{m}^\mu \hat{\bar{m}}_\mu$ with all the other contractions vanishing. The spacetime metric[8] is $g^{\mu\nu} = e_i^\mu e_j^\nu \eta^{ij}$, where $\eta$ is the Minkowski metric with $\eta^{12} = -1$ and $\eta^{34} = +1$, and the angular metric is $\gamma_{AB} = \hat{m}_{(A}\hat{\bar{m}}_{B)}$. Notice that because of the doubly-null choice of internal metric the dyad on the sphere is complex.

The NP formalism relies on the use of the spin coefficients, whose expression in the partial Bondi gauge is given in appendix C. One can check that these expressions are consistent with the geometrical meaning of the coefficients[9]. We have in particular that $\kappa = 0$ since $\ell$ forms a null congruence, and $\epsilon + \bar{\epsilon} = 2\partial_r \beta$ since these geodesics are not affinely parametrized (unless we are in NU gauge). Since $\rho \in \mathbb{R}$ this null congruence is hypersurface orthogonal (and the rotational optical scalar $\omega = \text{Im}(\rho)$ is vanishing), meaning that $\ell$ is *proportional* to the gradient of a scalar field. It is however not *equal* to the gradient of a scalar field since $\bar{\alpha} + \beta - \tau = -2\hat{m}^A \partial_A \beta \neq 0$. Finally, the tetrads $n, \hat{m}, \hat{\bar{m}}$ have a non-trivial parallel transport along $\ell$ since $\pi \neq 0 \neq \epsilon - \bar{\epsilon}$ [168].

We also give in appendix C the radial expansion of the spin coefficients, which is needed in order to write down the expansion of the NP evolution equations. These evolution equations are

---

[8]Note that our choice of signature for the metric is opposite to that in e.g. [8, 9, 168, 169]. As a result, the NP evolution equations (2.53) have a change of sign in front of all the spin coefficients.

[9]It is always possible to perform gauge transformations of the tetrads so as to set some of the coefficients to zero, but this is not necessary for our purposes. We note that in the tetrad formulation of gravity, where the frames are the fundamental variables, it could be that some of these transformations are not pure gauge but become large.

expressed in terms of the Weyl scalars, which we now compute, before studying the case $\Lambda = 0$ and then the Bondi mass loss for (A)dS.

Contrary to the previous section where we have stayed off-shell of the constraints coming from the Einstein equations in order to display the structure of the said equations, here we are going to make repeated use of the constraint $\mathfrak{B}^\Lambda_{AB} = 0$ as well as $\Lambda \mathfrak{L}_{AB} = 0$ (which holds either trivially in the flat case, of because the logarithmic term has to vanish in (A)dS). One must therefore keep in mind that most of the expressions written here are obtained on-shell in this sense.

### 2.4.1 Weyl scalars and covariant functionals

The Weyl scalars are defined as contractions of the Weyl tensor with the null tetrad (2.37), which can then be expanded in terms of the dyad for the angular metric. In order to compute these Weyl scalars we first define the rescaled frame $m^A := r\hat{m}^A$. With this, on-shell we find

$$\Psi_0 := -W_{\ell m \ell m} = \frac{1}{r^4}\mathfrak{L}_{AB}m^A m^B + \frac{\ln r}{r^5}3E^{\text{TF}}_{lAB}m^A m^B + \frac{1}{r^5}\mathcal{E}_{AB}m^A m^B + \mathcal{O}(r^{-6}), \tag{2.38a}$$

$$\Psi_1 := -W_{\ell n \ell m} = \frac{\ln r}{r^4}D^B\mathfrak{L}_{AB}m^A + \frac{1}{r^4}\mathcal{P}_A m^A + \mathcal{O}(r^{-5}), \tag{2.38b}$$

$$\Psi_2 := -W_{\ell m \bar{m} n} = -\frac{1}{2}\left(W_{\ell n \ell n} - W_{\ell n m \bar{m}}\right) = \frac{1}{r^3}\left(\mathcal{M} + i\widetilde{\mathcal{M}}\right) + \mathcal{O}(r^{-4}), \tag{2.38c}$$

$$\Psi_3 := -W_{n \bar{m} n \ell} = \frac{1}{r^2}\mathcal{J}_A \bar{m}^A + \mathcal{O}(r^{-3}), \tag{2.38d}$$

$$\Psi_4 := -W_{n \bar{m} n \bar{m}} = \frac{1}{r}\mathcal{N}_{AB}\bar{m}^A \bar{m}^B + \mathcal{O}(r^{-2}). \tag{2.38e}$$

It is important to note that this is not yet a true expansion in $r^{-1}$ since the frame $m^A$ itself is $r$-dependent. We come back to this important point below. The coefficients appearing here are however $r$-independent, and given by functionals which we now discuss.

First, in $\Psi_0$, after two terms due to the logarithmic branches, we find the so-called covariant spin-2 aspect defined as

$$\mathcal{E}_{AB} := 3E^{\text{TF}}_{AB} - \frac{5}{2}E^{\text{TF}}_{lAB} - \frac{1}{2}C\mathfrak{L}_{AB} + \frac{3}{16}C^{\text{TF}}_{AB}\big([CC] - 4D\big). \tag{2.39}$$

In $\Psi_1$ we find the covariant momemtum

$$\mathcal{P}_A := -\frac{3}{2}e^{-2\beta_0}U^A_3 - \mathfrak{L}_{AB}\partial^B\beta_0 - \frac{4}{3}D^B\mathfrak{L}_{AB} + \frac{3}{32}e^{2\beta_0}\partial_A\big(e^{-2\beta_0}\big(4D - [CC]\big)\big)$$
$$+ \frac{3}{4}C_{AB}\big(C^{BC}\partial_C\beta_0 - \partial^B C + D_C C^{BC}\big). \tag{2.40}$$

In $\Psi_2$ we find the covariant mass

$$\mathcal{M} := e^{-2\beta_0}M + \frac{1}{16}e^{-2\beta_0}\big(\partial_u + \partial_u \ln \sqrt{q} + U^A_0\partial_A + D_A U^A_0\big)\big(4D - [CC]\big)$$
$$+ \big(\partial^A C - D_B C^{AB}\big)\partial_A\beta_0 + \frac{\Lambda}{24}\big(E_l - 4E\big), \tag{2.41}$$

and the covariant dual mass

$$\widetilde{\mathcal{M}} := \left(\frac{1}{8}\mathbf{N}_{AB} - \frac{1}{4}D_A D_B\right)\epsilon^{BC}\left(C_C^A\right)^{\mathrm{TF}}, \tag{2.42}$$

where $\epsilon^{AB} = \varepsilon^{AB}/\sqrt{q}$ is the Levi–Civita tensor with respect to the leading metric and $\varepsilon^{AB}$ the Levi–Civita symbol. Here we see the explicit appearance of the generalized news (2.34). The mass and its dual together define the complex mass $\mathcal{M}_{\mathbb{C}} := \mathcal{M} + i\widetilde{\mathcal{M}}$.

The covariant mass $\mathcal{M}$, angular momentum $\mathcal{P}_A$, and the tensor $\mathcal{E}_{AB}$, which have respective spin $0, 1, 2$, are built from $M$, $U_A^3$, and $E_{AB}$, which are the terms of order $r^{-1}$ in the $g_{uu}$, $g_{uA}$, and $g_{AB}$ components of the metric. They are responsible in the flat case for the leading, subleading, and sub-subleading soft graviton theorems respectively [55]. Interestingly, the spin-2 is also involved in a subleading symmetry and a class of memory observables which are non-local in time [56, 170, 171].

The other covariant functionals are a bit lengthy to write down in the general case, so we turn off some boundary sources in order to keep the expressions simple. In $\Psi_3$ we encounter the so-called covariant energy current, which for $U_0^A = 0$ is given by

$$\mathcal{J}_A := \frac{1}{2}D^B\mathbf{N}_{AB} + \frac{1}{4}\partial_A R[q] - \frac{\Lambda}{6}\left(\mathcal{P}_A + \frac{1}{4}\left(C_{\mathrm{TF}}^{BC}D_B C_{CA}^{\mathrm{TF}} - C_{AB}^{\mathrm{TF}}D_C C_{\mathrm{TF}}^{BC} + D^B(CC_{AB}^{\mathrm{TF}})\right)\right). \tag{2.43}$$

Finally, $\Psi_4$ contains the radiation tensor, which for $\beta_0 = 0$ and $U_0^A = 0$ is given by the trace-free expression

$$\begin{aligned}
\mathcal{N}_{AB} := &\frac{\Lambda^2}{36}\mathcal{E}_{AB} - \frac{\Lambda^2}{24}[CC^{\mathrm{TF}}]C_{AB}^{\mathrm{TF}} - \frac{\Lambda}{12}R[q]C_{AB}^{\mathrm{TF}} \\
&+ \frac{1}{2}\partial_u\left(N_{AB} - \frac{1}{2}C_{AB}^{\mathrm{TF}}\partial_u \ln\sqrt{q}\right) - \frac{\Lambda}{4}q_{AB}\left([NC^{\mathrm{TF}}] - \frac{1}{2}[CC^{\mathrm{TF}}]\partial_u \ln\sqrt{q}\right) \\
&- \frac{1}{2}\left(D_A\partial_B - \frac{1}{2}q_{AB}D_C\partial^C\right)\partial_u \ln\sqrt{q} + \frac{\Lambda}{12}\left(D_{(A}D^C C_{B)C}^{\mathrm{TF}} - q_{AB}D^C D^D C_{CD}^{\mathrm{TF}}\right). \tag{2.44}
\end{aligned}$$

One can see that this radiation tensor is non-vanishing even in the case $N_{AB} = 0$ of vanishing news, signaling that the enlarged solution space which we are considering allows for shear-free gravitational radiation [172, 173]. This is sourced by the non-trivial time-dependency of the boundary metric and by the covariant spin-2, and is a necessary ingredient in order to have non-trivial radiation in (A)dS.

One should however keep in mind that there is so far no well-understood notion of news and radiation in (A)dS, so the name of $\mathcal{N}_{AB}$ might be a misnomer. It is however possible to identify from this tensor an important quantity which appears in the (A)dS mass loss and which is conjugated to $q_{AB}$ in the (A)dS symplectic structure. This is the trace-free tensor

$$\mathcal{S}^{AB} := \mathcal{N}^{AB} + \frac{\Lambda^2}{36}\mathcal{E}^{AB}. \tag{2.45}$$

In BS gauge this reduces exactly to $\mathcal{S}^{AB} = -\Lambda^2 J^{AB}/6$ where $J^{AB}$ is (2.41) of [67].

It is interesting to notice that in the flat case and without boundary sources the radiation tensor can be rewritten in terms of the time derivative of the generalized news as

$$\mathcal{N}_{AB}\big|_{\Lambda=0} = \frac{1}{2}\partial_u\mathbf{N}_{AB}\big|_{\Lambda=0} - \frac{1}{2}\left(D_A\partial_B - \frac{1}{2}q_{AB}D_C\partial^C\right)\partial_u \ln\sqrt{q}. \tag{2.46}$$

This can be rewritten more compactly if we introduce the Geroch tensor [174] (see also [34, 175, 176]). For this, we use the fact that any metric on the two-sphere can be written as $q_{AB} = e^{\Phi} q^{\circ}_{AB}$, where $\Phi$ is a time-dependent conformal factor (sometimes called the superboost field [175]) and $q^{\circ}_{AB}$ is the metric of the round two-sphere with $R[q^{\circ}] = 2$. Introducing the shifted generalized news

$$\hat{\mathbf{N}}_{AB} := \mathbf{N}_{AB} - T_{AB}, \qquad T_{AB} := \left( D_A \partial_B \Phi + \frac{1}{2} \partial_A \Phi \, \partial_B \Phi \right)^{\mathrm{TF}}, \qquad (2.47)$$

we then find that the radiation (2.46) can be written as

$$\mathcal{N}_{AB}\big|_{\Lambda=0} = \frac{1}{2} \partial_u \hat{\mathbf{N}}_{AB}. \qquad (2.48)$$

The Geroch tensor $T_{AB}$ is sometimes called the vacuum news. Furthermore, since the Geroch tensor satisfies

$$D^A T_{AB} + \frac{1}{2} \partial_B R[q] = 0, \qquad (2.49)$$

we see that the first two terms in the covariant energy current are

$$\mathcal{J}_A\big|_{\Lambda=0} = \frac{1}{2} D^B \hat{\mathbf{N}}_{AB}. \qquad (2.50)$$

We therefore consistently recover the relationship between the radiation, the energy current, and the shifted news (which is also called covariant news) in the case of a time-dependent boundary metric. It would be interesting to study further the proper notion of shifted news in (A)dS and/or in the presence of the boundary sources.

We now turn to an important remark about the Weyl scalars and the covariant functionals computed above. These "covariant functionals" generalize the expressions identified in [124, 125], using a symmetry argument, as the objects transforming as primary fields (i.e. homogeneously) under the action of $\mathrm{Diff}(S^2) \ltimes \mathrm{Weyl}$. We will see below, when computing the transformation laws, that the presence of the logarithmic terms does however break the covariance property of some of these functionals. We therefore use the terminology "covariant" in an abuse of language, to simply denote the fact that these expressions are identified as components of the Weyl tensor, and that they generalize the expressions of [124, 125].

An important point to notice is that the scalars (2.38) are computed from the projection of the Weyl tensor onto the full $r$-dependent spacetime tetrad (2.37), and not from the projection onto the leading components of this tetrad. Because of this, the expressions in (2.38) as they stand are not genuine expansions in $r^{-1}$, since the frame $m^A$ itself is $r$-dependent. We will see below the important implications of this simple fact for the consistency of the formalism.

Finally, we should comment on the fact that the inclusion of logarithmic terms in our solution space also has an impact on the peeling properties of the Weyl tensor. Indeed, we see in (2.38) that the fall-off of this latter depends on whether the logarithmic terms are present or not. In the absence of logarithmic branches (which is always the case in (A)dS while it is a choice in the flat case), we have the standard Penrose–Sachs peeling with fall-offs $\Psi_k = \mathcal{O}(r^{k-5})$ for $k \in \{0, 1, 2, 3, 4\}$.

In the presence of the logarithmic branches we get the relaxed and non-smooth fall-offs

$$\Psi_0 = \frac{\Psi_0^4}{r^4} + \frac{1}{r^5}\big(\Psi_0^{l5}\ln r + \Psi_0^5\big) + \mathcal{O}(r^{-6}), \tag{2.51a}$$

$$\Psi_1 = \frac{1}{r^4}\big(\Psi_1^{l4}\ln r + \Psi_1^4\big) + \mathcal{O}(r^{-5}), \tag{2.51b}$$

$$\Psi_k = \mathcal{O}(r^{k-5}) \text{ for } k \in \{2, 3, 4\}. \tag{2.51c}$$

While this peeling is obviously weaker than the Penrose–Sachs condition, one should keep in mind that this latter is simply an hypothesis, which might be too restrictive as has been discussed by several authors [126, 138, 141–147]. The fall-offs (2.51) are for example consistent with the hypothesis used in the study of the global non-linear stability of Minkowski spacetime [177], which is

$$\Psi_k = \mathcal{O}(r^{-7/2}) \text{ for } k \in \{0, 1, 2\}, \qquad\qquad \Psi_k = \mathcal{O}(r^{k-5}) \text{ for } k \in \{3, 4\}. \tag{2.52}$$

Here, instead of imposing a priori fall-off conditions we have chosen to simply follow the structure imposed by the solution space. In the presence of the logarithmic branches, the study (deferred to future work) of the symplectic potential, the charges, and of the possible interplay with the memory effects and the logarithmic soft theorems [151–153], will give possible physical criteria for interpreting the relaxed non-smooth peeling.

### 2.4.2 Evolution equations for $\Lambda = 0$

The evolution equations for the above functionals can now be derived from the NP formulation in terms of spin coefficients [169], where they are encoded in the Bianchi identities. They take the form[10]

$$n^\mu \partial_\mu \Psi_0 = \hat{m}^\mu \partial_\mu \Psi_1 - (4\gamma - \mu)\Psi_0 + 2(2\tau + \beta)\Psi_1 - 3\sigma\Psi_2, \tag{2.53a}$$

$$n^\mu \partial_\mu \Psi_1 = \hat{m}^\mu \partial_\mu \Psi_2 - \nu\Psi_0 - 2(\gamma - \mu)\Psi_1 + 3\tau\Psi_2 - 2\sigma\Psi_3, \tag{2.53b}$$

$$n^\mu \partial_\mu \Psi_2 = \hat{m}^\mu \partial_\mu \Psi_3 - 2\nu\Psi_1 + 3\mu\Psi_2 - 2(\beta - \tau)\Psi_3 - \sigma\Psi_4, \tag{2.53c}$$

$$n^\mu \partial_\mu \Psi_3 = \hat{m}^\mu \partial_\mu \Psi_4 - 3\nu\Psi_2 + 2(2\mu + \gamma)\Psi_3 - (4\beta - \tau)\Psi_4. \tag{2.53d}$$

It should be noted that these NP evolution equations are in fact also valid in the case $\Lambda \neq 0$ (see e.g. [82] or the appendix to [169]). As we will see in the next section, considering $\Lambda \neq 0$ only modifies the expansion of the spin coefficients and of these evolution equations.

In order to expand these evolution equations, we first have to rewrite the scalars (2.38) in a true Taylor expansion. We will then check for consistency that these NP evolution equations are consistent with the evolution equations derived above from the tensorial Einstein equations, and then write the mass loss formula in (A)dS.

---

[10]Due to an opposite choice of signature for the metric there is a change of sign between the equations displayed here and e.g. reference [169].

## Expansion of the Weyl scalars

In order to obtain the genuine Taylor expansion of the Weyl scalars, we need to take into account the radial dependency of the dyad. The rescaled frame $m^A = r\hat{m}^A$ admits the expansion

$$m^A = m_0^A + \frac{m_1^A}{r} + \mathcal{O}(r^{-2}), \tag{2.54a}$$

$$m_0^A = \sqrt{\frac{q_{\theta\theta}}{2q}}\left(\frac{\sqrt{q} + iq_{\theta\phi}}{q_{\theta\theta}}\delta_\theta^A - i\delta_\phi^A\right), \tag{2.54b}$$

$$m_1^A = \frac{1}{4}\left(C_B^A(\bar{m}_0^B - 2m_0^B) - (C_{BC}m_0^B m_0^C)m_0^A - \frac{1}{2}C\bar{m}_0^A\right), \tag{2.54c}$$

and similarly for the complex conjugates $\bar{m}_0^A$ and $\bar{m}_1^A$. The leading frames $m_0^A$ and $\bar{m}_0^A$ are null in the leading metric $q_{AB}$, and are related to this metric and to the volume form by $q^{AB} = m_0^{(A}\bar{m}_0^{B)}$ and $\epsilon^{AB} = -im_0^{[A}\bar{m}_0^{B]}$, where the bracket denotes anti-symmetrization. They also satisfy important relations given in appendix C, and which we use repeatedly below in order to establish the dictionary between the NP formalism and the tensorial Einstein equations.

Using the above expansion of the frame we can now rewrite the expansion of the Weyl scalars (2.38) as[11]

$$\Psi_0 = \frac{\Psi_0^4}{r^4} + \frac{1}{r^5}\left(\Psi_0^{l5}\ln r + \Psi_0^5\right) + \mathcal{O}(r^{-6}), \tag{2.55a}$$

$$\Psi_1 = \frac{1}{r^4}\left(\Psi_1^{l4}\ln r + \Psi_1^4\right) + \mathcal{O}(r^{-5}), \tag{2.55b}$$

$$\Psi_2 = \frac{\Psi_2^3}{r^3} + \mathcal{O}(r^{-4}), \tag{2.55c}$$

$$\Psi_3 = \frac{\Psi_3^2}{r^2} + \mathcal{O}(r^{-3}), \tag{2.55d}$$

$$\Psi_4 = \frac{\Psi_4^1}{r} + \mathcal{O}(r^{-2}). \tag{2.55e}$$

In this genuine expansion in $r^{-n}$, the "coefficients" which appear in the "fake" expansion (2.38) are mixed due to the radial dependency of the sphere frames. This does not affect the leading terms in the (polyhomogeneous) expansion, but is evidently important to take in to account when computing the subleading terms. For example, in $\Psi_0$ the term in $r^{-5}$ is modified and we have

$$\Psi_0^4 = \mathfrak{L} := \mathfrak{L}_{AB}m_0^A m_0^B, \qquad \Psi_0^{l5} = \mathcal{E}_l := 3E_{lAB}^{\text{TF}}m_0^A m_0^B, \qquad \Psi_0^5 = \mathcal{E} + 2\mathfrak{L}_{AB}m_0^A m_1^B, \tag{2.56}$$

where $\mathcal{E} := \mathcal{E}_{AB}m_0^A m_0^B$. For $\Psi_1$ the overleading logarithmic term does not propagate to the covariant momentum, and we simply have

$$\Psi_1^{l4} = D^B\mathfrak{L}_{AB}m_0^A, \qquad \Psi_1^4 = \mathcal{P} := \mathcal{P}_A m_0^A, \tag{2.57}$$

---

[11]Here the notation $\Psi_k^n$ denotes the coefficient at order $r^{-n}$ in the expansion of $\Psi_k$, and $\Psi_k^{ln}$ denotes a term in $r^{-n}\ln r$. This differs from the notation which is traditionally used to expand the Weyl scalars [127, 161, 162, 169], where authors use $\Psi_k^0$ to denote the leading term in each expansion. The notation used here will be more handy when studying the subleading components of the Weyl scalars and the Newman–Penrose charges.

while for the other scalars we simply get the leading contributions

$$\Psi_2^3 = \mathcal{M}_{\mathbb{C}} := \mathcal{M} + i\widetilde{\mathcal{M}}, \qquad \Psi_3^2 = \mathcal{J} := \mathcal{J}_A \bar{m}_0^A, \qquad \Psi_4^1 = \mathcal{N} := \mathcal{N}_{AB} \bar{m}_0^A \bar{m}_0^B. \qquad (2.58)$$

We will come back to the study of the subleading components (which in particular for $\Psi_0$ contain the spin-$s$ charges) in future work. It is clear that each subleading components $\Psi_k^n$, when written in terms of tensors and frames, will involve the contraction of overleading coefficients in (2.38) with subleading coefficients in the expansion (2.54a) of the frame (as it already happens for $\Psi_0^5$).

## NP evolution equations

Let us now present the expansion of the NP evolution equations (2.53). Our goal is to verify explicitly that these equations match the evolution equations derived in section 2.2 using the tensorial Einstein equations. Once this check is done, we can then rely on the NP equations to write the evolution of the mass and angular momentum, which we are still missing at this stage. We choose to turn off the boundary sources, i.e. set $\beta_0 = 0 = U_0^A$ at this stage. The study of the NP equations with boundary sources is postponed to future work. We also initially set $\Lambda = 0$ in order to study the effect of the logarithmic terms. On-shell of the constraint $\mathfrak{B}_{AB}^0 = 0$, we then find that $\gamma_0$ in (C.3a) reduces to $4\gamma_0 = -\partial_u \ln \sqrt{q}$.

We now expand (2.53) and collect the relevant equations. First, expanding (2.53a) at order $\mathcal{O}(r^{-4})$ gives

$$\left(\partial_u - 4\gamma_0\right)\mathfrak{L} = 0. \qquad (2.59)$$

The expansion at order $\mathcal{O}(r^{-5})$ gives

$$\left(\partial_u - 6\gamma_0\right)\Psi_0^{l5} = \left(m_0^A \partial_A + 2\beta_1\right)\Psi_1^{l4}, \qquad (2.60a)$$

$$\left(\partial_u - 6\gamma_0\right)\Psi_0^5 = \left(m_0^A \partial_A + 2\beta_1\right)\mathcal{P} - 3\sigma_2 \mathcal{M}_{\mathbb{C}} - 2\gamma_0 \Psi_0^{l5} + \left(\mu_1 + 2V_{+1} - 4\gamma_1\right)\mathfrak{L}, \qquad (2.60b)$$

where the first equation comes from the $\ln r$ term. Note that here $\beta_1$ is the term in the expansion (C.2b) of the spin coefficient $\beta$. Expanding (2.53b) at order $\mathcal{O}(r^{-4})$ gives

$$\left(\partial_u - 6\gamma_0\right)\Psi_1^{l4} = 0, \qquad (2.61a)$$

$$\left(\partial_u - 6\gamma_0\right)\mathcal{P} = m_0^A \partial_A \mathcal{M}_{\mathbb{C}} - 2\sigma_2 \mathcal{J} - 2\gamma_0 \Psi_1^{l4} - \nu_0 \mathfrak{L}, \qquad (2.61b)$$

where the first equation comes from the $\ln r$ term. Expanding (2.53c) at order $\mathcal{O}(r^{-3})$ gives

$$\left(\partial_u - 6\gamma_0\right)\mathcal{M}_{\mathbb{C}} = \left(m_0^A \partial_A - 2\beta_1\right)\mathcal{J} - \sigma_2 \mathcal{N}. \qquad (2.62)$$

Finally, expanding (2.53d) at order $\mathcal{O}(r^{-2})$ gives

$$\left(\partial_u - 6\gamma_0\right)\mathcal{J} = \left(m_0^A \partial_A - 4\beta_1\right)\mathcal{N}. \qquad (2.63)$$

These are the evolution equations for the covariant functionals and the logarithmic terms. We can now check the consistency of these evolution equations with the ones derived directly from the tensorial Einstein equations in section 2.2.

## Consistency with the tensorial Einstein equations

Let us first note that in the absence of boundary sources and when $\Lambda = 0$ the on-shell constraint $\mathfrak{B}^0_{AB} = 0$ plugged in (C.4) implies that $2\partial_u m^A_0 = -\partial_u \ln \sqrt{q} \, m^A_0 = 4\gamma_0 m^A_0$. This implies immediately that (2.59) is equivalent to the evolution equation $(\partial_u \mathfrak{L}_{AB}) m^A_0 m^B_0 = 0$, consistently with (2.17c).

Similarly, these relation together with (C.6d) can be used to rewrite (2.60a) in the form of the projected equation

$$\left( 2\partial_u E^{\mathrm{TF}}_{lAB} + E^{\mathrm{TF}}_{lAB} \partial_u \ln \sqrt{q} - \frac{2}{3} D_A D^C \mathfrak{L}_{BC} \right) m^A_0 m^B_0 = 0, \tag{2.64}$$

which is equivalent to (2.20) for $\Lambda = 0$ since the trace-free part of the last two terms in the bracket do not contribute by virtue of the nullness of $m^A_0$ in $q_{AB}$.

Then one can check the equivalence between (2.60b) and (2.21). We can obtain the former by contracting the latter with $m^A_0 m^B_0$, and using the fact that in the absence of boundary sources and for $\Lambda = 0$ we have

$$2(\partial_u - 6\gamma_0)\mathfrak{L}_{AB} m^A_0 m^B_1 \approx \left( \gamma_0 C - \frac{1}{2}\partial_u C - 4\gamma_1 \right) \mathfrak{L} \tag{2.65}$$

on-shell of $\partial_u \mathfrak{L}_{AB} = 0$, as well as the identity (C.6d). Equivalently, we can also use this on-shell relation (2.65) to rewrite (2.60b) as

$$\left( \partial_u - 6\gamma_0 \right)\mathcal{E} = \left( m^A_0 \partial_A + 2\beta_1 \right)\mathcal{P} - 3\sigma_2 \mathcal{M}_{\mathbb{C}} - 2\gamma_0 \mathcal{E}_l - \left( \frac{3}{4}R[q] + \frac{1}{2}(\partial_u - 2\gamma_0)C \right) \mathfrak{L}, \tag{2.66}$$

which is the contraction of (2.21) with $m^A_0 m^B_0$. We see that although (2.60b) initially involved $\Psi^5_0$, and therefore an object contracted with both $m^A_0$ and $m^A_1$, it is possible to use other evolution equations to obtain the rewriting (2.60b) in terms of projections solely on the celestial sphere. We expect that the same mechanism will work for the subleading evolution equations, and that they can therefore all be rewritten as projections onto the leading frames.

One can then check that (2.61a) is equivalent to $\left( \partial_u U^{l3}_A + U^{l3}_A \partial_u \ln \sqrt{q} \right) m^A_0 = 0$, which is the evolution equation obtained from the logarithmic term in (2.25b).

Next, one can show that (2.63) is tautological and implied by the definition of the energy current $\mathcal{J}$ and the radiation $\mathcal{N}$, consistently with the fact that we have not encountered this evolution equation in the tensorial Einstein equations. Indeed, using (C.6e) one can show that (2.63) is the contraction with $\bar{m}^A_0$ of the tensorial equation $\left( \partial_u - 4\gamma_0 \right)\mathcal{J}_A = D^B \mathcal{N}_{AB}$, which in turn follows from the above definitions of the energy current and the radiation tensor.

We are now left with two evolution equations which we have not yet derived from the tensorial Einstein equation (although we have identified where they appear). This is (2.61b) for the angular momentum and (2.62) for the mass. To obtain the tensorial form of (2.61b) we use (C.6f) and (C.6g), and also the fact that without boundary sources and in the flat case $\nu_0 = 2\bar{m}^A_0 \partial_A \gamma_0$. With this one can show that the NP equation is the contraction with $m^A_0$ of the evolution equation

$$\left( \partial_u - 4\gamma_0 \right)\mathcal{P}_A = \partial_A \mathcal{M} + \epsilon_{AB} \partial^B \widetilde{\mathcal{M}} + C^{\mathrm{TF}}_{AB} \mathcal{J}^B - 2D^B \left( \gamma_0 \mathfrak{L}_{AB} \right). \tag{2.67}$$

This shows that the contribution from the logarithmic branch does not affect the evolution of angular momentum when integrated on the celestial sphere.

Finally, we can write the tensorial form of the evolution equation (2.62) for the covariant mass. Using the identities (C.6h) and (C.6i) we get

$$\left(\partial_u - 6\gamma_0\right)\mathcal{M} = \frac{1}{2}D_A\mathcal{J}^A + \frac{1}{4}C_{AB}\mathcal{N}^{AB}. \tag{2.68}$$

This is the flat Bondi mass loss. This concludes the study of the evolution equations in the flat case.

## Removing the logarithmic terms

For comparison with previous results in the literature let us consider the case without logarithmic terms. The above evolution equations in NP form then read

$$\left(\partial_u - 6\gamma_0\right)\mathcal{J} = \left(m_0^A\partial_A - 4\beta_1\right)\mathcal{N}, \tag{2.69a}$$

$$\left(\partial_u - 6\gamma_0\right)\mathcal{M}_{\mathbb{C}} = \left(m_0^A\partial_A - 2\beta_1\right)\mathcal{J} - \sigma_2\mathcal{N}, \tag{2.69b}$$

$$\left(\partial_u - 6\gamma_0\right)\mathcal{P} = m_0^A\partial_A\mathcal{M}_{\mathbb{C}} - 2\sigma_2\mathcal{J}, \tag{2.69c}$$

$$\left(\partial_u - 6\gamma_0\right)\mathcal{E} = \left(m_0^A\partial_A + 2\beta_1\right)\mathcal{P} - 3\sigma_2\mathcal{M}_{\mathbb{C}}, \tag{2.69d}$$

and are organized in a hierarchy with the functionals successively sourcing each other.

At this point it becomes convenient to assign "quantum numbers" to the various functionals, which can be used as a book-keeping device for the structure of the various equations (such as the evolution equations and the transformation laws). These are the conformal (or scaling) dimension $w$, the spin $j$, and the Newman–Penrose spin (or helicity) $s$. The conformal dimension assigned to the line element is $w(\mathrm{d}s^2) = 0$, while for the coordinates (and their differential one-forms) $w(u) = -w(r) = w(x^A) = -1$ and for the vector fields $w(\partial_u) = -w(\partial_r) = w(\partial_A) = +1$. The terms in the expansion of $\gamma_{AB}$ and $U^A$ start respectively with $w(q_{AB}) = 0$ and $w(U_0^A) = 0$, and then have increasing conformal weights. Then, we assign spin $j > 0$ to objects with $j$ lower angular indices as $j(\gamma_{AB}) = 2$, and spin $-j < 0$ to objects with $j$ upper indices as $j(U^A) = -1$. Finally, the frames are assigned an helicity (or spin-weight) $s(m_0^A) = +1$ and $s(\bar{m}_0^A) = -1$, while they both have $w(m_0^A) = w(\bar{m}_0^A) = 0$ and $j(m_0^A) = j(\bar{m}_0^A) = -1$. Full contraction with the frames therefore returns a scalar and trades spin $j$ for helicity $s$. For example $T_{AB}$ with weight $w$ and spin $j = 2$ gets mapped to the scalar (i.e. spin $j = 0$) $\mathcal{T} := T_{AB}m_0^Am_0^B$ with weight $w$ and helicity $s = +2$.

The NP evolution equations can then be rewritten more compactly if we label the various functionals by their helicity as $\mathcal{Q}_s = \Psi_{2-s}^{3+s} = \{\mathcal{N}, \mathcal{J}, \mathcal{M}_{\mathbb{C}}, \mathcal{P}, \mathcal{E}\}$ and introduce the spin-weighted derivatives

$$\eth_u = \partial_u + 2(j - w)\gamma_0, \qquad \eth = \left(m_0^A\partial_A + 2s\beta_1\right), \qquad \bar{\eth} = \left(\bar{m}_0^A\partial_A - 2s\bar{\beta}_1\right). \tag{2.70}$$

The equations (2.69) then take the simple form

$$\eth_u\mathcal{Q}_s = \eth\mathcal{Q}_{s-1} - (s+1)\sigma_2\mathcal{Q}_{s-2}, \qquad \text{for } -1 \leq s \leq 2, \tag{2.71}$$

where the various weights and helicities are given by[12]

| | $\mathcal{N}$ | $\mathcal{J}$ | $\mathcal{M}_\mathbb{C}$ | $\mathcal{P}$ | $\mathcal{E}$ | $\mathcal{E}_l$ | $\mathfrak{L}$ | $\bar{\eth}_u$ | $\bar{\eth}$ | $\nu_0$ | $\gamma_0$ | $\mu_1$ | $\beta_1$ | $\sigma_2$ | $m_0^A$ | $\bar{m}_0^A$ |
|---|---|---|---|---|---|---|---|---|---|---|---|---|---|---|---|---|
| $s$ | $-2$ | $-1$ | $0$ | $1$ | $2$ | $2$ | $2$ | $0$ | $1$ | $-1$ | $0$ | $0$ | $1$ | $2$ | $1$ | $-1$ |
| $w$ | $3$ | $3$ | $3$ | $3$ | $3$ | $3$ | $2$ | $1$ | $1$ | $2$ | $1$ | $2$ | $1$ | $1$ | $0$ | $0$ |

We recover the equations of motion written in [161], with the functionals $\mathcal{Q}_s$ now generalized to the partial Bondi gauge. In the presence of the logarithmic terms there is no such compact rewriting of the evolution equations (although of course each evolution equation can be written separately). Even in the absence of logarithmic terms, the subleading evolution equations become quite intricate [162].

One should note that the subleading component $m_1^A$ of the frame, given in (2.54c), does not posses a definite helicity. Because of this, the subleading evolution equations (or the leading equations in the presence of logarithmic terms) will involve terms which also have no definite helicity. For example $\Psi_0^5$ has no definite helicity since it involves the term in (2.91). This is also the case of the spin coefficient $\gamma_1$ appearing in the evolution equation (2.60b), as can be seen in (C.3b). The terms in the radial expansion of the spin coefficients may also have no well-defined helicity (see footnote 12). We will encounter below examples of quantities which can be rewritten on the celestial sphere (i.e. on a basis of leading frames $m_0^A$) but which have no well-defined helicity. This is the case e.g. for (2.91). This mixing of helicities is a genuine feature which cannot be bypassed. We also note that this mixing was observed in celestial OPEs [44, 54].

### 2.4.3 Evolution equations for $\Lambda \neq 0$

We now briefly study the evolution equations in the case of (A)dS, focusing in particular on the evolution of the mass. When $\Lambda \neq 0$, the NP evolution equations are still given by (2.53). Now however, because the vector $n$ and some of the spin coefficients contain overleading terms with respect to the flat case, some coefficients and Weyl scalars have to be expanded further. Using the same notation as above (and again setting the boundary sources to zero) we then obtain evolution equations which can be repackaged in terms of the spin-$s$ functionals as

$$\left( \bar{\eth}_u - \frac{\Lambda}{8}(s+1)C\mathcal{Q}_s \right) \mathcal{Q}_s = \bar{\eth}\mathcal{Q}_{s-1} - (s+1)\sigma_2 \mathcal{Q}_{s-2} + \frac{\Lambda}{6}\Psi_{2-s}^{4+s}, \tag{2.72}$$

for $-1 \leq s \leq 2$. We see that $\Lambda \neq 0$ brings in the subleading terms $\Psi_{2-s}^{4+s}$ in the expansion of the Weyl scalars.

Let us now focus on $s = 0$ in order to obtain the evolution of the mass in (A)dS. In order to compute $\Psi_2^4$, we need to also extend the solution space to subleading order. This subleading solution

---

[12]Note that the spin coefficients $\nu_0$ and $\gamma_0$ only have a well-defined helicity in the case $\Lambda = 0$ and $\Lambda = 0 = U_0^A$ respectively. The helicity of $\mu_1$, $\beta_1$ and $\sigma_2$ is however always well-defined.

space is given in appendix F. With this we can continue with the expansion (2.55c) to find

$$\Psi_2 = \frac{\Psi_2^3}{r^3} + \frac{1}{r^4}\left(\Psi_2^4 + \Psi_2^{l4}\ln r\right) + \mathcal{O}(r^{-5}), \qquad \Psi_2^3 = \mathcal{M}_\mathbb{C}. \tag{2.73}$$

The general off-shell expressions for $\mathrm{Re}\big(\Psi_2^4\big)$ and $\mathrm{Re}\big(\Psi_2^{l4}\big)$ in the case of an arbitrary cosmological constant are reported in appendix F. For $\Lambda \neq 0$, all the logarithmic terms disappear on-shell of the constraints, one has $\mathrm{Re}\big(\Psi_2^{l4}\big) = 0$, and we are left with

$$\mathrm{Re}\big(\Psi_2^4\big) = -\frac{1}{2}D_A\mathcal{P}^A - \frac{3}{4}C\mathcal{M} + \frac{\Lambda}{24}C_{AB}\mathcal{E}^{AB}. \tag{2.74}$$

The NP equation for the evolution of the mass is then

$$\left(\partial_u - 6\gamma_0\right)\mathcal{M}_\mathbb{C} = \left(m_0^A\partial_A - 2\beta_1\right)\mathcal{J} - \sigma_2\mathcal{N} + \frac{\Lambda}{8}C\mathcal{M}_\mathbb{C} + \frac{\Lambda}{6}\Psi_2^4. \tag{2.75}$$

Using the fact that (C.3a) gives $4\mathrm{Re}(\gamma_0) = V_{+2}$, the real part of the above equation finally leads to the (A)dS mass loss formula

$$\boxed{\left(\partial_u + \frac{3}{2}\partial_u\ln\sqrt{q} - \frac{\Lambda}{4}C\right)\mathcal{M} = \frac{1}{2}D_A\left(\mathcal{J}^A - \frac{\Lambda}{6}\mathcal{P}^A\right) + \frac{1}{4}C_{AB}\mathcal{S}^{AB}.} \tag{2.76}$$

This is in agreement with (2.52) of [67]. One can indeed check with some rewriting that in BS gauge and in the absence of logarithmic terms we have $\mathcal{J}^A - \Lambda\mathcal{P}^A/6 = -\Lambda N_{(\Lambda)}^A/3$ and $\mathcal{S}^{AB} = -\Lambda^2 J^{AB}/6$, where $N_{(\Lambda)}^A$ and $J^{AB}$ are the quantities defined respectively in equations (2.40) and (2.41) of [67]. We note that when $\Lambda = 0$ the functional $\mathcal{S}^{AB}$ reduces to $\mathcal{N}^{AB}$, so that the above evolution equation reduces to its flat limit (2.68).

## 2.5   Intermediate summary

Starting from the partial Bondi gauge including logarithmic terms in (2.3), we have solved the Einstein equations in a radial expansion for an arbitrary value of $\Lambda$. We summarize in table 2 below the structure of the solution space and the role of the various constraints and evolution equations. The data which is *a priori* free is given by the mass $M$, the angular momentum $U_3^A$, the boundary sources $\beta_0$ and $U_0^A$, and the (smooth and logarithmic) components of the angular metric. We have seen and we recall below how some of this data can be repackaged into covariant functionals, which are the quantities appearing in the NP formalism. Some of this data is also subject to evolution equations or constraints, such that at the end of the day not all the data is free. This turns out to depend on whether the cosmological constant is vanishing or not. We also recall in the table below which assumption (if any) on the boundary sources has been used at which stage. This indicates what has to be relaxed in future work.

In the case $\Lambda \neq 0$ the Einstein equations enforce the vanishing of the trace-free parts of the logarithmic terms. We gather in the table the location of the various equations which remove these logarithmic terms at the first few orders. When further completing the partial Bondi gauge to reach

the BS or NU gauge, there are then constraints on the traces of the terms in the expansion of $\gamma_{AB}$. In particular, it is important to notice that the traces of the logarithmic terms are always vanishing in BS or NU gauge on-shell of the $\Lambda \neq 0$ Einstein equations. This vanishing is imposed by the determinant condition in BS gauge, and by the vanishing of $\beta$ at all orders in NU gauge. This happens at all order and with all the logarithmic terms, and the final result is that there are no logarithmic terms in the (A)dS solution space.

Moreover, when $\Lambda \neq 0$ the shear $C_{AB}^{\mathrm{TF}}$ is completely determined in terms of the boundary metric (more precisely $U_0^A$ and the time derivative of $q_{AB}$) through the constraint $\mathfrak{B}_{AB}^{\Lambda} = 0$. This is drastically different from the flat case, in which $C_{AB}^{\mathrm{TF}}$ is free and independent from $q_{AB}$. As explained in [67], the $(AB)$ Einstein equations give at every order an equation of the type $\Lambda \gamma_{AB}^n = \partial_u \gamma_{AB}^{n-1} + (\dots)$ for $n > 1$. This therefore does *not* constrain $E_{AB}$ in terms of the time evolution of $D_{AB}$ since $n > 1$. For example equation (I.1) shows that $F_{AB}^{\mathrm{TF}}$ is indeed determined by $\partial_u \mathcal{E}_{AB}$ (which contains $\partial_u E_{AB}^{\mathrm{TF}}$). This means that the solution space is entirely determined by 11 functions of $(u, x^A)$. These are the 8 unconstrained functions $(\beta_0, U_0^A, q_{AB}, E_{AB}^{\mathrm{TF}})$, and the 3 functions $(M, U_3^A)$ subject to evolution equations. By contrast, in the flat case the equation (I.1) has to be interpreted as an evolution equation constraining $E_{AB}^{\mathrm{TF}}$. It is therefore interesting to note that, in (A)dS where $E_{AB}^{\mathrm{TF}}$ is unconstrained, this field sources the mass loss (2.76) together with the radiation tensor. $E_{AB}^{\mathrm{TF}}$ is therefore part of the radiative data in (A)dS, even when $\partial_u q_{AB} = 0$.

When $\Lambda = 0$, nothing prevents the logarithmic terms from being included in the solution space, and they are furthermore subject to evolution equations which we also recall in the table. These evolution equations depend only on the boundary metric and are not sourced by the shear. Among the data there are, first, the 6 functions $(\beta_0, U_0^A, q_{AB})$ with the time evolution of $q_{AB}$ constrained by $\mathfrak{B}_{AB}^0 = 0$. There is then an infinite tower of functions appearing in the radial expansion of the angular metric (namely $E_{AB}^{\mathrm{TF}}$, $F_{AB}^{\mathrm{TF}}$, ... ) and admitting a non-trivial time evolution as in (I.1) [28]. The logarithmic terms have a non-trivial time evolution as well, but this latter depends only on the boundary metric and sources, as for example in (2.17c) and (2.20). When the boundary metric has no time dependency and the boundary sources are turned off, the evolution of the logarithmic terms in $\gamma_{AB}$ can be integrated. For example $\mathfrak{L}_{AB}$ is a constant of the motion by virtue of (2.17c), and therefore $E_{lAB}$ evolves linearly in $u$ according to (2.20). This is in accordance with equation (12) of [138], which is a particular case of our evolution equations.

In summary, the solution space contains an infinite tower of constrained functionals $\mathcal{Q}_{s \geq 2}$ in the flat case, together with possible logarithmic terms if one chooses to include them in the solution space. We will come back to the study of these higher spin functionals in future work as here we have stopped at $s = 2$. In (A)dS, on the other hand, the quantities $\mathcal{Q}_{s+1}$ are fixed algebraically by the time evolution of $\mathcal{Q}_s$, leaving only $\mathcal{Q}_2 \sim \mathcal{E}$ free [67]. This is the reason for which, in the last three lines of table 2 below, we have written that (2.72) gives the evolution equations for $\mathcal{Q}_s$ with $s > 2$ only. Indeed, for $s = 2$ the equation for $\mathcal{Q}_2$ is (I.1), which is *not* an equation constraining $E_{AB}^{\mathrm{TF}}$ but rather an equation determining $F_{AB}^{\mathrm{TF}}$ in terms of $\partial_u E_{AB}^{\mathrm{TF}}$.

| condition on $\Lambda$ | quantity or equation | notation and origin | simplification |
|:---:|:---:|:---:|:---:|
| – | mass and momentum | $M$ and $U_3^A$ | – |
| – | boundary sources | $\beta_0$ and $U_0^A$ | – |
| – | smooth terms in $\gamma_{AB}$ | $q_{AB}, C_{AB}, D_{AB}, E_{AB}, \dots$ | – |
| – | logarithmic terms in $\gamma_{AB}$ | $E_{lAB}, F_{lAB}, F_{l^2AB}, \dots$ | – |
| – | covariant spin-2 | $\mathcal{E}_{AB}$ (2.39) | – |
| – | covariant momentum | $\mathcal{P}_A$ (2.40) | – |
| – | covariant mass | $\mathcal{M}$ (2.41) | – |
| – | energy current | $\mathcal{J}_A$ (2.43) | $U_0^A = 0$ |
| – | radiation | $\mathcal{N}_{AB}$ (2.44) | $\beta_0 = 0 = U_0^A$ |
| – | $C = 0$ | BS gauge (3.2a) | – |
| – | $E_l = 0$ | BS gauge (3.2c) or NU gauge (2.7c) | – |
| – | $F_{l^2} = 0$ | BS gauge (3.2d) or NU gauge (F.1a) | – |
| $\Lambda \neq 0$ | $F_l = 0$ | BS gauge (3.2d) + $\left.\mathrm{E}^{\mathrm{TF}}_{AB}\right|_{\mathcal{O}(r^{-1})}$ | – |
| $\Lambda \neq 0$ | $F_l = 0$ | NU gauge (F.1b) + $\left.\mathrm{E}^{\mathrm{TF}}_{AB}\right|_{\mathcal{O}(r^{-1})}$ | – |
| $\Lambda \neq 0$ | $\mathfrak{L}_{AB} = 0 = U_{l3}^A$ | $\left.\mathrm{E}^{\mathrm{TF}}_{AB}\right|_{\mathcal{O}(r^0)} \Leftrightarrow$ (2.17b) | – |
| | $E^{\mathrm{TF}}_{lAB} = 0$ | $\left.\mathrm{E}^{\mathrm{TF}}_{AB}\right|_{\mathcal{O}(r^{-1})} \Leftrightarrow$ (2.17c) | – |
| | $F^{\mathrm{TF}}_{lAB} = 0$ | $\left.\mathrm{E}^{\mathrm{TF}}_{AB}\right|_{\mathcal{O}(r^{-2})} \Leftrightarrow$ (2.20) | – |
| | $F^{\mathrm{TF}}_{l^2AB} = 0$ | $\left.\mathrm{E}^{\mathrm{TF}}_{AB}\right|_{\mathcal{O}(r^{-2})} \Leftrightarrow$ (2.19) | – |
| – | evolution of $q_{AB}$ | $\left.\mathrm{E}^{\mathrm{TF}}_{AB}\right|_{\mathcal{O}(r)} : \mathfrak{B}^{\Lambda}_{AB} = 0$ in (2.18) | – |
| $\Lambda = 0$ | evolution of $\mathcal{E}_{AB}$ | $\left.\mathrm{E}^{\mathrm{TF}}_{AB}\right|_{\mathcal{O}(r^{-2})} \Leftrightarrow$ (2.21) $\Leftrightarrow$ (2.60b) | $\beta_0 = 0 = U_0^A$ |
| | evolution of $\mathcal{P}_A$ | $\left.\mathrm{E}_{uA}\right|_{\mathcal{O}(r^{-2})} \Leftrightarrow$ (2.67) $\Leftrightarrow$ (2.61b) | $\beta_0 = 0 = U_0^A$ |
| | evolution of $\mathcal{M}$ | $\left.\mathrm{E}_{uu}\right|_{\mathcal{O}(r^{-2})} \Leftrightarrow$ (2.68) $\Leftrightarrow$ (2.62) | $\beta_0 = 0 = U_0^A$ |
| | evolution of $\mathfrak{L}_{AB}$ | $\left.\mathrm{E}^{\mathrm{TF}}_{AB}\right|_{\mathcal{O}(r^{-1})} \Leftrightarrow$ (2.17c) $\Rightarrow$ (2.59) | – |
| | evolution of $E^{\mathrm{TF}}_{lAB}$ | $\left.\mathrm{E}^{\mathrm{TF}}_{AB}\right|_{\mathcal{O}(r^{-2})} \Leftrightarrow$ (2.20) $\Leftrightarrow$ (2.60a) | $\beta_0 = 0 = U_0^A$ |
| | evolution of $U_A^{l3}$ | $\left.\mathrm{E}^{\mathrm{TF}}_{uA}\right|_{\mathcal{O}(r^{-2})} \Leftrightarrow$ (2.25b) $\Leftrightarrow$ (2.61a) | $\beta_0 = 0 = U_0^A$ |
| $\Lambda \neq 0$ | evolution of $\mathcal{Q}_{s<2}$ | (2.72) | $\beta_0 = 0 = U_0^A$ |
| | evolution of $\mathcal{M}$ | $\left.\mathrm{E}_{uu}\right|_{\mathcal{O}(r^{-2})} \Leftrightarrow$ (2.75) $\Leftrightarrow$ (2.76) | $\beta_0 = 0 = U_0^A$ |
| | determination of $F^{\mathrm{TF}}_{AB}$ | $\left.\mathrm{E}^{\mathrm{TF}}_{AB}\right|_{\mathcal{O}(r^{-2})} \Leftrightarrow$ (I.1) | $\beta_0 = 0 = U_0^A$ |

Table 2: Structure of the solution space in partial Bondi gauge.

## 2.6 Asymptotic symmetries

Although the partial Bondi gauge is an incomplete gauge fixing, it still enables to define residual symmetries and to compute the ensuing transformation laws. This is because the radial expansions of all the free functions in the line element (2.1) have been determined. We can therefore obtain general expressions for the asymptotic symmetries and their action on the solution space, before then reducing these expressions to the BS or NU gauge. For this let us first derive the expression for the vector fields generating the residual symmetries.

### 2.6.1 Vector fields

Preserving the partial Bondi gauge, i.e. imposing $\pounds_\xi g_{rr} = 0 = \pounds_\xi g_{rA} = 0$, imposes that the vector field $\xi = \xi^u \partial_u + \xi^r \partial_r + \xi^A \partial_A$ has temporal and angular components given by

$$\xi^u = f, \qquad \xi^A = Y^A + I^A, \tag{2.77}$$

with

$$\partial_r f = 0 = \partial_r Y^A, \qquad I^A = -\int_r^\infty dr' \, e^{2\beta} \gamma^{AB} \partial_B f. \tag{2.78}$$

From this we can already compute the Lie derivatives

$$\pounds_\xi \gamma_{AB} = \left(f\partial_u + \pounds_Y + \pounds_I\right)\gamma_{AB} + \xi^r \partial_r \gamma_{AB} - \gamma_{(AC}U^C \partial_{B)}f, \tag{2.79a}$$

$$\pounds_\xi \ln\gamma = (f\partial_u + \xi^r \partial_r)\ln\gamma + 2\mathcal{D}_A\xi^A - 2U^A\partial_A f, \tag{2.79b}$$

$$\pounds_\xi g_{ur} = -e^{2\beta}\left(2\xi^\alpha \partial_\alpha\beta + \partial_r\xi^r + U^A\partial_A f + \partial_u f\right). \tag{2.79c}$$

These expressions will be used below in sections 3 and 4 when looking at the conditions which preserve the BS and NU gauges.

The radial part of the vector field is then determined by the requirement that (2.79a) be compatible with the ansatz (2.3). Preserving the form of the angular metric imposes that the radial part has an expansion of the form

$$\xi^r = r\xi^r_{+1} + \xi^r_0 + \frac{\xi^r_1}{r} + \frac{1}{r^2}\left(\xi^r_{l2}\ln r + \xi^r_2\right) + \mathcal{O}(r^{-3}), \tag{2.80}$$

where the various coefficients of the expansion are a priori arbitrary functions of $(u, x^A)$.

In summary, the residual symmetries are generated by $\xi$, parametrized by the free functions of $(u, x^A)$ which are the supertranslations $f$, the superrotations $Y^A$, and the functions appearing in the expansion of $\xi^r$. These vector fields preserve the on-shell fall-off conditions (2.4) without any extra restrictions on the generators. The functions appearing in the expansion of $\xi^r$ are for the moment arbitrary, but will be determined in terms of the field content of the solution space upon reduction of the partial Bondi gauge to the BS or NU gauge. Since the choice of BS or NU gauge amounts to a choice of radial coordinate, it is natural that this is what determines the form of the radial component of the vector field. We show below in section 3 that in BS gauge $\xi^r$ contains a single free function (in addition to $f$ and $Y^A$), which appears at order $\mathcal{O}(r)$ and parametrizes Weyl

transformations. In section 4 we then show that in NU gauge in addition to this Weyl parameter there is a free function also at order $\mathcal{O}(r^0)$, which parametrizes the radial translations. One should then compute the charges in order to see which part of the vector field is actually associated with a true (i.e. non-gauge) asymptotic symmetry. We know that the Weyl transformations are associated to a non-vanishing charge [125], and are therefore physical, while for the radial translations there is so far no computation of the charge. This lengthy task is postponed to future work, and for the time being we focus on the algebra and transformation laws.

### 2.6.2 Transformation laws

In order to obtain the transformation laws of various functionals on phase space, we first expand the angular part of the asymptotic Killing vector as

$$
\begin{aligned}
\xi^A &= Y^A + \frac{I_1^A}{r} + \frac{I_2^A}{r^2} + \frac{I_3^A}{r^3} + \mathcal{O}(r^{-4}) \\
&= Y^A - e^{2\beta_0} \left( \frac{q^{AB}}{r} - \frac{C^{AB}}{2r^2} + \frac{1}{3r^3} \left( 2\beta_2 q^{AB} + \gamma_4^{AB} \right) \right) \partial_B f + \mathcal{O}(r^{-4}),
\end{aligned}
\tag{2.81}
$$

with $\beta_2$ and $\gamma_4^{AB}$ given respectively in (2.7b) and (D.7a). The transformation laws for the boundary sources and volume element are

$$
\delta_\xi \ln \sqrt{q} = 2\xi_{+1}^r + f\partial_u \ln \sqrt{q} - U_0^A \partial_A f + D_A Y^A,
\tag{2.82a}
$$

$$
\delta_\xi \beta_0 = \left( f\partial_u + \pounds_Y \right)\beta_0 + \frac{1}{2} \left( \xi_{+1}^r + \partial_u f + U_0^A \partial_A f \right),
\tag{2.82b}
$$

$$
\delta_\xi U_0^A = \left( f\partial_u + \pounds_Y + \partial_u f + U_0^C \partial_C f \right) U_0^A - \partial_u Y^A - \frac{\Lambda}{3} e^{4\beta_0} \partial^A f,
\tag{2.82c}
$$

where $\pounds_Y X^A = [Y, X]^A = Y^B \partial_B X^A - X^B \partial_B Y^A$. This shows that $\xi_{+1}^r$ is related to the Weyl rescalings of the induced boundary metric. These transformation laws also give the constraints on the symmetry generators which have to be satisfied if the boundary sources are turned off and/or if the boundary volume element is kept fixed. We also note that the transformation of $U_0$ does not depend on $\xi^r$.

Looking at the radial expansion of (2.79a), we can now read the transformation laws of the various subleading components of the angular metric. We find

$$\delta_\xi q_{AB} = \big(f\partial_u + \pounds_Y + 2\xi^r_{+1}\big)q_{AB} - U^0_{(A}\partial_{B)}f, \tag{2.83a}$$

$$\delta_\xi C_{AB} = \big(f\partial_u + \pounds_Y + \xi^r_{+1}\big)C_{AB} + \big(\pounds_{I_1} + 2\xi^r_0\big)q_{AB} - \big(U^1_{(A} + \bar U^0_{(A}\big)\partial_{B)}f$$

$$= \big(f\partial_u + \pounds_Y + \xi^r_{+1}\big)C_{AB} + 2\xi^r_0 q_{AB} - 2e^{2\beta_0}\big(2\partial_{(A}\beta_0\partial_{B)}f + D_A\partial_B f\big) - \bar U^0_{(A}\partial_{B)}f, \tag{2.83b}$$

$$\delta_\xi D_{AB} = \big(f\partial_u + \pounds_Y\big)D_{AB} + \big(\pounds_{I_1} + \xi^r_0\big)C_{AB} + \big(\pounds_{I_2} + 2\xi^r_1\big)q_{AB} - \big(U^2_{(A} + \bar U^1_{(A} + \bar{\bar U}^0_{(A}\big)\partial_{B)}f$$

$$= \big(f\partial_u + \pounds_Y\big)D_{AB} + \xi^r_0 C_{AB} + 2\xi^r_1 q_{AB} - \bar{\bar U}^0_{(A}\partial_{B)}f$$

$$+ \frac{1}{2}e^{2\beta_0}\Big(D^C C_{(AC} - C_{(AC}D^C - \partial_{(A}C - 2C_{(AC}\partial^C\beta_0\Big)\partial_{B)}f$$

$$+ \frac{1}{2}e^{2\beta_0}\Big(D_{(A}C_{B)C} - 2D_C C_{AB} - 2C_{(AC}\partial_{B)}\beta_0\Big)\partial^C f, \tag{2.83c}$$

$$\delta_\xi E_{AB} = \big(f\partial_u + \pounds_Y - \xi^r_{+1}\big)E_{AB} + \xi^r_{+1}E_{lAB} + \pounds_{I_1}D_{AB} + \big(\pounds_{I_2} + \xi^r_1\big)C_{AB} + \big(\pounds_{I_3} + 2\xi^r_2\big)q_{AB}$$

$$- \big(U^3_{(A} + \bar U^2_{(A} + \bar{\bar U}^1_{(A} + E_{(AC}U^C_0\big)\partial_{B)}f, \tag{2.83d}$$

$$\delta_\xi E_{lAB} = \big(f\partial_u + \pounds_Y - \xi^r_{+1}\big)E_{lAB} + 2\xi^r_{l2}q_{AB} - E_{l(AC}U^C_0\partial_{B)}f - U^{l3}_{(A}\partial_{B)}f, \tag{2.83e}$$

where $\pounds_Y C_{AB} = Y^C\partial_C C_{AB} + C_{(AC}\partial_{B)}Y^C$ and we have denoted $\bar U_A = C_{AB}U^B$ and $\bar{\bar U}_A = D_{AB}U^B$. Using $\delta C = q^{AB}\delta C_{AB} - \delta q_{AB}C^{AB}$ we then find the variations of the traces, which are

$$\delta_\xi C = \big(f\partial_u + \pounds_Y - \xi^r_{+1}\big)C + 4\xi^r_0 - 2e^{2\beta_0}\big(4\partial_A\beta_0 + D_A\big)\partial^A f, \tag{2.84a}$$

$$\delta_\xi D = \big(f\partial_u + \pounds_Y - 2\xi^r_{+1}\big)D + \xi^r_0 C + 4\xi^r_1 - e^{2\beta_0}\Big(C_{AB}\big(4\partial^B\beta_0 + D^B\big) + 2\partial_A C - 2D^B C_{AB}\Big)\partial^A f,$$

$$\delta_\xi E_l = \big(f\partial_u + \pounds_Y - 3\xi^r_{+1}\big)E_l + 4\xi^r_{l2} - 2U^A_{l3}\partial_A f. \tag{2.84b}$$

From this we then get that the variations of the trace-free tensors appearing in the solution space are given by[13]

$$\delta_\xi C^{\mathrm{TF}}_{AB} = \big(f\partial_u + \pounds_Y + \xi^r_{+1}\big)C^{\mathrm{TF}}_{AB} - C^{\mathrm{TF}}_{(AC}U^C_0\partial_{B)}f - 2e^{2\beta_0}\big(2\partial_{(A}\beta_0\partial_{B)}f + D_A\partial_B f\big)^{\mathrm{TF}}, \tag{2.85a}$$

$$\delta_\xi E^{\mathrm{TF}}_{lAB} = \big(f\partial_u + \pounds_Y - \xi^r_{+1}\big)E^{\mathrm{TF}}_{lAB} - E^{\mathrm{TF}}_{l(AC}U^C_0\partial_{B)}f - \big(U^{l3}_{(A}\partial_{B)}f\big)^{\mathrm{TF}}, \tag{2.85b}$$

$$\delta_\xi \mathfrak{L}_{AB} = \big(f\partial_u + \pounds_Y\big)\mathfrak{L}_{AB} - \mathfrak{L}_{(AC}U^C_0\partial_{B)}f, \tag{2.85c}$$

$$\delta_\xi \mathfrak{B}^0_{AB} = \big(f\partial_u + \partial_u f + \pounds_Y + 2\xi^r_{+1}\big)\mathfrak{B}^0_{AB} - \mathfrak{B}^0_{(AC}U^C_0\partial_{B)}f + \mathfrak{B}^0_{AB}U^C_0\partial_C f + \frac{\Lambda}{3}\Big(D_{(A}\big(e^{4\beta_0}\partial_{B)}f\big)\Big)^{\mathrm{TF}},$$

$$\delta_\xi \mathfrak{B}^\Lambda_{AB} = \big(f\partial_u + \partial_u f + \pounds_Y + 2\xi^r_{+1}\big)\mathfrak{B}^\Lambda_{AB} - \mathfrak{B}^\Lambda_{(AC}U^C_0\partial_{B)}f + \mathfrak{B}^\Lambda_{AB}U^C_0\partial_C f. \tag{2.85d}$$

The last three variations are used as a consistency check. They show that imposing the constraint $\mathfrak{B}^\Lambda_{AB} = 0$ and/or removing the logarithmic terms does not give any restrictions on the symmetry parameters. The variation (2.85b) shows that the logarithmic term $U^{l3}_A$ gives rise to a logarithmic term $E_{lAB}$ in the angular metric.

---

[13]One should note that $\delta_\xi C^{\mathrm{TF}}_{AB} = \delta_\xi\big(C^{\mathrm{TF}}_{AB}\big) \neq \big(\delta_\xi C_{AB}\big)^{\mathrm{TF}}$, and similarly for the other tensors, so these are the variations of trace-free tensors and *not* the trace-free variations.

We now give for completeness the variation of the "covariant" spin-2 functional, which can be computed from the definition (2.39) and the above transformation laws. We find

$$\delta_\xi \mathcal{E}_{AB} = \left(f\partial_u + \pounds_Y - \xi_{+1}^r\right)\mathcal{E}_{AB} + 3\xi_{+1}^r E_{lAB}^{\mathrm{TF}} - 2\xi_0^r \mathfrak{L}_{AB} - \mathcal{E}_{(AC}U_0^C\partial_{B)}f + 2e^{2\beta_0}\left(\mathcal{P}_{(A}\partial_{B)}f\right)^{\mathrm{TF}}$$
$$- e^{2\beta_0}\left(\left(\mathfrak{L}_{(AC}D_{B)}\partial^C f\right)^{\mathrm{TF}} - \left(\partial^C f D_{(A}\mathfrak{L}_{B)C}\right)^{\mathrm{TF}} + \partial^C f\left(4\partial_C\beta_0 + D_C\right)\mathfrak{L}_{AB}\right). \tag{2.86}$$

As announced above, we see that this functional does not transform homogeneously even for the subset of symmetries corresponding to $f = 0$ (which is $\mathrm{Diff}(S^2) \ltimes \mathrm{Weyl}$ in the BMSW gauge studied in [124, 125]). Furthermore, we see that since $\mathcal{E}_{AB}$ and $E_{lAB}$ have the same weight for $\xi_{+1}^r$ in their transformation laws, it is not possible to find a combination which is homogeneous for $f = 0$. Similarly, for the momentum we find

$$\delta_\xi \mathcal{P}_A = \left(f\partial_u + \pounds_Y - 2\xi_{+1}^r\right)\mathcal{P}_A - U_0^B \mathcal{P}_B \partial_A f + D^B\left(\xi_{+1}^r \mathfrak{L}_{BA}\right) + 3e^{2\beta_0}\left(\mathcal{M}q_{AB} - \widetilde{\mathcal{M}}\epsilon_{AB}\right)\partial^B f$$
$$+ (\text{terms in } \partial_A f), \tag{2.87}$$

so the logarithmic term coming with $\xi_{+1}^r$ gives an anomaly. It is interesting to note that these anomalous terms in the transformation of $\mathcal{E}_{AB}$ and $\mathcal{P}_A$ are related to the overleading logarithmic terms appearing at the same $1/r^n$ order in the expansion (2.38). This suggests that all the leading terms in the expansion (2.38), or equivalently (2.55), do not suffer from these anomalies and are actually covariant when $f = 0$. This is confirmed by the transformation (2.85c) of $\mathfrak{L}_{AB}$, and by the transformation of the leading term in (2.38b), which is

$$\delta_\xi\left(D^B\mathfrak{L}_{AB}\right) = \left(f\partial_u + \pounds_Y - 2\xi_{+1}^r\right)\left(D^B\mathfrak{L}_{AB}\right) + \partial^B f\left(\partial_u + \partial_u \ln\sqrt{q} + U_0^C D_C + 2D_C U_0^C\right)\mathfrak{L}_{AB}$$
$$- \frac{1}{2}\partial_A f\left(\mathfrak{L}^{BC}\partial_u q_{BC} + 2U_0^B D^C \mathfrak{L}_{BC}\right) - \partial^C f\left(2D^B U_C^0 \mathfrak{L}_{AB} + \mathfrak{L}_A^B\partial_u q_{BC}\right). \tag{2.88}$$

We do not compute here the lengthy transformations of $\mathcal{M}$, $\mathcal{J}_A$ and $\mathcal{N}_{AB}$, but conjecture based on this observation that these functionals transform homogeneously when $f = 0$, and therefore truly deserve the name "covariant" (while, we recall, $\mathcal{E}_{AB}$ and $\mathcal{P}_A$ are only covariant in the absence of logarithmic terms). One should note that $\mathcal{E}_{AB}$ and $\mathcal{P}_A$ are however covariant when $\Lambda \neq 0$, since in this case all the logarithmic terms vanish.

It is also interesting to study the transformations of the NP scalars projected on the leading frames. For this we first compute the variation of the leading frame itself. Decomposing $Y$ on the null frames as $Y^A = \mathcal{Y}\bar{m}_0^A + \bar{\mathcal{Y}}m_0^A$ with $\mathcal{Y} = Y_A m_0^A$ and $\bar{\mathcal{Y}} = Y_A \bar{m}_0^B$, we find

$$\delta_\xi m_0^A = \left(f\partial_u - \xi_{+1}^r\right)m_0^A + \frac{1}{2}m_B^0\left(\partial^{(A}fU_0^{B)} - D^{(A}Y^{B)}\right) + \frac{1}{2}m_0^A\left(m_0^C m_0^B - \bar{m}_0^C\bar{m}_0^B\right)\left(\partial_B f U_C^0 - D_C Y_B\right)$$
$$= \left(f\partial_u - \xi_{+1}^r\right)m_0^A + \frac{1}{2}m_0^A\left(U_0(\eth + \bar{\eth})f + \bar{U}_0(\eth - \bar{\eth})f - (\eth + \bar{\eth})\mathcal{Y} - (\eth - \bar{\eth})\bar{\mathcal{Y}}\right)$$
$$- \bar{m}_0^A\left(\eth\mathcal{Y} - U_0\eth f\right), \tag{2.89}$$

with $U_0 = U_0^A m_A^0$ and $\bar{U}_0 = U_0^A \bar{m}_A^0$, and where we have used $m_0^A \bar{\eth}f + \bar{m}_0^A\eth f = \partial^A f$. Note that $f$

is of helicity 0. With this we then find

$$\delta_\xi \mathfrak{L} = \left( f\partial_u - 2\xi^r_{+1} + \mathcal{Y}\bar{\eth} + \bar{\mathcal{Y}}\eth + (\eth + \bar{\eth})(\bar{\mathcal{Y}} - \mathcal{Y}) + (U_0 - \bar{U}_0)(\eth + \bar{\eth})f \right)\mathfrak{L}, \tag{2.90a}$$

$$\delta_\xi \mathcal{E}_l = \left( f\partial_u - 3\xi^r_{+1} + \mathcal{Y}\bar{\eth} + \bar{\mathcal{Y}}\eth + (\eth + \bar{\eth})(\bar{\mathcal{Y}} - \mathcal{Y}) + (U_0 - \bar{U}_0)(\eth + \bar{\eth})f \right)\mathcal{E}_l - 6m_0^A U_A^{l3}\eth f, \tag{2.90b}$$

$$\delta_\xi \mathcal{E} = \left( f\partial_u - 3\xi^r_{+1} + \mathcal{Y}\bar{\eth} + \bar{\mathcal{Y}}\eth + (\eth + \bar{\eth})(\bar{\mathcal{Y}} - \mathcal{Y}) + (U_0 - \bar{U}_0)(\eth + \bar{\eth})f \right)\mathcal{E} + \xi^r_{+1}\mathcal{E}_l - 2\xi^r_0 \mathfrak{L}$$

$$+ e^{2\beta_0}\left( 4\mathcal{P}\bar{\eth}f - 4\partial_A\beta_0\partial^A f\mathfrak{L} - 2(\bar{\eth}\eth f)\mathfrak{L} - \eth f\bar{\eth}\mathfrak{L} - \bar{\eth}f\eth\mathfrak{L} \right), \tag{2.90c}$$

which shows once again that $\mathfrak{L}$ and $\mathcal{E}_l$ transform homogeneously when $f = 0$ while $\mathcal{E}$ has an anomaly due to the logarithmic terms.

As explained at the end of section 2.4.2, subleading term in the expansion (2.55) of the Weyl tensor, such as $\Psi_0^5 = \mathcal{E} + \mathfrak{L}_{AB}m_0^A m_1^B$, do not have a well-defined helicity. This quantity can however still be written in terms of a projection on the celestial basis $m_0^A$. Explicitly, this comes from the rewriting

$$\mathfrak{L}_1 := \mathfrak{L}_{AB}m_0^A m_1^B = \frac{1}{4}\left( 2(\sigma_2 - \bar{\sigma}_2) - C \right)\mathfrak{L}, \tag{2.91}$$

where the right-hand side is indeed a projection onto $m_0^A m_0^B$, although it is the sum of terms of respective helicity $+4$, 0, and $+2$. The transformation law for this object is

$$\delta\mathfrak{L}_1 = \left( f\partial_u - 3\xi^r_{+1} + \mathcal{Y}\bar{m}_0^A\partial_A + \bar{\mathcal{Y}}m_0^A\partial_A \right)\mathfrak{L}_1 - \xi^r_0 \mathfrak{L}$$

$$+ \left( (\eth + \bar{\eth})(\bar{\mathcal{Y}} - \mathcal{Y}) + 4(\bar{\mathcal{Y}}\beta_1 - \mathcal{Y}\bar{\beta}_1) + (U_0 - \bar{U}_0)(\eth + \bar{\eth})f \right)\left( \sigma_2 - \frac{1}{4}C \right)\mathfrak{L}$$

$$+ \frac{1}{2}e^{2\beta_0}\mathfrak{L}\left( \eth\bar{\eth} + \bar{\eth}\eth + \eth\eth - \bar{\eth}\bar{\eth} \right)f + \mathfrak{L}e^{2\beta_0}2\left( \eth\beta_0\bar{\eth}f + \eth\beta_0\eth f + \bar{\eth}\beta_0\eth f - \bar{\eth}\beta_0\bar{\eth}f \right), \tag{2.92}$$

which can be combined with (2.90c) to show that $\Psi_0^5$ does indeed *not* transform homogeneously.

As a conclusion for this subsection, we can note that all the transformation laws computed above, for any phase space function $\mathcal{O}$, take the general form

$$\delta_\xi\mathcal{O} = \left( f\partial_u + \pounds_Y + (j - w)\xi^r_{+1} \right)\mathcal{O} + \left( U_0 \cdot \mathcal{O} \cdot \partial f \right) + (\dots), \tag{2.93}$$

where the terms in $(\dots)$ do not depend on $\mathcal{O}$. The component $\xi^r_{+1}$ of the radial part of the vector field, which later on will be related in BS and NU gauge to the Weyl transformations, appears with a factor controlled by the spin and the conformal weight. For the various functionals considered here these labels are given by

| | $\mathcal{Q}_s$ | $\mathcal{E}_l$ | $\mathfrak{L}$ | $q_{AB}$ | $C_{AB}$ | $D_{AB}$ | $E_{AB}$ | $E_{lAB}$ | $\mathcal{P}_A$ | $\mathfrak{L}_{AB}$ | $\mathfrak{B}_{AB}^\Lambda$ | $C$ | $D$ | $m_0^A$ | $\bar{m}_0^A$ |
|---|---|---|---|---|---|---|---|---|---|---|---|---|---|---|---|
| $j$ | 0 | 0 | 0 | 2 | 2 | 2 | 2 | 2 | 1 | 2 | 2 | 0 | 0 | $-1$ | $-1$ |
| $w$ | 3 | 3 | 2 | 0 | 1 | 2 | 3 | 3 | 3 | 2 | 2 | 0 | 1 | 2 | 0 | 0 |

The second term in (2.93) is a contraction of the boundary source $U_0^A$ with an angular derivative of $f$ and the functional $\mathcal{O}$, whose particular index structure depends also on the spin of $\mathcal{O}$.

The tensor $C_{AB}$ has spin and weight such that $\eth_u C_{AB} = (\partial_u + 2\gamma_0)C_{AB}$, which is indeed the combination appearing in the generalized news (2.34). This latter, just like the news $N_{AB}$, has conformal dimension $w = 2$, so $\eth_u \mathbf{N}_{AB} = \partial_u \mathbf{N}_{AB}$ in agreement with (2.46).

### 2.6.3 Symmetry algebra

Let us finally discuss the algebra formed by the vector fields preserving the partial Bondi gauge. With the above transformation laws at our disposal, we can use the modified (or adjusted) Lie bracket [28] to compute the bracket between vector fields where the coefficients of the expansion (2.80) have an arbitrary field dependency. This generically leads however to algebroids with field-dependent structure functions. In order to bring the algebra in a more readable form, it is convenient to imagine having performed a field-dependent redefinition of the radial coefficients in order to write the vector field as

$$\xi^\mu = \left( f, h\,r + k + \frac{1}{2}e^{2\beta_0}\big(4\partial_A\beta_0 + D_A\big)\partial^A f + \mathcal{O}(r^{-1}), Y^A - \frac{1}{r}e^{2\beta_0}\partial^A f + \mathcal{O}(r^{-2}) \right). \qquad (2.94)$$

The function $h$ parametrizes the Weyl transformations and is common to the BS and NU gauges, while $k$ parametrizes the radial translations which only appear in the NU gauge. Expression (2.94) is precisely the form that the asymptotic Killing vector will take in NU gauge, or in BS gauge when setting $k = 0$.

There is a subtlety to be noted at this stage however, which is that (2.80) contains a priori an infinite tower of arbitrary functions. Here we are considering in (2.94) only the contributions from the leading functions $\xi^r_{+1}$ and $\xi^r_0$, which are the only ones which will survive the gauge fixing to BS and NU gauge later on. In any case, one can expect that computing the charges with (2.80) will reveal that the functions $\xi^r_{n>0}$ are pure gauge.

Using the modified bracket on (2.94) in order to take into account the field-dependency, we find

$$\big[\xi(f_1, h_1, k_1, Y_1), \xi(f_2, h_2, k_2, Y_2)\big]_\star = \big[\xi(f_1, h_1, k_1, Y_1), \xi(f_2, h_2, k_2, Y_2)\big] - \big(\delta_{\xi_1}\xi_2 - \delta_{\xi_2}\xi_1\big)$$
$$= \xi(f_{12}, h_{12}, k_{12}, Y_{12}), \qquad (2.95)$$

with

$$f_{12} = f_1\partial_u f_2 + Y_1^A\partial_A f_2 - \delta_{\xi_1}f_2 - (1 \leftrightarrow 2), \qquad (2.96a)$$
$$h_{12} = f_1\partial_u h_2 + Y_1^A\partial_A h_2 - \delta_{\xi_1}h_2 - (1 \leftrightarrow 2), \qquad (2.96b)$$
$$k_{12} = f_1\partial_u k_2 + Y_1^A\partial_A k_2 - h_1 k_2 - \delta_{\xi_1}k_2 - (1 \leftrightarrow 2), \qquad (2.96c)$$
$$Y_{12}^A = f_1\partial_u Y_2^A + Y_1^B\partial_B Y_2^A - \delta_{\xi_1}Y_2^A - (1 \leftrightarrow 2). \qquad (2.96d)$$

When $\delta(f, Y^A, h, k) = 0$, i.e. for field-independent parameters, we therefore obtain an algebra given by the double semi-direct sum

$$\big(\text{diff}(\mathcal{I}^+) \loplus \mathbb{R}_h\big) \loplus \mathbb{R}_k. \qquad (2.97)$$

Here $\text{diff}(\mathcal{I}^+)$ stands for the diffeomorphisms $\text{diff}(S^2)$ of the two-sphere, generated by $Y^A$, together with the temporal diffeomorphisms $\text{diff}(\mathbb{R})$ generated by $f$. Let us stress once again that this is only the asymptotic algebra, and that (2.80) contains a priori more independent functions. In the BS and NU gauge however only $h$ and $k$ survive.

From these basic commutation relations, we can then compute the structure functions of the algebra (or algebroid) obtain with any field-dependent redefinition of the parameters. We see for

example from (2.82) that freezing the boundary volume or turning off the boundary sources imposes field-dependent consistency conditions on the parameters, which must be taken into account in order to compute the algebra of vector fields in these special cases.

In the three-dimensional construction of the NU gauge with Weyl and radial translation symmetries, a change of slicing is required in order to obtain integrable charges [123]. This change of slicing is a field-dependent redefinition of the symmetry generators. Based on this three-dimensional result, one can guess that in the four-dimensional NU gauge with boundary sources a change of slicing will also be necessary in order to arrive at a "genuine" slicing, defined by the requirement that the charges be integrable in the absence of news [95]. If such a field-dependent redefinition of the generators is required in order to reach a genuine slicing, then the structure (2.96) of the algebra (2.97) will change accordingly. One can conjecture that the resulting algebra will be the semi-direct product of the $\Lambda$-BMS algebra with two abelian components (corresponding to the Weyl transformations and the radial translations), but the study of the charges is necessary in order to confirm this.

It is interesting to note that the symmetry algebra (2.97) bears similarities with the so-called extended corner symmetry algebra recently discussed in the literature. In [178, 179], it was argued that the corner symmetry algebra canonically associated with co-dimension 2 boundaries has the structure $\mathcal{A}^c = \big(\mathrm{diff}(S^2) \oplus \mathfrak{gl}(2, \mathbb{R})\big) \oplus \mathbb{R}^2$, which becomes $\mathcal{A}^c_{\mathcal{I}^+} = \big(\mathrm{diff}(S^2) \oplus \mathrm{Weyl}\big) \oplus \mathbb{R}^2$ asymptotically and in Bondi gauge [104] (i.e. a subgroup of the linear group becomes the Weyl factor). Similarly to what happens in three dimensions [123], one can expect that a change of slicing for the asymptotic Killing vector fields will lead to a rewriting of the algebra (2.97) in the form $\mathcal{A}^c_{\mathcal{I}^+}$, where $\mathrm{Weyl} \equiv \mathbb{R}_h$ and where the $\mathbb{R}^2$ encompasses the supertranslations and the radial translations $\mathbb{R}_k$. In this sense, the symmetry algebra (2.97) is an extension of the BMSW algebra which is closer to the universal corner algebra $\mathcal{A}^c$. This answers in part the question of which gauge and boundary conditions can lead to an asymptotic realization of $\mathcal{A}^c$. One can expect that the inclusion of $\mathfrak{gl}(2, \mathbb{R})$ will require a relaxation of the condition $g_{rA} = 0$ imposed in the partial Bondi gauge.

## 2.7 Symplectic potential

The symplectic potential is the object which defines the covariant phase space. It enables to identify the flux, gives rise to the symplectic structure which identifies the canonical pairs, and serves as the starting point for the computation of the charges. As is well-known, in some situations the symplectic structure may diverge asymptotically. Instead of curing these divergencies by strengthening the fall-off conditions, it is now accepted that a procedure of symplectic renormalization should instead be used [67, 69, 91, 98, 99, 165, 175, 180]. This is done by supplementing the symplectic potential with a corner term, which in turn can be understood as the symplectic potential of a boundary Lagrangian. In the three-dimensional study of the (generalized) NU gauge with cosmological constant and boundary sources, this was achieved in [123]. The initial step is to study the bare symplectic potential coming from the Einstein–Hilbert action.

The symplectic potential and its components of interest in the partial Bondi gauge are given

by[14]

$$\Theta^\mu = \sqrt{-g}\big(g^{\alpha\beta}\delta\Gamma^\mu_{\alpha\beta} - g^{\mu\alpha}\delta\Gamma^\beta_{\alpha\beta}\big), \tag{2.98a}$$

$$\Theta^u = \sqrt{\gamma}\Big(e^{2\beta}\gamma^{AB}\delta\Gamma^u_{AB} + \delta\big(\Gamma^r_{rr} + \Gamma^A_{rA}\big)\Big), \tag{2.98b}$$

$$\Theta^r = \sqrt{\gamma}\left(\delta\big(\Gamma^u_{uu} + \Gamma^A_{uA} - \Gamma^r_{ur}\big) + U^A\delta\big(\Gamma^u_{Au} + \Gamma^B_{AB} - \Gamma^r_{Ar}\big) + \frac{V}{r}\delta\Gamma^A_{rA} + e^{2\beta}\gamma^{AB}\delta\Gamma^r_{AB}\right). \tag{2.98c}$$

Our goal is to compute the divergent and finite pieces of the temporal and radial fluxes in the partial Bondi gauge, in order to then restrict the results to the BS and NU gauges. To do so, we note that $\sqrt{\gamma} = \mathcal{O}(r^2)$, so that the terms multiplying this volume above must be expanded to $\mathcal{O}(r^{-3})$ in order to obtain the finite and divergent pieces. An additional complication in the NU gauge (and therefore also in the partial Bondi gauge) is that the co-dimension 2 volume form has a non-trivial radial expansion (D.10b) due to the relaxation of the determinant condition. This has the effect of mixing various orders in the terms multiplying the volume in (2.98).

Let us start by analyzing the time component of the symplectic potential. This gives the symplectic structure on a constant-$u$ hypersurface. Using (D.10) and the expansions

$$\Gamma^u_{AB} = \frac{1}{2}e^{-2\beta_0}\big(2rq_{AB} + C_{AB}\big) + \mathcal{O}(r^{-1}), \tag{2.99a}$$

$$\Gamma^r_{rr} = \mathcal{O}(r^{-3}), \tag{2.99b}$$

$$\Gamma^A_{rA} = \frac{2}{r} - \frac{C}{2r^2} + \mathcal{O}(r^{-3}), \tag{2.99c}$$

we find

$$\Theta^u = \frac{1}{2}\sqrt{q}\Big(2r\delta\big(\ln q - 4\beta_0\big) + C_{AB}\delta q^{AB} + C\delta\big(\ln q - 2\beta_0\big)\Big) + \mathcal{O}(r^{-1}). \tag{2.100}$$

This is in agreement with (B.19) of [125] when $C = 0 = \beta_0$ (i.e. in BS gauge and without boundary sources). Again, it should be noted that here $C_{AB}$ is not trace-free if we are not in BS gauge.

The computation of the radial part of the potential is extremely lengthy, and detailed in appendix H. The only simplifying assumption which we use in this appendix to compute the radial potential is the absence of logarithmic terms in the solution space. We defer the study of holographic renormalization in the presence of logarithmic terms to future work. For clarity and simplicity we are going to report here the potential when the boundary sources are turned off by setting $\beta_0 = 0 = U_0^A$ (the full result is in the appendix). With these simplifications, the constraint $\mathfrak{B}^\Lambda_{AB} = 0$ which we have found in the solution space becomes

$$\partial_u q_{AB} \overset{!}{=} q_{AB}\partial_u \ln \sqrt{q} + \frac{\Lambda}{3}C^{\text{TF}}_{AB}. \tag{2.101}$$

Furthermore, it is more transparent to write the potential in BS and NU gauge separately. The expansion of the radial part of the potential is

$$\Theta^r = r^3\Theta^r_3 + r^2\Theta^r_2 + r\Theta^r_1 + \Theta^r_0 + \mathcal{O}(r^{-1}). \tag{2.102}$$

---

[14]We work in units in which $16\pi G = 1$, so that the Lagrangian is $L = \sqrt{-g}\,(R - 2\Lambda)$.

In the absence of boundary sources, the divergent pieces in BS gauge are

$$\Theta^r_3\big|_{\mathrm{BS}} = \frac{2\Lambda}{3}\delta\sqrt{q}, \tag{2.103a}$$

$$\Theta^r_2\big|_{\mathrm{BS}} = -\delta\partial_u\sqrt{q}, \tag{2.103b}$$

$$\Theta^r_1\big|_{\mathrm{BS}} = \frac{1}{2}\partial_u\big(\sqrt{q}\,q^{AB}\delta C^{\mathrm{TF}}_{AB}\big) - \delta\left(\sqrt{q}\,R[q] + \frac{\Lambda}{8}\sqrt{q}\,[CC]\right), \tag{2.103c}$$

while in NU gauge they are given by

$$\Theta^r_3\big|_{\mathrm{NU}} = \Theta^r_3\big|_{\mathrm{BS}}, \tag{2.104a}$$

$$\Theta^r_2\big|_{\mathrm{NU}} = \Theta^r_2\big|_{\mathrm{BS}} + \frac{\Lambda}{2}\delta\big(\sqrt{q}\,C\big), \tag{2.104b}$$

$$\Theta^r_1\big|_{\mathrm{NU}} = \Theta^r_1\big|_{\mathrm{BS}} - \frac{1}{2}\partial_u\big(\delta\sqrt{q}\,C\big) + \frac{\Lambda}{8}\delta\Big(\sqrt{q}(2C^2 - [CC])\Big). \tag{2.104c}$$

We see that all the divergent pieces of the potential (in either gauge) are total variations or total time derivatives. This is what enables to identify the counter-terms which must be used for the symplectic renormalization. It is actually expected that the divergent pieces of the potential take this general form, as this follows from the on-shell conservation of the symplectic current. Indeed, by definition of the symplectic potential we have on-shell that $\delta L \approx \mathrm{d}\Theta = \partial_u\Theta^u + \partial_r\Theta^r + \partial_A\Theta^A$, so the divergent pieces in $\Theta^r$ are necessarily the sum of total variations and total derivatives. This is the argument which explains why it is always possible to implement symplectic renormalization, and the above explicit rewriting of the divergent components of the potential therefore gives a useful consistency check. We can therefore perform the symplectic renormalization of the potential in the BS or NU gauges.

The finite piece $\Theta^r_0 = \sqrt{q}\,\theta^r_0$ of the potential is very lengthy and can be found in appendix H. In the absence of boundary sources we report its expression in BS and NU gauge in (H.17) and (H.18) respectively. Because the study of these terms is more involved, we postpone it to the sections below devoted specifically to the BS and NU gauges.

## 3   The BS gauge

We now explain in little more details how the partial Bondi gauge is reduced to the Bondi–Sachs (BS) gauge upon imposing the determinant condition. This provides a useful consistency check with known expressions in the literature, and also enables us to obtain the expression for the (A)dS symplectic potential with free boundary volume (i.e. the Weyl sector). We also explain how the reduction to the BS gauge determines the radial part of the asymptotic Killing vector.

### 3.1   Gauge fixing

As discussed at length in the previous section, the partial Bondi gauge used to bring the metric in the form (2.1) still leaves a freedom in the choice of the radial coordinate. In the BS gauge this is fixed by the so-called determinant condition

$$\mathcal{C}_\gamma := \partial_r\left(\frac{\gamma}{r^4}\right) = 0, \tag{3.1}$$

where $\gamma = \det \gamma_{AB}$. When this condition is imposed the radial coordinate is the areal distance (see (2.36)). In order to satisfy this determinant condition, with $\gamma = qr^4$, the components in the expansion (2.3) of the angular metric need to have a fixed trace. This can explicitly be seen for example in (D.10b). Each subleading term in the angular metric is then determined up to a trace-free tensor. Indeed, imposing the determinant condition at each order leads to

$$\mathcal{C}_\gamma = \mathcal{O}(r^{-3}) \qquad \Rightarrow \qquad C = 0, \tag{3.2a}$$

$$\mathcal{C}_\gamma = \mathcal{O}(r^{-4}) \qquad \Rightarrow \qquad D_{AB} = \frac{1}{4}q_{AB}[CC] + \mathscr{D}_{AB}, \tag{3.2b}$$

$$\mathcal{C}_\gamma = \mathcal{O}(r^{-5}) \qquad \Rightarrow \qquad \begin{cases} E_l = 0 \quad \Rightarrow \quad E_{lAB} = \mathscr{E}_{lAB}, \\ E_{AB} = \frac{1}{2}q_{AB}[C\mathscr{D}] + \mathscr{E}_{AB}, \end{cases} \tag{3.2c}$$

$$\mathcal{C}_\gamma = \mathcal{O}(r^{-6}) \qquad \Rightarrow \qquad \begin{cases} F_{l^2} = 0 \quad \Rightarrow \quad F_{l^2 AB} = \mathscr{F}_{l^2 AB}, \\ F_l = [CE_l] \quad \Rightarrow \quad F_{lAB} = \frac{1}{2}q_{AB}[C\mathscr{E}_l] + \mathscr{F}_{lAB}, \\ F_{AB} = \frac{1}{8}q_{AB}\left(4[C\mathscr{E}] + 2[\mathscr{D}\mathscr{D}] - \frac{1}{4}[CC]^2\right) + \mathscr{F}_{AB}, \end{cases} \tag{3.2d}$$

where $\mathscr{D}_{AB}$, $\mathscr{E}_{AB}$, $\mathscr{F}_{AB}$, $\mathscr{E}_{lAB}$, $\mathscr{F}_{lAB}$ and $\mathscr{F}_{l^2 AB}$ are all trace-free.

We see as expected that the determinant condition starts by setting $C = 0$. This implies in particular that in BS gauge we have $\mathfrak{L}_{AB} = \mathscr{D}_{AB}$, which is indeed the tensor which is set to zero in order to remove the logarithmic branches in BS gauge [181]. As explained in (E.5), it is important to recall that due to the determinant condition in BS gauge we have $\mathcal{D}_A P^A = D_A P^A$ for any vector $P^A$.

## 3.2 Solution space

In the "traditional" way of studying the solution space in BS gauge, one starts from the onset with the determinant condition, and therefore the terms in the expansion of the angular metric are determined up a trace-free part as in (3.2). Solving the $(rr)$ Einstein equation then leads to

$$\mathrm{E}_{rr} = \mathcal{O}(r^{-4}) \qquad \Rightarrow \qquad \beta_1 = 0, \tag{3.3a}$$

$$\mathrm{E}_{rr} = \mathcal{O}(r^{-5}) \qquad \Rightarrow \qquad \beta_2 = -\frac{1}{32}[CC], \tag{3.3b}$$

$$\mathrm{E}_{rr} = \mathcal{O}(r^{-6}) \qquad \Rightarrow \qquad \begin{cases} \beta_3 = -\frac{1}{12}[C\mathscr{D}], \\ \beta_{l3} = 0, \end{cases} \tag{3.3c}$$

$$\mathrm{E}_{rr} = \mathcal{O}(r^{-7}) \qquad \Rightarrow \qquad \begin{cases} \beta_{l^2 4} = 0, \\ \beta_{l4} = -\frac{3}{32}[CE_l], \\ \beta_4 = -\frac{1}{32}\left(3[C\mathscr{E}] + 2[\mathscr{D}\mathscr{D}] - \frac{1}{4}\big([CC]^2 + [CE_l]\big)\right). \end{cases} \tag{3.3d}$$

One can check that this is indeed consistent with plugging (3.2) in the solutions (2.7) obtained in the partial Bondi gauge. The construction of the solution space and the reduction to the BS or NU

gauge are of course commuting procedures. One can therefore obtain the solution space in BS gauge from that in the partial Bondi gauge by using everywhere in the previous section the expressions (3.2) obtained from the determinant condition.

In the case $\Lambda \neq 0$, as explained above and summarized in table 2, the BS gauge removes all the logarithmic terms. More precisely, the determinant condition (3.2) sets the traces of the logarithmic terms to be vanishing, while the trace-free parts are set to vanish by the $(AB)$ Einstein equations. As explained in [67], the $(AB)$ Einstein equations then give at every order an equation of the type $\Lambda \gamma_{AB}^n = \partial_u \gamma_{AB}^{n-1} + (\dots)$ for $n > 1$, which therefore does not constrain $\mathscr{E}_{AB}$. For example equation (I.1) shows that $\mathscr{F}_{AB}$ is indeed determined by $\partial_u \mathscr{E}_{AB}$. This means that the solution space is entirely determined by 11 functions of $(u, x^A)$. These are the 8 unconstrained functions $(\beta_0, U_0^A, q_{AB}, \mathscr{E}_{AB})$, and the 3 functions $(M, U_3^A)$ subject to evolution equations.

In the case $\Lambda = 0$, we still have the 6 functions $(\beta_0, U_0^A, q_{AB})$ but the time evolution of $q_{AB}$ is constrained by $\mathfrak{B}_{AB}^0 = 0$. There is then an infinite tower of functions appearing in the radial expansion of the angular metric (namely $\mathscr{E}_{AB}, \mathscr{F}_{AB}, \dots$) and admitting a non-trivial time evolution [28]. The logarithmic terms have a non-trivial time evolution as well, but this latter depends only on the boundary metric and sources, as for example in (2.17c) and (2.20). When the boundary metric has no time dependency and the boundary sources are turned off, the evolution of the logarithmic terms in $\gamma_{AB}$ can be integrated. For example $\mathfrak{L}_{AB} = \mathscr{D}_{AB}$ is a constant of the motion by virtue of (2.17c), and therefore $E_{lAB} = \mathscr{E}_{lAB}$ evolves linearly in $u$ according to (2.20). This is in accordance with equation (12) of [138], which is a particular case of our evolution equations.

## 3.3 Symplectic potential

In section 2.7 we have already presented the temporal component of the potential, as well as the divergent pieces of the radial potential. We now focus on the finite piece of the radial potential, gathering here once again the results of appendix H and restricting for clarity to the case of vanishing boundary sources. This will enable us to perform some consistency checks with the existing literature, and to present a new expression for the potential in (A)dS.

We recall that in BS gauge we have $C = 0$ and $N_{AB} = \partial_u C_{AB}^{\mathrm{TF}} = \partial_u C_{AB}$, and that the determinant condition and the solution space give (3.2) and (3.3). The potential in BS gauge can be found by imposing these conditions in (H.14d) and multiplying by the volume. This gives

$$
\begin{aligned}
\Theta_0^r\big|_{\mathrm{BS}} = {} & \frac{1}{4}\sqrt{q}\, q^{AB}\delta\left[C_{AB}\left(R[q] + \frac{5\Lambda}{8}[CC]\right) - 2\Lambda\mathscr{E}_{AB} - 2D_A D^C C_{BC}\right] \\
& + \delta\sqrt{q}\left(2M + \frac{1}{2}D_A D_B C^{AB} + \frac{1}{8}[CC]\partial_u \ln\sqrt{q}\right) - \frac{1}{2}\sqrt{q}\,C^{AB}\delta N_{AB} \\
& + \delta\left(2\sqrt{q}\,M + \frac{3}{8}\sqrt{q}\,\partial_u[CC] + \frac{1}{2}[CC]\partial_u\sqrt{q}\right) \\
& - \frac{1}{2}\sqrt{q}\,D_A\big(D_B C^{AB}\delta\ln\sqrt{q}\big) - \frac{1}{16}\partial_u\big(3\sqrt{q}\,\delta[CC] + 2[CC]\delta\sqrt{q}\big).
\end{aligned}
\tag{3.4}
$$

The last three terms are respectively a total variation, a total angular derivative, and a corner term.

This general expression features $\Lambda$, $\delta\sqrt{q}$, and $\partial_u\sqrt{q}$. The cosmological constant introduces the spin-2 $\mathscr{E}_{AB}$ in the potential and will enable us to reach the flat limit and also the (A)dS rewriting of the potential. The variation of the boundary volume gives us access to the Weyl sector and introduces the mass in the potential. Finally, the time derivative of the leading metric will lead to a $u$-dependent generalization of BMSW, and also allow us to have non-trivial radiation in (A)dS.

We can now check the consistency of this result in various limits. First, we note that the flat limit is well-defined. This is the advantage of the Bondi gauge over the Starobinsky–Fefferman–Graham gauge [67–69]. Taking $\Lambda = 0$ as well as $\partial_u q_{AB} = 0$ leads to the time-independent BMSW gauge of [125]. The above expression is then in agreement with (2.36) or (B.18) of [125] (up to the last corner term). Restricting in addition to $\delta\sqrt{q} = 0$, we can massage the potential to put it in the form of equation (5.17) of [175]. These limits provide a first consistency check for the expression of the potential in BS gauge when $\Lambda = 0$.

Let us now rewrite the symplectic potential in the case $\Lambda \neq 0$ in order to compare it with the result of [67]. For this we use the constraint (2.101) to rewrite the shear in terms of the time evolution of the leading metric as $\Lambda C^{AB} = -3\big(\partial_u q^{AB} + q^{AB}\partial_u \ln\sqrt{q}\big)$. Using this the (A)dS potential (3.4) can be rewritten as

$$\Theta^r_0\big|^{\Lambda\neq 0}_{\mathrm{BS}} = \frac{1}{2}\sqrt{q}\,q^{AB}\delta\Pi_{AB} + \delta\sqrt{q}\left(2M + \frac{1}{2}D_A D_B C^{AB} + \frac{1}{8}[CC]\partial_u \ln\sqrt{q}\right)$$
$$+ \delta\left(2\sqrt{q}\,M + \frac{3}{8}\sqrt{q}\,\partial_u[CC] + \frac{1}{2}[CC]\partial_u\sqrt{q}\right) - \frac{1}{2}\sqrt{q}\,D_A\big(D_B C^{AB}\delta\ln\sqrt{q}\big)$$
$$- \frac{1}{16}\partial_u\left(3\sqrt{q}\,\delta[CC] + 2[CC]\delta\sqrt{q} - \frac{24}{\Lambda}\sqrt{q}\,q^{AB}\delta N_{AB}\right), \tag{3.5}$$

where we have identified

$$\Pi_{AB} := \frac{1}{2}C_{AB}\left(R[q] + \frac{5\Lambda}{8}[CC]\right) - \Lambda\mathscr{E}_{AB} - \frac{1}{2}D_{(A}D^C C_{B)C} - \frac{3}{\Lambda}\partial_u N_{AB}. \tag{3.6}$$

In order to match exactly [67] we can further rewrite this potential using the tensor $\mathcal{S}_{AB}$ defined in (2.45) and appearing in the (A)dS mass loss formula. We recall that in BS gauge and in the absence of logarithmic terms this trace-free tensor is given by

$$\mathcal{S}_{AB} = \frac{\Lambda^2}{6}\mathscr{E}_{AB} - \frac{5\Lambda^2}{96}[CC]C_{AB} - \frac{\Lambda}{12}R[q]C_{AB}$$
$$+ \frac{1}{2}\partial_u\left(N_{AB} - \frac{1}{2}C_{AB}\partial_u \ln\sqrt{q}\right) - \frac{\Lambda}{4}q_{AB}\left([NC] - \frac{1}{2}[CC]\partial_u \ln\sqrt{q}\right)$$
$$- \frac{1}{2}\left(D_A D_B - \frac{1}{2}q_{AB}D_C D^C\right)\partial_u \ln\sqrt{q} + \frac{\Lambda}{12}\left(D_{(A}D^C C_{B)C} - q_{AB}D^C D^D C_{CD}\right). \tag{3.7}$$

Using the identities (E.9) and $q^{AB}\delta(q_{AB}T) = 2(\delta T + T\delta\ln\sqrt{q})$, a lengthy rewriting leads to

$$
\begin{aligned}
\Theta_0^r\big|_{\mathrm{BS}}^{\Lambda\neq 0} = {}& -\frac{3}{\Lambda}\sqrt{q}\,q^{AB}\delta\mathcal{S}_{AB} + \delta\sqrt{q}\left(2\mathcal{M} + \frac{3}{\Lambda}D^A D_A \partial_u \ln\sqrt{q}\right) \\
& + \delta\left(2\sqrt{q}\,M - \frac{5}{8}\partial_u(\sqrt{q}\,[CC]) - \frac{1}{2}\sqrt{q}\,D^A D^B C_{AB} - \frac{3}{2\Lambda}R[q]\partial_u\sqrt{q}\right) \\
& - \frac{1}{2}\sqrt{q}\,D_A\left(D_B C^{AB}\delta\ln\sqrt{q} + \frac{3}{\Lambda}\Big(2D^A\delta\ln\sqrt{q} + D_B\delta q^{AB} - \delta q^{AB}D_B\Big)\partial_u\ln\sqrt{q}\right) \\
& + \partial_u\left(\frac{1}{16}\sqrt{q}\,\delta[CC] + \frac{3}{2\Lambda}\sqrt{q}\,q^{AB}\delta N_{AB} - \frac{3}{4\Lambda}q^{AB}\delta C_{AB}\partial_u\sqrt{q} + \frac{3}{2\Lambda}R[q]\delta\sqrt{q}\right).
\end{aligned}
$$

(3.8)

This shows that the canonical partner to $\sqrt{q}\,q^{AB}$ is the quantity $\mathcal{S}_{AB}$ identified in the (A)dS mass loss formula (2.76), consistently with [67]. We note, as introduced in (2.45), that this quantity is not just the radiation tensor, but a shift of this later by the covariant spin-2. We also obtain here a generalization of the (A)dS potential to the case $\Lambda$-BMSW, and we see in particular that the scalar quantity conjugated to the boundary volume naturally involves the covariant mass (2.41) (written here in BS gauge and in the absence of boundary sources).

## 3.4 Vector fields and symmetries

We have seen in section 2.6 that the partial Bondi gauge admits asymptotic Killing vectors, where the functions in the expansion of the radial part (2.80) of the vector field are a priori free. The reduction to BS or NU gauge, which amounts to picking a radial coordinate, puts additional conditions on this radial expansion. In order to see this, we use the fact that the asymptotic Killing vectors in BS gauge must preserve the determinant gauge condition. This amounts to the condition $\partial_r(\pounds_\xi \ln\gamma) = 0$, where (2.79b) in BS gauge becomes

$$
\pounds_\xi \ln\gamma = f\partial_u \ln\gamma + \frac{4}{r}\xi^r + 2D_A\xi^A - 2U^A\partial_A f. \tag{3.9}
$$

This in turn fixes the radial part of the vector field to

$$
\xi^r = \frac{r}{2}\Big(2h - U_0^A\partial_A f - D_A I^A + U^A\partial_A f\Big), \qquad \partial_r h = 0, \tag{3.10}
$$

where $h(u, x^A)$ is an integration "constant" arising because the determinant gauge condition is differential. Here we have already used the freedom in redefining $h$ up to an $r$-independent function. The reason for performing this redefinition is to obtain the asymptotic expression (2.94) for the vector field, which as we have shown leads to an algebra (instead of an algebroid). The expansion of $\xi^r$ is

$$
\xi^r = rh + \frac{1}{2}e^{2\beta_0}\big(4\partial_A\beta_0 + D_A\big)\partial^A f + \mathcal{O}(r^{-1}). \tag{3.11}
$$

The radial component (3.10) together with (2.77) define the asymptotic Killing vector in BS gauge, where $f$, $Y^A$ and $h$ parametrize respectively supertranslations (which is an abuse of language since

here we also allow for $\Lambda \neq 0$), superrotations, and Weyl transformations. We note that there are no other constraints on these symmetry parameters, and that they have in particular an arbitrary time dependency. For field-independent $(f, Y^A, h)$ the algebra of these vector fields is simply (2.97) with $k = 0$, i.e.

$$\text{diff}(\mathcal{I}^+) \leftthreetimes \mathbb{R}_h. \tag{3.12}$$

This is nothing but the Weyl extension of the $\Lambda$-BMS symmetry studied in [67, 69] (although in these references the charge algebra has also been computed).

The transformation laws follow immediately from the general expressions derived in section 2.6 once we use the radial expansion (3.11). In particular we find

$$\delta_\xi \ln \sqrt{\gamma} = 2h + f \partial_u \ln \sqrt{q} - U_0^A \partial_A f + D_A Y^A, \tag{3.13a}$$

$$\delta_\xi q_{AB} = \left( f \partial_u + \pounds_Y + 2h \right) q_{AB} - U_{(A}^0 \partial_{B)} f, \tag{3.13b}$$

$$\delta_\xi \beta_0 = \left( f \partial_u + \pounds_Y \right) \beta_0 + \frac{1}{2} \left( h + \partial_u f + U_0^A \partial_A f \right), \tag{3.13c}$$

$$\delta_\xi U_0^A = \left( f \partial_u + \pounds_Y + \partial_u f + U_0^C \partial_C f \right) U_0^A - \partial_u Y^A - \frac{\Lambda}{3} e^{4\beta_0} \partial^A f, \tag{3.13d}$$

$$\delta_\xi C_{AB} = \left( f \partial_u + \pounds_Y + h \right) C_{AB} - \bar{U}_{(A}^0 \partial_{B)} f - 2 e^{2\beta_0} \left( 2 \partial_{(A} \beta_0 \partial_{B)} f + D_A \partial_B f \right)^{\text{TF}}. \tag{3.13e}$$

We recall that in BS gauge $\sqrt{\gamma} = r^2 \sqrt{q}$. We also note that with $\xi_0^r$ determined as in (3.11) the partial Bondi gauge transformation law (2.84a) becomes $\delta_\xi C = \left( f \partial_u + \pounds_Y - h \right) C$, so we can consistently set $C = 0$ in BS gauge.

Finally, let us mention for completeness the constraints on the symmetry parameters which arise when turning off the boundary sources and considering a time-independent boundary volume. This is the context of the time-independent BMSW study of [125] (the only difference being that we have both the cases $\Lambda \neq 0$ and $\Lambda = 0$ here). We can see from (3.13a), (3.13c), and (3.13d) that setting $\partial_u \sqrt{q} = 0$ and $\beta_0 = 0 = U_0^A$ imposes respectively the consistency conditions

$$2 \partial_u h + \partial_u \left( D_A Y^A \right) = 0, \qquad h = -\partial_u f, \qquad \partial_u Y^A = -\frac{\Lambda}{3} \partial^A f. \tag{3.14}$$

When $\Lambda = 0$ we therefore get $\partial_u Y^A = 0$ and $\partial_u^2 f = 0$, which implies that $f = T + uW$, where $T$ and $W$ are functions of $x^A$ only which parametrize the time-independent supertranslations and Weyl transformations. The resulting BMSW vector field is the one used in [54, 55, 124, 125] to identify the covariant functionals as the primary fields under the action of $\text{Diff}(S^2) \times$ Weyl parametrized by $(Y^A, W)$, and to identify the asymptotic equations of motion (equivalent to the NP ones) by looking for primaries under the action of $T$. With these restrictions on the solution space and the symmetry parameters, we can take the time derivative of (3.13e) to find that the transformation of the news is given by

$$\delta_\xi N_{AB} \big|_{\text{BMSW}} = \left( f \partial_u + \pounds_Y \right) N_{AB} - 2 D_A \partial_B W + q_{AB} D_C \partial^C W. \tag{3.15}$$

In general, the time derivative of the transformation (2.85a) of $C_{AB}^{\text{TF}}$ in the partial Bondi gauge, or the transformation of the generalized news (2.34), will be much more complicated due to the

presence of time-dependency and the boundary sources. It should however be studied in the future in order to understand the vacuum structure in the partial Bondi gauge, and in particular what will play the role of the Geroch tensor.

# 4 The NU gauge

We finally turn to the study of the NU gauge. While this latter has been studied previously in the Newman–Penrose formalism in absence of boundary sources and with $C = 0$ [127, 128, 136], here the partial Bondi gauge enables us to present a treatment which is closer to that of the BS gauge.

## 4.1 Gauge fixing

Here again, the goal is to complete the partial Bondi gauge with a fourth condition. The historical Newman–Unti gauge condition which realizes this is $g_{ur} = -1$ [9]. We recall from (2.35) that this amounts to choosing the radial coordinate as the affine parameter for the null congruence.

Following the idea of Barnich and Troessaert to replace the algebraic determinant condition by a differential condition in order to unleash the Weyl rescalings [28, 29], here we can also replace the algebraic condition of Newman–Unti by a differential condition. What we take as the definition of the NU gauge is then the generalized (or relaxed) Newman–Unti condition

$$\partial_r g_{ur} = 0 \;\Rightarrow\; g_{ur} = -e^{2\beta_0}, \tag{4.1}$$

with $\beta_0(u, x^A)$ arbitrary. We see that in order to describe this (generalized) NU gauge it is crucial that we have included from the onset the boundary source $\beta_0$ in our analysis of the partial Bondi gauge. The NU gauge condition is then simply the requirement that $\beta_{n>0} = 0$. We are going to see that allowing for $\beta_0$ in the NU gauge is the ingredient which enables to access the Weyl rescalings. As in the BS gauge, the Weyl transformations therefore appear when going from an algebraic gauge condition to a differential one. We note that the Weyl transformations are already described in the Newman–Penrose analysis of the NU gauge in [127], where stereographic coordinates are used. We will see shortly that the NU gauge also allows for an extra radial symmetry generator which is implicitly set to zero in this last reference.

## 4.2 Solution space

If one builds the solution space starting from the NU gauge condition $g_{ur} = -e^{2\beta_0}$, the $(rr)$ Einstein equation determines the expansion of the angular metric as

$$\mathrm{E}_{rr} = \mathcal{O}(r^{-5}) \qquad \Rightarrow \qquad D_{AB} = \frac{1}{8}q_{AB}[CC] + \mathscr{D}_{AB}, \tag{4.2a}$$

$$\mathrm{E}_{rr} = \mathcal{O}(r^{-6}) \qquad \Rightarrow \qquad \begin{cases} E_l = 0 \\ E_{AB} = \frac{1}{3}q_{AB}\left([C\mathscr{D}] - \frac{1}{4}C[CC^{\mathrm{TF}}]\right) + \mathscr{E}_{AB}, \end{cases} \tag{4.2b}$$

where $\mathscr{D}_{AB}$ and $\mathscr{E}_{AB}$ are both trace-free. One can check that this is indeed consistent with the gauge condition, since plugging these solutions in (2.7) will lead to vanishing coefficients for $\beta_{n>0}$.

We do not give here the on-shell expression for $F_{AB}$, which is obtained at the next order, because it gets too lengthy and it is not necessary for our purposes. It can however be computed from the vanishing of $\beta_4$ given in (F.1c).

Equivalently, we can also reduce the solution space in partial Bondi gauge to that in NU gauge by imposing the vanishing of the coefficients (2.7) of the expansion of $\beta$. At each order this vanishing fixes the trace of a subleading component of the angular metric, and we can then determine this component accordingly, up to a trace-free part. This gives the expressions (4.2) above.

## 4.3  Symplectic potential

The temporal component of the potential and the divergent pieces of the radial potential have been given above in (2.100) and (2.104). One can clearly see by comparison with (2.103) the difference between the BS and NU gauges. It should be noted that these two gauges do *not only* differ by $C$, but also by terms arising from the fact that the two solution spaces are actually different, as can be seen e.g. by comparing (3.2) and (4.2).

In the absence of boundary sources, the constant piece of the radial potential in NU gauge is given in (H.18). Since this expression is quite involved, let us study it in the flat case and with $\partial_u q_{AB} = 0 = \delta\sqrt{q}$. For comparison with the BS gauge, we should furthermore rewrite the potential explicitly in terms of $C_{\mathrm{TF}}^{AB}$. This can be done using in particular (A.1h). Denoting $N_{AB} = \partial_u C_{AB}^{\mathrm{TF}}$, we then find with the simplifying assumptions that

$$
\boxed{
\begin{aligned}
\Theta_0^r\big|_{\mathrm{NU}} &= \frac{1}{4}\sqrt{q}\left(2N_{AB}\delta C_{\mathrm{TF}}^{AB} - \delta q^{AB}\left(C_{AB}^{\mathrm{TF}}R[q] - 2D_A D^C C_{BC}^{\mathrm{TF}} + D_A\partial_B C\right) + \delta C R[q]\right) \\
&\quad + \delta(\dots) + \partial_u(\dots) + \sqrt{q}\,D_A(\dots).
\end{aligned}
}
\tag{4.3}
$$

We have omitted the total derivatives and variations since they do not play a role. Already in this simplified setup, where we have removed the cosmological constant, the boundary sources, the Weyl factor, and the time-dependency of the leading metric, we see that the trace $C$ appears in two places in the potential. It modifies the canonical pair conjugated to $q^{AB}$, and also appears as conjugated to the two-dimensional Ricci scalar $R[q]$. We therefore conclude that the finite piece of the potential in NU gauge differs from that in BS gauge by trace terms. These terms however drop if we further restrict the setup to that of the original BMS analysis, with a fixed two-sphere metric $q_{AB}^\circ$ such that $\delta q_{AB}^\circ = 0 = \delta R[q^\circ]$. Any relaxation of these historical boundary conditions will give rise to a difference between the BS and NU gauges.

The fact that the BS and NU potentials differ is a surprise, and it suggests that the two gauges may actually be physically inequivalent. This does not contradict the equivalence statements made e.g. in [126, 127, 129], because these references set $C = 0$. We already see on the simplified setup leading to (4.3) that with $C \neq 0$ there is an extra canonical pair in the phase space. The general formula (H.18) shows that for a more general setup (e.g. with $\Lambda \neq 0$ and a time-dependent Weyl factor) the BS and NU potentials will differ also by terms not involving $C$. Once again, this is because the solution space is different as we have seen above. As we are about to see in the next section, it turns out that the presence of $C$ in NU gauge is intimately related to the existence of an

extra radial asymptotic symmetry. Ultimately, one should of course compute whether $C$ and/or the extra symmetry parameter appear in the charges for these asymptotic symmetries. We defer this heavy task to future work, and present for the time being the structure of the symmetries.

## 4.4 Vector fields and symmetries

In order to find the asymptotic Killing vectors in NU gauge, we simply need to study how the NU gauge condition (4.1) constrains the expansion (2.80) of the radial component found in the partial Bondi gauge. Preserving the NU gauge condition requires $\partial_r(\pounds_\xi g_{ur}) = 0$, where in NU gauge (2.79c) becomes

$$\pounds_\xi g_{ur} = -e^{2\beta_0}\Big(2\big(f\partial_u + \xi^A\partial_A\big)\beta_0 + \partial_r\xi^r + U^A\partial_A f + \partial_u f\Big). \tag{4.4}$$

This in turn fixes the radial part of the vector field to satisfy

$$\partial_r\xi^r = h + U_0^A\partial_A f - 2I^A\partial_A\beta_0 - U^A\partial_A f, \qquad \partial_r h = 0, \tag{4.5}$$

where $h(u, x^A)$ is an integration "constant", which we have already shifted here in an $r$-independent manner in order to obtain a vector field forming an algebra instead of an algebroid. Upon further integrating (4.5), we see that the radial component $\xi^r$ will contain two integration functions, $h(u, x^A)$ and a new $k(u, x^A)$, so that $\xi$ has in total 5 free functions $(f, Y^A, h, k)$ of $(u, x^A)$. To obtain the asymptotic expansion of the vector field we first use (2.81) in NU gauge, and then integrate (4.5) to get

$$\xi^r = r\,h + k + \frac{1}{2}e^{2\beta_0}\big(4\partial_A\beta_0 + D_A\big)\partial^A f + \mathcal{O}(r^{-1}), \tag{4.6}$$

where a log term has dropped because of (2.10a), and where we have once again used the freedom in redefining the integration constant in order to shift $k$. This expansion of the radial vector field should be compared with its BS analogue (3.11). With our choices of integration constants, we see that these vectors conveniently differ at leading order only by the presence of the additional symmetry parameter $k$ in NU gauge. At lower orders the components $\xi^r$ in BS and NU gauge actually also differ, in a way which we do not make explicit here but which can easily be computed.

The transformation laws can be found by plugging the form of the radial vector field in NU gauge into the general formulas of section 2.6. Because at leading order the vector field is the same in BS and NU gauge up to the new symmetry parameter $k$, most of the transformation laws are also the same up to the action of $k$. More precisely, here in NU gauge we get

$$\delta_\xi \ln\sqrt{\gamma} = 2h + f\partial_u \ln\sqrt{q} - U_0^A\partial_A f + D_A Y^A + \mathcal{O}(r^{-1}), \tag{4.7}$$

which agrees at leading order with the BS counterpart (3.13a). Note that here this equation is computing the transformation of the expansion (D.10b), while in BS gauge the transformation is exact because of the determinant condition. We see that here $h$ plays also the role of a Weyl rescaling, and that it is allowed to appear in (4.5) precisely because we have considered the differential version (4.1) of the NU gauge. The presence of the Weyl symmetry is therefore related to the inclusion of the

boundary source $\beta_0$ in NU gauge, while it is related to the inclusion of a fluctuating boundary volume $\sqrt{q}$ in BS gauge. In either gauge, the Weyl symmetry is allowed because we are using a differential gauge condition rather than an algebraic one. The algebra formed by the vector field in NU gauge, which contains the free functions $(f, Y^A, h, k)$, is the time-dependent algebra $\big(\mathrm{diff}(\mathcal{I}^+) \oplus \mathbb{R}_h\big) \oplus \mathbb{R}_k$.

The transformation laws for the fields $\beta_0$, $U_0^A$, and $q_{AB}$ are the same as in BS gauge and given again by (3.13c), (3.13d), and (3.13b). The influence of the new symmetry generator $k$ is to act on subleading components of the angular metric, and we find

$$\big(\delta_\xi C_{AB}\big)\big|_{\mathrm{NU}} = \big(\delta_\xi C_{AB}\big)\big|_{\mathrm{BS}} + 2k q_{AB}, \tag{4.8}$$

while for the trace we find

$$\delta_\xi C = \big(f\partial_u + \pounds_Y - h\big)C + 4k. \tag{4.9}$$

We therefore see that (4.8) has acquired a shift in $k$ with respect to the BS analogue (3.13e). The transformation laws (3.13c), (3.13d), (3.13b), (4.7), and (4.8), are the generalization to four-dimensional gravity of the three-dimensional results (2.12) in [123]. Note that (4.8) should be read with care since in BS gauge $C_{AB}$ is trace-free while this is not the case in NU gauge. What we mean by this equation is simply that the transformation of $C_{AB}$ in NU gauge is given by the same formula (3.13e) as the transformation in BS gauge, although now with a *trace-full* $C_{AB}$. In any case, this is consistent with plugging $\xi_{+1}^r$ and $\xi_0^r$ in (2.83b).

Equation (4.9) shows that the new symmetry parameter $k$ appearing in the NU asymptotic Killing vector is related to the presence of the non-vanishing trace $C$ in the said gauge. This trace $C$ can be set to zero by a choice of origin for the affine parameter (which in NU gauge is the radial coordinate $r$), but the extra symmetry parameter at order $r^0$ in $\xi^r$ can shift this value. Said the other way around, one can argue that the presence of this symmetry allows to fix $C = 0$. This is used explicitly e.g. in [129]. But the question remains whether this symmetry is gauge or physical, i.e. associated to a (non-)vanishing charge. In the three-dimensional analogue of the generalized NU gauge studied here, there is indeed a non-vanishing charge associated with these radial translations, and they furthermore form an Heisenberg algebra with the Weyl charges [123]. In order to study the (possible) charge associated with $k$ in the present four-dimensional context, one should first investigate the simplified solution space leading to the potential (4.3) (where $C$ survives). The study of the Weyl sector is made much more complicated by the fact that in NU gauge it requires the presence of the boundary source $\beta_0$, which as we have shown in appendix H renders the analysis of the symplectic potential extremely heavy. One could however simply use the Iyer–Wald or Barnich–Brandt formula (which can be applied without knowledge of the symplectic potential) to get a first idea of what the structure of the charges will be, but we expect, based on results from the three-dimensional theory, that extracting the proper charges will require the use of a corner ambiguity and also a possible change of slicing [91, 95, 96, 123]. All these aspects will be investigated in forthcoming work.

# 5  Perspectives

In this article we have studied the structure of asymptotically locally (A)dS and flat gravity in Bondi coordinates, using the partial Bondi gauge (2.1). This partial gauge, built with only three gauge fixing conditions and with no determinant condition, enables to treat together the Bondi–Sachs (BS) and Newman–Unti (NU) gauges, which can each be reached at any later stage of a calculation in the partial gauge by further imposing a specific fourth condition. So far, the Newman–Unti solution space has been studied in the literature using the Newman–Penrose formalism [127, 161, 162]. Here we have presented an analysis of the solution space which is by design similar to the analysis in BS gauge. This enables to clearly contrast the two gauge choices.

We have recalled why the NU gauge features an extra mode, which is the trace of $C_{AB}$, while this trace is set to vanish in BS gauge by the determinant condition. The NU gauge also admits an extra asymptotic symmetry, which is parametrized by a function at order $\mathcal{O}(r^0)$ in the radial part of the asymptotic Killing vector. Together with the supertranslations, the superrotations, and the Weyl transformations, which are also present in BS gauge, this extra component in NU gauge leads to the asymptotic symmetry algebra

$$\left( \text{diff}(\mathcal{I}^+) \uplus \mathbb{R}_h \right) \uplus \mathbb{R}_k. \tag{5.1}$$

The extra radial translation $k$ present in NU gauge acts precisely by shifting the trace $C$. The fate of this trace mode and that of the radial symmetry are therefore related. Indeed, one should now compute the asymptotic charges in order to see whether the radial symmetry (used to fix the value of $C$ in [127, 129]) is pure gauge or physical (as in the three-dimensional case [123]). We have chosen to postpone this investigation to future work, but the key ingredients for this calculation have been laid out here.

In this study of the partial Bondi gauge we have also included logarithmic terms in the solution space by working with the polyhomogeneous expansion (2.3). We have seen that, consistently, all these logarithmic terms are forced to vanish when $\Lambda \neq 0$ when completing the partial gauge to BS or NU gauge and going on-shell. When $\Lambda = 0$ however, the logarithmic terms are involved in evolution equations and also modify the Weyl scalars. Moreover, we have kept in the construction of the solution space a free boundary metric with time-dependent fields $\beta_0$, $U_0^A$, and $q_{AB}$. The solution space constructed here is therefore an extension of that in [67] in two respects: the relaxation to the partial Bondi gauge, and the inclusion of the logarithmic terms. The prospects are however similar, namely to also study sourced holography and understand radiation in (A)dS. We have seen how, in this endeavor, the Newman–Penrose formalism using the Weyl scalars and spin coefficients proves to be very useful. This has enabled us in particular to recover the (A)dS evolution equation (2.76) for the mass, and to identify the quantity $\mathcal{S}^{AB}$ sourcing this equation and conjugated to $q_{AB}$ in the symplectic structure (3.8), as the radiation tensor shifted by the covariant spin-2. This use of the NP formalism bypasses the symmetry arguments used in [124] to identify the covariant functionals and derive the evolution equations.

There are many interesting prospects to be explored in future work, and which rely on some of the extensions of the asymptotic structure and the solution space presented here. A non-exhaustive list is as follows:

- **Radial translation charges.**
  An immediate and important follow-up is to compute the asymptotic charges and the asymptotic charge algebra in the partial Bondi gauge. The question is that of the fate of the new generator $k$ in NU gauge, or equivalently the term of order $\mathcal{O}(r^0)$ in the partial Bondi gauge expansion (2.80). In three dimensions, we have shown in [123] that this radial translation is a physical symmetry with a non-vanishing charge, and a choice of corner term and integrable slicing was necessary in order to obtain integrable charges. In the four-dimensional case, one can expect that similar subtleties will arise, and that there will be a preferred genuine slicing in which the charges become integrable in the absence of radiation. Although, as we have seen, there is no flux-balance law associated with the trace mode $C$, it would be interesting to investigate whether and how this changes in the presence of matter [134, 135] and in modified theories of gravity [130–133].

- **Subleading Weyl scalars and higher spin charges.**
  In the flat case there is an infinite tower of functionals whose time evolution is constrained. These are the subleading components of the angular metric, which appear in "covariant" form in the subleading terms $\Psi_0^{n>5}$ in $\Psi_0$. We have seen here how $E_{AB}^{\mathrm{TF}}$ appears in covariant form $\mathcal{E}_{AB}$ in $\Psi_0^5$. These subleading terms are related to the higher spin charges [51, 54] and to the Newman–Penrose conserved charges [182–185]. We have explained here how these subleading terms can have mixed helicities. It would be interesting for future work to analyse these subleading terms in details (not necessarily in the partial Bondi gauge), in order to understand their relationship with the higher spin symmetries and to relate them with the operators defined in the context of celestial holography.

- **Logarithmic terms.**
  Another aspect of the present work which should be explored further concerns the polyhomogeneous expansion and the role of the logarithmic terms. While in the asymptotically locally (A)dS case the logarithmic terms are necessarily vanishing on-shell, in the flat case they are subject to evolution equations which we have derived. The question is now how these logarithmic terms affect the symplectic renormalisation of the potential and the construction of the charges. We have also seen how the logarithmic terms relax the standard fall-offs of Weyl scalars, and one should now study how they modify the subleading terms in the Weyl scalars and the Newman–Penrose charges [182–185]. Moreover, it would be interesting to understand whether the logarithmic terms in the solution space are related to the log soft theorem [151–153]. Finally, let us mention an example where logarithmic modes play an important role. This is the case of three-dimensional chiral gravity [186], where it has been shown that there is a new mode, the "logarithmic primary", which violates the usual Brown–Henneaux boundary conditions and requires a larger class of boundary conditions [187–194]. While the setup is quite different from the present analysis, this shows nonetheless that logarithmic modes can arise in certain contexts. One should also extend the present construction to include matter in order to see whether the logarithmic terms can lead to new flux-balance laws and memory effects.

- **Radiation in (A)dS.**

  The notion of radiation for spacetimes with $\Lambda \neq 0$ is poorly understood [70–85]. Although here we have not directly discussed notions pertaining to gravitational waves and quadrupole formulas in (A)dS, we have seen how certain well-defined quantities in asymptotically flat spacetimes generalize to the case $\Lambda \neq 0$ and $\partial_u q_{AB} \neq 0$ (the question of whether these remain the relevant quantities is however open). For example, we have proposed a notion of generalized news in (2.34), and also identified the role of the unconstrained spin-2 in the quantity $\mathcal{S}^{AB}$ sourcing the mass loss and entering the symplectic structure. We hope to clarify in future work the role of these quantities when studying the memory effects in (A)dS.

- **Robinson–Trautman solutions.**

  Our extended solution space, which includes in particular arbitrary time-dependent sources for the boundary metric, is adapted to the study of the algebraically special Robinson–Trautman spacetimes [156–160, 175]. These spacetimes are interesting in their own right in order to study sourced holography and memory effects, but could also provide a simplified setup in which to study the role of the trace mode $C$ and the radial symmetry in NU gauge.

## Acknowledgments

It is our pleasure to thank Miguel Campiglia, Adrien Fiorucci, Laurent Freidel, Daniel Grumiller, Yannick Herfray, Alok Laddha, Pujian Mao, Blagoje Oblak, Marios Petropoulos, Daniele Pranzetti, Ana-Maria Raclariu, Romain Ruzziconi, Ali Seraj, and Simone Speziale for numerous discussions. We also warmly thank Miguel Campiglia and the Universidad de la República for their hospitality during the Montevideo Workshop on Celestial Symmetries, where part of this work was completed. Research at Perimeter Institute is supported in part by the Government of Canada through the Department of Innovation, Science and Economic Development Canada and by the Province of Ontario through the Ministry of Colleges and Universities.

## Appendices

## A Notations and conventions

We gather here some notations and conventions used throughout the main text, as well as some useful identities.

- All the angular indices $A, B, \ldots$ are lowered and raised with the leading order angular metric $q_{AB}$. We therefore have $C^{AB} = q^{AC}q^{BD}C_{BD}$, and $U_A = q_{AB}U^B$.

- The one and only exception to this rule is with the full angular metric $\gamma_{AB}$, whose true inverse such that $\gamma^{AC}\gamma_{BC} = \delta^A_B$ is denoted $\gamma^{AB}$.

- It is sometimes more compact to lower and raise indices with other two-dimensional tensors, in particular in the study of the symplectic potential in appendix H, so in these cases we write

$$U_A = q_{AB}U^B, \qquad \tilde{U}_A = \gamma_{AB}U^B, \qquad \bar{U}_A = C_{AB}U^B, \qquad \bar{\bar{U}}_A = D_{AB}U^B.$$

- The contraction of indices is denoted with square brackets, and in particular we have

$$[MN] = q_{AB}M^A N_B, \qquad [\tilde{M}N] = \gamma_{AB}M^A N^B, \qquad [\bar{M}N] = C_{AB}M^A N^B,$$

as well as objects such as $[CC] = C^{AB}C_{AB}$.

- The traces in the metric $q_{AB}$ are denoted by index-free objects, e.g. $C = q^{AB}C_{AB}$.

- Trace-free components are denoted by $C_{AB}^{\mathrm{TF}} = C_{AB} - \frac{1}{2}Cq_{AB}$, and it is easy to see that we therefore have $[C^{\mathrm{TF}}D^{\mathrm{TF}}] = [CD^{\mathrm{TF}}] = [C^{\mathrm{TF}}D]$.

- The covariant derivative associated with $\gamma_{AB}$ is $\mathcal{D}_A$ and satisfies $\gamma_{AB}\mathcal{D}_C U^B = \mathcal{D}_C \tilde{U}_A$.

- The covariant derivative associated with $q_{AB}$ is $D_A$ and satisfies $q_{AB}D_C U^B = D_C U_A$.

- For the expansions in $r$ we use the big-$\mathcal{O}$ notation, so that solving an equation $\mathrm{E} = \mathcal{O}(r^{-n})$ means that the terms in $r^{-n+1}$ and above have been killed. By abuse of notation we also use $\mathcal{O}(r^{-n})$ even for the polyhomogeneous expansions containing logarithmic terms.

- We define symmetrization as $P_{(A}Q_{B)} = P_A Q_B + P_B Q_A$, i.e. without factor $1/2$.

A few useful identities used throughout the main text are

$$C_{AB}\delta q^{AB} + C^{AB}\delta q_{AB} = 0, \tag{A.1a}$$

$$q_{AB}\delta C^{AB} + q^{AB}\delta C_{AB} = 2\delta C, \tag{A.1b}$$

$$C_{(AC}D^C_{B)} = \big([CD] - CD\big)q_{AB} + DC_{AB} + CD_{AB}, \tag{A.1c}$$

$$C_{AC}C^C_B = \frac{1}{2}\big([CC] - C^2\big)q_{AB} + CC_{AB}, \tag{A.1d}$$

$$C_{AC}C^{CD}C_{DB} = \frac{1}{2}C\big([CC] - C^2\big)q_{AB} + \frac{1}{2}\big([CC] + C^2\big)C_{AB}, \tag{A.1e}$$

$$\frac{1}{2}\delta[CC] = \big(C^2 - [CC]\big)\delta \ln \sqrt{q} + C^{AB}\big(\delta C_{AB} - C\delta q_{AB}\big), \tag{A.1f}$$

$$\big(\delta \ln \sqrt{q} + \delta\big)\partial_u \ln \sqrt{q} = \frac{1}{\sqrt{q}}\delta\partial_u \sqrt{q}, \tag{A.1g}$$

$$\partial_u C_{AB}\delta C^{AB} = \partial_u C_{AB}^{\mathrm{TF}}\delta C_{\mathrm{TF}}^{AB} - \frac{1}{2}C_{AB}^{\mathrm{TF}}\delta\big(C\partial_u q^{AB}\big)$$

$$+ \left(\frac{1}{2}q_{AB}\partial_u + \partial_u q_{AB}\right)\big(C\delta C_{\mathrm{TF}}^{AB}\big)$$

$$+ \frac{1}{2}\left(\frac{1}{2}C^2\partial_u q_{AB}\delta q^{AB} + \partial_u C\delta C + C\delta C\partial_u \ln \sqrt{q} - C\partial_u C\delta \ln \sqrt{q}\right). \tag{A.1h}$$

We give other useful identities involving covariant derivatives in appendix E below.

## B  Christoffel coefficients

We gather here the components of the Christoffel connection for the metric (2.1) in partial Bondi gauge. Noting that here the repeated angular indices in $\Gamma^A_{\mu A}$ are summed, and using the notation $\tilde{U}_A = \gamma_{AB} U^B$, we have

$$\Gamma^u_{\mu r} = 0, \tag{B.1a}$$

$$\Gamma^A_{rr} = 0, \tag{B.1b}$$

$$\Gamma^u_{uu} = \frac{1}{2} e^{-2\beta} \partial_r \left( e^{2\beta} \frac{V}{r} + \gamma_{AB} U^A U^B \right) + 2 \partial_u \beta, \tag{B.1c}$$

$$\Gamma^u_{Au} = -\frac{1}{2} e^{-2\beta} \partial_r \tilde{U}_A + \partial_A \beta, \tag{B.1d}$$

$$\Gamma^u_{AB} = \frac{1}{2} e^{-2\beta} \partial_r \gamma_{AB}, \tag{B.1e}$$

$$\Gamma^r_{uu} = \frac{1}{2} e^{-2\beta} \left( \left( \frac{V}{r} \partial_r + U^A \partial_A \right) g_{uu} + \partial_u \gamma_{AB} U^A U^B \right) + \frac{1}{2r} \left( 2V \partial_u \beta - \partial_u V \right), \tag{B.1f}$$

$$\Gamma^r_{ur} = -\frac{1}{2} e^{-2\beta} \left( \partial_r \left( e^{2\beta} \frac{V}{r} \right) + \frac{1}{2} \gamma_{AB} \partial_r \left( U^A U^B \right) \right) - U^A \partial_A \beta, \tag{B.1g}$$

$$\Gamma^r_{rr} = 2 \partial_r \beta, \tag{B.1h}$$

$$\Gamma^r_{uA} = -\frac{1}{2} e^{-2\beta} \left( \frac{V}{r} \partial_r \tilde{U}_A + U^B \partial_u \gamma_{AB} + U^B \partial_B \tilde{U}_A + \tilde{U}_B \partial_A U^B \right) - \frac{1}{2r} \partial_A V, \tag{B.1i}$$

$$\Gamma^r_{Ar} = \frac{1}{2} e^{-2\beta} \gamma_{AB} \partial_r U^B + \partial_A \beta, \tag{B.1j}$$

$$\Gamma^r_{AB} = \frac{1}{2} e^{-2\beta} \left( \left( \partial_u + \frac{V}{r} \partial_r \right) \gamma_{AB} + \mathcal{D}_A \tilde{U}_B + \mathcal{D}_B \tilde{U}_A \right), \tag{B.1k}$$

$$\Gamma^A_{uu} = \frac{1}{2} U^A \left( e^{-2\beta} \partial_r g_{uu} + 4 \partial_u \beta \right) - \frac{1}{2} \gamma^{AB} \left( \partial_B g_{uu} + 2 \partial_u \tilde{U}_B \right), \tag{B.1l}$$

$$\Gamma^A_{uB} = -\frac{1}{2} e^{-2\beta} U^A \partial_r \tilde{U}_B + \frac{1}{2} \gamma^{AC} \left( \partial_C \tilde{U}_B - \partial_B \tilde{U}_C + \partial_u \gamma_{BC} \right) + U^A \partial_B \beta, \tag{B.1m}$$

$$\Gamma^A_{uA} = -\frac{1}{4} e^{-2\beta} \left( \partial_r \gamma_{AB} U^A U^B + \partial_r [\tilde{U} U] \right) + \partial_u \ln \sqrt{\gamma} + U^A \partial_A \beta, \tag{B.1n}$$

$$\Gamma^A_{rB} = \frac{1}{2} \gamma^{AC} \partial_r \gamma_{BC}, \tag{B.1o}$$

$$\Gamma^A_{rA} = \partial_r \ln \sqrt{\gamma}, \tag{B.1p}$$

$$\Gamma^A_{ur} = \frac{1}{2} \gamma^{AB} \left( \partial_B e^{2\beta} - \partial_r \tilde{U}_B \right), \tag{B.1q}$$

$$\Gamma^C_{AB} = \Gamma^C_{AB}[\gamma] + \frac{1}{2} e^{-2\beta} U^C \partial_r \gamma_{AB}. \tag{B.1r}$$

# C  Spin coefficients and frame identities

Using the vectors $\ell$ and $n$ defined in (2.27) and the null frames (2.37), the Ricci rotation coefficients are defined by

$$\gamma_{ijk} := e_j^\mu e_k^\nu \nabla_\nu e_{i\mu}. \tag{C.1}$$

Below we give their explicit expression in the partial Bondi gauge, as well as their on-shell expansion using the expansion (2.54a) for the frames. We find

$$\alpha = \frac{1}{2}(\gamma_{124} - \gamma_{344}) = \frac{1}{2}\left(\frac{V}{2r}\Gamma_{rA}^u - \Gamma_{rA}^r - \hat{\bar{m}}^B \nabla_A \hat{m}_B\right)\hat{\bar{m}}^A = \frac{1}{2r}D_A \bar{m}_0^A + \mathcal{O}(r^{-2}), \tag{C.2a}$$

$$\beta = \frac{1}{2}(\gamma_{123} - \gamma_{343}) = \frac{1}{2}\left(\frac{V}{2r}\Gamma_{rA}^u - \Gamma_{rA}^r - \hat{\bar{m}}^B \nabla_A \hat{m}_B\right)\hat{m}^A = -\frac{1}{2r}D_A m_0^A + \mathcal{O}(r^{-2}), \tag{C.2b}$$

$$\gamma = \frac{1}{2}(\gamma_{122} - \gamma_{342}) = \frac{1}{4r^2}e^{-2\beta}(r\partial_r V - V) - \frac{1}{2}n^\mu \hat{\bar{m}}^A \nabla_\mu \hat{m}_A = \frac{\Lambda}{6}r + \gamma_0 + \frac{\gamma_1}{r} + \mathcal{O}(r^{-2}), \tag{C.2c}$$

$$\epsilon = \frac{1}{2}(\gamma_{121} - \gamma_{341}) = \frac{1}{8}\partial_r \gamma_{AB}(\hat{\bar{m}}^A \hat{\bar{m}}^B - \hat{m}^A \hat{m}^B) - \partial_r \beta$$

$$= \frac{1}{8r^2}C_{AB}(\bar{m}_0^A \bar{m}_0^B - m_0^A m_0^B) + \mathcal{O}(r^{-3}), \tag{C.2d}$$

$$\pi = \gamma_{421} = -\left(\frac{1}{2}e^{-2\beta}\gamma_{AB}\partial_r U^B + \partial_A \beta\right)\hat{\bar{m}}^A = \frac{1}{r^2}\left(e^{-2\beta_0}U_A^2 + C_{AB}\partial^B \beta_0\right)\bar{m}_0^A + \mathcal{O}(r^{-3}), \tag{C.2e}$$

$$\nu = \gamma_{422} = \left(e^{-2\beta}\frac{V}{4r}\left(2\partial_A \beta - e^{-2\beta}\gamma_{AB}\partial_r U^B\right) - \Gamma_{\mu A}^r n^\mu\right)\hat{\bar{m}}^A$$

$$= \frac{\Lambda}{3}r\partial_A \beta_0 \bar{m}_0^A + \left(\frac{1}{2}e^{-2\beta_0}\partial_A V_{+2}\bar{m}_0^A + \frac{\Lambda}{3}\partial_A \beta_0 \bar{m}_1^A\right) + \mathcal{O}(r^{-1}), \tag{C.2f}$$

$$\mu = \gamma_{423} = -\frac{1}{2}e^{-2\beta}\left(\partial_u \ln\sqrt{\gamma} + \frac{V}{2r}\partial_r \ln\sqrt{\gamma} + D_A U^A\right) = -\frac{\Lambda}{6}r - \frac{\Lambda}{24}C + \frac{\mu_1}{r} + \mathcal{O}(r^{-2}), \tag{C.2g}$$

$$\lambda = \gamma_{424} = -\frac{1}{2}e^{-2\beta}\left(\left(\partial_u + \frac{V}{2r}\partial_r\right)\gamma_{AB} + 2D_A \tilde{U}_B\right)\hat{\bar{m}}^A \hat{\bar{m}}^B \tag{C.2h}$$

$$= -\frac{1}{2}\left(\frac{\Lambda}{6}C_{AB}^{\mathrm{TF}} - e^{-2\beta_0}\mathfrak{B}_{AB}^\Lambda\right)\bar{m}_0^A \bar{m}_0^B + \mathcal{O}(r^{-1}), \tag{C.2i}$$

$$\kappa = \gamma_{131} = 0, \tag{C.2j}$$

$$\tau = \gamma_{132} = \left(\partial_A \beta - \frac{1}{2}e^{-2\beta}\gamma_{AB}\partial_r U^B\right)\hat{m}^A = \frac{2}{r}\partial_A \beta_0 m_0^A + \mathcal{O}(r^{-2}), \tag{C.2k}$$

$$\sigma = \gamma_{133} = \frac{1}{2}\partial_r \gamma_{AB}\hat{m}^A \hat{m}^B = -\frac{1}{2r^2}C_{AB}m_0^A m_0^B + \mathcal{O}(r^{-3}), \tag{C.2l}$$

$$\rho = \gamma_{134} = \frac{1}{2}\partial_r \ln\sqrt{\gamma} = \frac{1}{r} - \frac{C}{4r^2} + \mathcal{O}(r^{-3}), \tag{C.2m}$$

where

$$\gamma_0 = \frac{1}{4}e^{-2\beta_0}\Big(V_{+2} + m_0^A\partial_A(U_B^0\bar{m}_0^B) - \bar{m}_0^A\partial_A(U_B^0 m_0^B) - \bar{m}_B^0 U_0^A D_A m_0^B\Big)$$
$$+ \frac{1}{8}e^{-2\beta_0}(\bar{m}_0^A\bar{m}_0^B - m_0^A m_0^B)\left(\mathfrak{B}_{AB}^\Lambda + 2D_A U_B^0 - \frac{\Lambda}{6}e^{2\beta_0}C_{AB}^{\mathrm{TF}}\right), \tag{C.3a}$$

$$\gamma_1\big|_{U_0=0} = \frac{\Lambda}{96}(4D - [CC]) + \frac{\Lambda}{24}\left(\frac{1}{4}CC_{AB}^{\mathrm{TF}} - \mathfrak{L}_{AB}\right)(m_0^A m_0^B - \bar{m}_0^A\bar{m}_0^B)$$
$$+ \frac{1}{8}e^{-2\beta_0}(\partial_u q_{AB} - \mathfrak{B}_{AB}^\Lambda)(m_0^A m_1^B - \bar{m}_0^A\bar{m}_1^B) + (D_A\bar{m}_0^A m_0^B - D_A m_0^A\bar{m}_0^B)\partial_B\beta_0$$
$$+ \frac{1}{8}e^{-2\beta_0}\left(N_{AB} - \frac{1}{2}C\mathfrak{B}_{AB}^\Lambda\right)(m_0^A m_0^B - \bar{m}_0^A\bar{m}_0^B), \tag{C.3b}$$

$$\mu_1 = \frac{\Lambda}{96}(2C^2 + 4D - 3[CC])$$
$$- \frac{1}{8}e^{-2\beta_0}\Big((2\partial_u + \partial_u\ln\sqrt{q} + D_A U_0^A)C + 4(V_{+1} + D_A U_1^A + \Gamma_{1AB}^A U_0^B)\Big). \tag{C.3c}$$

The frames (2.54) used in the NP formalism satisfy important identities. First we have the time derivative

$$\partial_u m_0^A = \left(\frac{1}{4}\partial_u q_{AB}(\bar{m}_0^A\bar{m}_0^B - m_0^A m_0^B) - \frac{1}{2}\partial_u\ln\sqrt{q}\right)m_0^A - \frac{1}{2}(\partial_u q_{BC}m_0^B m_0^C)\bar{m}_0^A, \tag{C.4}$$

which can in turn be used in (2.54c) to compute the time evolution $\partial_u m_1^A$. Then, we have other identities involving the time evolution, which are

$$\bar{m}_0^A\bar{m}_0^B(\partial_u q_{AC})q^{CD}(\partial_u q_{DB}) = \frac{2}{3}\bar{m}_0^A\bar{m}_0^B(\Lambda C_{AB} - 3\mathfrak{B}_{AB}^\Lambda)\partial_u\ln\sqrt{q}, \tag{C.5a}$$

$$\bar{m}_0^A\bar{m}_0^B(\partial_u^2 q_{AB}) = \bar{m}_0^A\bar{m}_0^B\left(\frac{\Lambda}{3}(\partial_u + \partial_u\ln\sqrt{q})C_{AB} - (\partial_u + \partial_u\ln\sqrt{q})\mathfrak{B}_{AB}^\Lambda\right). \tag{C.5b}$$

Finally we have the identities

$$C_{AB}m_0^A m_0^B = C_{AB}^{\mathrm{TF}}m_0^A m_0^B = iq_{AC}\epsilon^{CD}C_{DB}^{\mathrm{TF}}m_0^A m_0^B, \tag{C.6a}$$
$$m_0^B D_B m_0^A = m_0^A D_B m_0^B, \tag{C.6b}$$
$$m_0^B D_B\bar{m}_0^A = -\bar{m}_0^A D_B m_0^B, \tag{C.6c}$$
$$(m_0^A\partial_A + 2\beta_1)(V_A m_0^A) = (D_A V_B)m_0^A m_0^B, \tag{C.6d}$$
$$(m_0^A\partial_A - 4\beta_1)(T_{AB}\bar{m}_0^A\bar{m}_0^B) = (D^B T_{AB}^{\mathrm{TF}})\bar{m}_0^A, \tag{C.6e}$$
$$2\sigma_2 V_A\bar{m}_0^A = -C_{AB}^{\mathrm{TF}}V^A m_0^B, \tag{C.6f}$$
$$(V_A\bar{m}_0^A)(T_{AB}m_0^A m_0^B) = T_{AB}^{\mathrm{TF}}V^A m_0^B, \tag{C.6g}$$
$$2\,\mathrm{Re}\big((m_0^A\partial_A - 2\beta_1)(V_A\bar{m}_0^A)\big) = D_A V^A, \tag{C.6h}$$
$$4\,\mathrm{Re}(\sigma_2 T_{AB}\bar{m}_0^A\bar{m}_0^B) = -C_{AB}^{\mathrm{TF}}T^{AB} = -C_{AB}T_{\mathrm{TF}}^{AB}, \tag{C.6i}$$

which hold for any vector $V_A$ and any symmetric tensor $T_{AB}$, and which can easily be complex-conjugated to obtain mirror relations.

To compute the above expansions of the spin coefficients we have also used the expansion of the normal $n^\mu = (n^u, n^r, n^A)$, which is

$$n^u = e^{-2\beta_0} + \mathcal{O}(r^{-2}), \tag{C.7a}$$

$$n^r = r^2 \frac{\Lambda}{6} + \frac{r}{2} e^{-2\beta_0} V_{+2} + \frac{1}{2} e^{-2\beta_0} V_{+1} + \frac{\Lambda}{96} (4D - [CC])$$
$$+ \frac{1}{2r} e^{-2\beta_0} \left( V_0 + \frac{1}{96} ([CC] - 4D)(6\partial_u \ln \sqrt{q} + 6D_A U_0^A - e^{2\beta_0} \Lambda C) \right) + \mathcal{O}(r^{-2}), \tag{C.7b}$$

$$n^A = e^{-2\beta_0} U_0^A + \frac{2}{r} \partial^A \beta_0 + \mathcal{O}(r^{-2}). \tag{C.7c}$$

## D  Expansion of the inverse angular metric

Here we explain how to obtain analytically the asymptotic expansion of the determinant and the inverse of the angular metric. We initially assume that there is no logarithmic term in the expansion of the angular metric, but the construction can straightforwardly be applied to take such terms into account.

Let us denote all the matrices appearing in (2.3) by $q \equiv q_{AB}$, $C \equiv C_{AB}$, and so on. We also denote by $q^{-1} \equiv q^{AB}$ the inverse to $q$, by $C^* \equiv C^{AB} = q^{-1} C q^{-1}$ the tensor with indices raised by $q_{AB}$, and by a dot the matrix product $C^*.D^* = C^* q D^*$ in $q$. We then use the fact that, for arbitrary matrices $A$ and $B$ such that $A$ and $A + B$ are non-singular, we have

$$(A + B)^{-1} = A^{-1} \left( \sum_{n=0}^{\infty} (-BA^{-1})^n \right). \tag{D.1}$$

Applying this formula to $\gamma$ (and discarding once again for simplicity the logarithmic terms) with $A = r^2 q$ and $B = rC + D + \mathcal{O}(r^{-1})$ enables to extract the coefficients in

$$\gamma^{-1} = \sum_{n=2}^{\infty} \frac{\gamma_n^*}{r^n}. \tag{D.2}$$

In particular, this gives

$$\gamma_4^* = -D^* + C^*.C^*, \tag{D.3a}$$

$$\gamma_5^* = -E^* + C^*.D^* + D^*.C^* - C^*.C^*.C^*, \tag{D.3b}$$

$$\gamma_6^* = -F^* + D^*.D^* + C^*.E^* + E^*.C^* - C^*.C^*.D^* - C^*.D^*.C^* - D^*.C^*.C^* + C^*.C^*.C^*.C^* . \tag{D.3c}$$

This can then be rewritten using the fact that for $2 \times 2$ matrices we have

$$AB + BA = (\mathrm{tr}A)B + (\mathrm{tr}B)A - \det(A + B)\mathbb{I} + (\det A)\mathbb{I} + (\det B)\mathbb{I}, \tag{D.4a}$$

$$\det(A + B) = \det A + \det B + (\det A)\mathrm{tr}(A^{-1}B), \tag{D.4b}$$

where $\mathrm{tr}(\cdot)$ is the usual matrix trace. Introducing now the trace with respect to $q$ defined as $\mathrm{tr}_q(C) \equiv \mathrm{tr}(q^{-1}C) = q^{AB}C_{AB}$ and using the notation $[CC] \equiv \mathrm{tr}(C^*C) = C^{AB}C_{AB}$ we find

$$\det(q^{-1}C) = \frac{1}{2} (\mathrm{tr}_q(C)^2 - [CC]), \tag{D.5a}$$

$$\det(q^{-1}C)\mathrm{tr}(C^{-1}D) = \det(q^{-1}D)\mathrm{tr}(D^{-1}C) = [CD] - \mathrm{tr}_q(C)\mathrm{tr}_q(D), \tag{D.5b}$$

which we can use to obtain

$$C^*.C^* = \frac{1}{2}\big([CC] - \mathrm{tr}_q(C)^2\big)q^{-1} + \mathrm{tr}_q(C)C^*, \tag{D.6a}$$

$$C^*.D^* + D^*.C^* = \big([CD] - \mathrm{tr}_q(C)\mathrm{tr}_q(D)\big)q^{-1} + \mathrm{tr}_q(D)C^* + \mathrm{tr}_q(C)D^*, \tag{D.6b}$$

$$C^*.C^*.C^* = \frac{1}{2}\Big(\mathrm{tr}_q(C)\big([CC] - \mathrm{tr}_q(C)^2\big)q^{-1} + \big([CC] + \mathrm{tr}_q(C)^2\big)C^*\Big). \tag{D.6c}$$

Combining these formulas and switching back to the notation of the main text finally yields

$$\gamma_4^{AB} = -D^{AB} + CC_{\mathrm{TF}}^{AB} + \frac{1}{2}[CC]q^{AB}, \tag{D.7a}$$

$$\gamma_5^{AB} = -E^{AB} + CD_{\mathrm{TF}}^{AB} + DC_{\mathrm{TF}}^{AB} - \frac{1}{2}\big([CC] + C^2\big)C^{AB} + \frac{1}{2}\big(C^3 - C[CC] + 2[CD]\big)q^{AB}, \tag{D.7b}$$

$$\gamma_6^{AB} = -F^{AB} + CE_{\mathrm{TF}}^{AB} + EC_{\mathrm{TF}}^{AB} + DD_{\mathrm{TF}}^{AB} - \frac{1}{2}\big([CC] + C^2\big)D^{AB} + \big(C\big([CC] - D\big) - [CD]\big)C^{AB}$$

$$+ \frac{1}{4}\Big([CC]\big([CC] - 2D\big) + 4[CE] + 2[DD] - 4C[CD] + C^2\big(6D - C^2\big)\Big)q^{AB}. \tag{D.7c}$$

The subleading components $\gamma_4^{AB}$ and $\gamma_5^{AB}$ are unfortunately necessary because they appear in the symplectic potential and when solving respectively for $\beta_2$ and $\beta_3$ in section 2.2.

To obtain the determinant, we proceed with the same notations as above and use formula (D.4b) to expand

$$\det(\gamma) = r^4 \det(q) + r^2 \det(C) + \det(D) + \det\left(\frac{E}{r} + \mathcal{O}(r^{-2})\right)$$

$$+ \det(q)\,\mathrm{tr}\left[q^{-1}\left(r^3 C + r^2 D + rE + D + \frac{F}{r} + \mathcal{O}(r^{-2})\right)\right]$$

$$+ \det(C)\mathrm{tr}\left[C^{-1}\left(rD + E + \frac{D}{r} + \mathcal{O}(r^{-2})\right)\right]$$

$$+ \det(D)\mathrm{tr}\left[D^{-1}\left(\frac{E}{r} + \mathcal{O}(r^{-2})\right)\right]. \tag{D.8}$$

To go back to the notations of the main text we then note that

$$\frac{\det(C)}{\det(q)} = \frac{1}{2}\big(C^2 - [CC]\big), \qquad \frac{\det(C)}{\det(q)}\mathrm{tr}(C^{-1}D) = CD - [CD], \tag{D.9}$$

and similarly with the other tensors appearing in the angular metric. Combining all these results, we can for example extract the next order in the partial Bondi gauge solution (2.7), which is used in appendix F.

At the end of the day, we can also reintroduce the logarithmic terms in order to expand the determinant and the inverse of the polyhomogeneous metric (2.3). In this case we find that the

angular metric and the volume element have an expansion of the form

$$\gamma^{AB} = \frac{q^{AB}}{r^2} - \frac{C^{AB}}{r^3} + \frac{\gamma_4^{AB}}{r^4} + \frac{1}{r^5}\left(\gamma_5^{AB} - E_l^{AB}\ln r\right)$$
$$+ \frac{1}{r^6}\left(\gamma_6^{AB} + \left([CE_l]q^{AB} - CE_l q^{AB} + E_l C^{AB} + CE_l^{AB} - F_l^{AB}\right)\ln r - F_{l^2}^{AB}(\ln r)^2\right)$$
$$+ \mathcal{O}(r^{-7}), \tag{D.10a}$$

$$\frac{\sqrt{\gamma}}{\sqrt{q}} = r^2 + \frac{C}{2}r + \frac{1}{4}\left(2D - [CC^{\mathrm{TF}}]\right) + \frac{1}{2r}\left(E + \frac{1}{4}C[CC^{\mathrm{TF}}] - [CD^{\mathrm{TF}}] + E_l\ln r\right)$$
$$+ \frac{1}{4r^2}\left(\frac{1}{32}\left(3C^4 - 4[CC]^2 - 4C^2[CC]\right) - \frac{3}{4}C^2 D + \frac{1}{2}D[CC] + \frac{1}{2}D^2 + C[CD] - [DD]\right.$$
$$\left. + CE - 2[CE] + 2F + \left(CE_l - 2[CE_l] + 2F_l\right)\ln r + 2F_{l^2}(\ln r)^2\right) + \mathcal{O}(r^{-3}). \tag{D.10b}$$

All of these expressions can of course be obtained with a computer, although there can be a lot of guess work involved as more and more combinations of the tensors appear as one goes to lower order in the radial expansion.

## E Expansion of the angular connection

In this appendix we assume once again for simplicity and clarity that there are no logarithmic terms in the expansion of the angular metric. The Levi–Civita connection associated with the angular metric $\gamma_{AB}$ then admits an expansion of the form

$$\Gamma_{AB}^C[\gamma] = \Gamma_{AB}^C[q] + \frac{\Gamma_{1AB}^C}{r} + \frac{\Gamma_{2AB}^C}{r^2} + \frac{\Gamma_{4AB}^C}{r^3} + \mathcal{O}(r^{-4}), \tag{E.1}$$

with

$$\Gamma_{1AB}^C = \frac{1}{2}\left(D_A C_B^C + D_B C_A^C - D^C C_{AB}\right), \tag{E.2}$$

$$\Gamma_{2AB}^C = \frac{1}{2}q^{CD}\left(\partial_A D_{DB} + \partial_B D_{AD} - \partial_D D_{AB}\right) - \frac{1}{2}C^{CD}\left(\partial_A C_{DB} + \partial_B C_{AD} - \partial_D C_{AB}\right)$$
$$+ \gamma_{MN}^4 q^{MC}\Gamma_{AB}^N[q], \tag{E.3}$$

$$\Gamma_{3AB}^C = \frac{1}{2}q^{CD}\left(\partial_A E_{DB} + \partial_B E_{AD} - \partial_D E_{AB}\right) - \frac{1}{2}C^{CD}\left(\partial_A D_{DB} + \partial_B D_{AD} - \partial_D D_{AB}\right)$$
$$+ \frac{1}{2}\gamma_5^{CD}\left(\partial_A C_{DB} + \partial_B C_{AD} - \partial_D C_{AB}\right) + \gamma_{MN}^5 q^{MC}\Gamma_{AB}^N[q], \tag{E.4}$$

with $\gamma_{AB}^4$ and $\gamma_{AB}^5$ given above in (D.7). When the determinant condition is satisfied we have at all orders the properties

$$\Gamma_{AB}^B[\gamma] \overset{\mathrm{BS}}{=} \Gamma_{AB}^B[q] = \partial_A \ln\sqrt{q}, \qquad \mathcal{D}_A P^A \overset{\mathrm{BS}}{=} D_A P^A, \tag{E.5}$$

or in terms of the above expansion $\Gamma_{nAB}^B \overset{\mathrm{BS}}{=} 0$, $\forall n$.

We now give a few identities involving combinations of covariant derivatives. In the computation of (H.11) we use identities of the form

$$D_{(A}\bar{P}_{B)} - C_{(AC}D_{B)}P^C - 2\Gamma_{1AB}^C P_C = P^C D_C C_{AB}, \tag{E.6a}$$

$$D_A\bar{P}^A - C^{AB}D_A P_B - \Gamma_1^A P_A = \Gamma_{1AB}^B P^A = \frac{1}{2}P^A\partial_A C, \tag{E.6b}$$

which hold for any vector $P^A$ with $\bar{P}^A = C^{AB}P_B$, and where we have denoted $q^{AB}\Gamma_{1AB}^C = \Gamma_1^C$. With the expansion (2.9) we have

$$\mathcal{D}_A U^A = D_A U_0^A + \frac{1}{r}\left(D_A U_1^A + \Gamma_{1AB}^A U_0^B\right) + \mathcal{O}(r^{-2}). \tag{E.7}$$

For any symmetric trace-free tensor $T_{AB}$ we have

$$D^C D_C T_{AB} - \left(D_{(A}D^C T_{B)C}\right)^{\mathrm{TF}} = R[q]T_{AB}. \tag{E.8}$$

Finally we note that

$$\left(\delta + \delta\ln\sqrt{q}\right)R[q] + 2D^A D_A\delta\ln\sqrt{q} = -D_A D_B\delta q^{AB}, \tag{E.9}$$

which also holds when replacing e.g. $\delta \to \partial_u$ or $\delta \to \partial_A$.

# F    Subleading solution space

Here we report part of the structure of the solution space at subleading order. This is necessary in order to compute the evolution equations in the case $\Lambda \neq 0$ and also to understand how the logarithmic branches disappear in the case $\Lambda \neq 0$. Since we compute the evolution equations in section 2.4 with the boundary sources turned off, i.e. $\beta_0 = 0 = U_0^A$, we also do so in this appendix.

The solution for $\beta$ is actually independent of the boundary sources $\beta_0$ and $U_0^A$. Going further in the expansion of (2.6), we find terms in $r^{-4}\left(\beta_{l^2 4}(\ln r)^2 + \beta_{l4}\ln r + \beta_4\right)$, with

$$\beta_{l^2 4} = -\frac{3}{8}F_{l^2}, \tag{F.1a}$$

$$\beta_{l4} = \frac{3}{32}CE_l + \frac{9}{32}[CE_l^{\mathrm{TF}}] - \frac{3}{8}F_l + \frac{1}{4}F_{l^2}, \tag{F.1b}$$

$$32\,\beta_4 = -12F + 4F_l + 9[EC^{\mathrm{TF}}] + 3CE - CE_l - \frac{15}{4}[E_lC^{\mathrm{TF}}] + 4[\mathfrak{L}D] + D^2$$
$$+ \frac{1}{8}D\left(3C^2 - 22[CC]\right) - 5C[\mathfrak{L}C] - \frac{3}{32}C^2[CC] + \frac{5}{8}[CC]^2. \tag{F.1c}$$

The expansion (2.9) then continues with terms in $r^{-4}\left(U_{l^2 4}^A(\ln r)^2 + U_{l4}^A\ln r + U_4^A\right)$, with

$$U_{l^2 4}^A = 0, \tag{F.2a}$$

$$U_{l4}^A = -\frac{3}{4}\left(\frac{1}{2}CU_{l3}^A + C_B^A U_{l3}^B - D_B(E_l^{\mathrm{TF}})^{AB} - \frac{1}{3}\partial^A E_l\right), \tag{F.2b}$$

$$U_4^A = (\text{lengthy}). \tag{F.2c}$$

Finally, we find that the expansion (2.13) continues with $r^{-1}\big(V_{l^2 1}(\ln r)^2 + V_{l1}\ln r + V_1\big)$, where

$$V_{l^2 1} = \frac{5\Lambda}{12}F_{l^2}, \tag{F.3a}$$

$$V_{l1} = \frac{\Lambda}{12}\left(\frac{5}{4}CE_l - 2F_{l^2} + 5F_l\right) + \frac{1}{2}D_A U_{l3}^A - \frac{1}{4}CV_{l0} - \frac{1}{2}\partial_u E_l - \frac{11}{8}[E_l\mathcal{B}^\Lambda] - \frac{3}{4}E_l\partial_u \ln\sqrt{q}, \tag{F.3b}$$

$$V_1 = \text{(lengthy)}. \tag{F.3c}$$

The expressions for $U_4^A$ and $V_1$ have been used for the computation of the NP (A)dS mass loss equation, but are too lengthy to be displayed here.

We now give the real parts of the subleading terms in the Weyl scalar $\Psi_4$ in the absence of boundary sources. Off-shell of the $(AB)$ Einstein equations we have

$$\mathrm{Re}\big(\Psi_2^{l4}\big) = -\frac{3}{4}D_A U_{l3}^A + \frac{1}{4}[E_l\mathcal{B}^\Lambda] - \frac{\Lambda}{8}\left([E_l C] + \frac{1}{2}[C\mathfrak{L}]\right), \tag{F.4a}$$

$$\mathrm{Re}\big(\Psi_2^4\big) = -\frac{1}{2}D^A\mathcal{P}_A - \frac{3}{4}C\mathcal{M} + \frac{\Lambda}{24}C_{AB}\mathcal{E}^{AB}$$
$$+ \frac{\Lambda}{48}\left(5[CE_l^{\mathrm{TF}}] + [C\mathfrak{L}]\right) - \frac{1}{2}D_A D_B\mathfrak{L}^{AB} + \frac{1}{4}[N\mathfrak{L}] - \frac{1}{8}[C\mathfrak{L}]\partial_u \ln\sqrt{q}. \tag{F.4b}$$

On-shell of the constraints and in the case (A)dS we therefore get $\mathrm{Re}\big(\Psi_2^{l4}\big) = 0$ and (2.74), which is what we have used in the main text.

## G  Alternative resolution of $\mathbf{E}_{ur} = 0$

The field equation $\mathbf{E}_{ur} = 0$ can rewritten using its trace to obtain the equivalent form $R_{ur} = \Lambda g_{ur}$. At the difference with $\gamma^{AB}R_{AB} = 2\Lambda$, which is first order, this equation is second order in $V$ since we have

$$R_{ur} = \big(\partial_r - \partial_r \ln\sqrt{\gamma}\big)\Gamma_{ur}^r + \partial_A\Gamma_{ur}^A + \partial_u\partial_r(\ln\sqrt{\gamma} + 2\beta)$$
$$+ \frac{1}{4}(\partial_u\gamma_{AB})(\partial_r\gamma^{AB}) + \frac{1}{2}\gamma^{AB}\Gamma_{AC}^C[\gamma]\big(\partial_B e^{2\beta} - \partial_r\tilde{U}_B\big), \tag{G.1}$$

which contains $\partial_r\Gamma_{ur}^r$. The expansion which solves this equation is of the form

$$V = r^3 V_{+3} + r^2 V_{+2} + r\big(V_{+l1}\ln r + V_{+1}\big) + V_{l0}\ln r + 2M + \mathcal{O}(r^{-1}), \tag{G.2}$$

which differs from (2.13) by the presence of an additional $r\ln r$ term. The rewritten equation sets

$$R_{ur} - \Lambda g_{ur} = \mathcal{O}(r^{-1}) \quad \Rightarrow \quad V_{+3} = \frac{\Lambda}{3}e^{2\beta_0}, \tag{G.3a}$$

$$R_{ur} - \Lambda g_{ur} = \mathcal{O}(r^{-2}) \quad \Rightarrow \quad V_{+2} = \frac{\Lambda}{6}e^{2\beta_0}C - \partial_u\ln\sqrt{q} - D_A U_0^A, \tag{G.3b}$$

$$R_{ur} - \Lambda g_{ur} = \mathcal{O}(r^{-3}) \quad \Rightarrow \quad V_{+l1} = -\frac{1}{2}[C\mathcal{B}^\Lambda], \tag{G.3c}$$

$$R_{ur} - \Lambda g_{ur} = \mathcal{O}(r^{-4}) \quad \Rightarrow \quad V_{l0} = [D\mathcal{B}^\Lambda] - \frac{1}{4}C[C\mathcal{B}^\Lambda] + \frac{\Lambda}{3}e^{2\beta_0}E_l, \tag{G.3d}$$

where the logarithmic terms involve $\mathfrak{B}_{AB}^{\Lambda}$ defined in (2.18). As expected, we see that with this method of resolution the logarithmic terms involve the constraints whose vanishing is implied by the $(AB)$ Einstein equations. The construction of the solution space is therefore consistent if we follow the route presented in this appendix or that of section 2.2.

## H Computation of $\Theta^r$

Here we present the details of the calculation of the radial part $\Theta^r$ of the symplectic potential in the partial Bondi gauge, assuming there are no logarithmic terms in the angular metric and the solution space. We first rewrite the radial component of the potential in the form

$$\Theta^r = \sqrt{\gamma}\left(\delta\big(\Gamma_{uu}^u + \Gamma_{uA}^A - \Gamma_{ur}^r\big) + U^A\delta\big(\Gamma_{Au}^u + \Gamma_{AB}^B - \Gamma_{Ar}^r\big) + \frac{V}{r}\delta\Gamma_{rA}^A + e^{2\beta}\gamma^{AB}\delta\Gamma_{AB}^r\right)$$
$$=: \sqrt{\gamma}\big(\delta\mathbb{A} + U^A\delta\mathbb{B}_A + \mathbb{C} + \mathbb{D}\big). \tag{H.1}$$

We then study each term separately using the off-shell expansion of the connection components (B.1). The only simplification which we use is to set $\beta_1 = 0$ since at the end of the day this on-shell condition is common to both the BS and NU gauges. We also assume that the logarithmic branches have been removed so that e.g. $U_{l3}^A$ does not appear in the expansion of $U^A$.

The first term in the above rewriting is

$$\mathbb{A} = \Gamma_{uu}^u + \Gamma_{uA}^A - \Gamma_{ur}^r = r\mathbb{A}_{+1} + \mathbb{A}_0 + \frac{\mathbb{A}_1}{r} + \frac{\mathbb{A}_2}{r^2} + \mathcal{O}(r^{-3}), \tag{H.2}$$

with

$$\mathbb{A}_{+1} = 2V_{+3}, \tag{H.3a}$$

$$\mathbb{A}_0 = -e^{-2\beta_0}[U_0U_1] + 2[U_0\partial\beta_0] + V_{+2} + \partial_u(\ln\sqrt{q} + 2\beta_0), \tag{H.3b}$$

$$\mathbb{A}_1 = -e^{-2\beta_0}\big([U_1U_1] + 2[U_0U_2] + [\bar{U}_0U_1]\big) + 2[U_1\partial\beta_0] - 4\beta_2V_{+3} + \frac{1}{2}\partial_uC, \tag{H.3c}$$

$$\mathbb{A}_2 = -e^{-2\beta_0}\Big(3\big([U_0U_3] + [U_1U_2]\big) + [\bar{U}_1U_1] + 2[\bar{U}_0U_2] + [\bar{\bar{U}}_0U_1] - 2\beta_2[U_0U_1]\Big)$$
$$+ 2\big([U_0\partial\beta_2] + [U_2\partial\beta_0]\big) - 2M - 4\beta_2V_{+2} - 6\beta_3V_{+3} + \frac{1}{4}\partial_u\big(2D - [CC] + 8\beta_2\big). \tag{H.3d}$$

The second term is

$$\mathbb{B}_A = \Gamma_{Au}^u + \Gamma_{AB}^B - \Gamma_{Ar}^r = \mathbb{B}_{0A} + \frac{\mathbb{B}_{1A}}{r} + \frac{\mathbb{B}_{2A}}{r^2} + \mathcal{O}(r^{-3}), \tag{H.4}$$

with

$$\mathbb{B}_{0A} = e^{-2\beta_0}U_A^1 + \partial_A\ln\sqrt{q}, \tag{H.5a}$$

$$\mathbb{B}_{1A} = e^{-2\beta_0}\big(2U_A^2 + \bar{U}_A^1\big) + \frac{1}{2}\partial_AC, \tag{H.5b}$$

$$\mathbb{B}_{2A} = e^{-2\beta_0}\big(3U_A^3 + 2\bar{U}_A^2 + \bar{\bar{U}}_A^1 - 2\beta_2U_A^1\big) + \frac{1}{4}\partial_A\big(2D - [CC]\big). \tag{H.5c}$$

The third term is

$$\mathbb{C} = \frac{V}{r}\delta\Gamma^A_{rA} = \mathbb{C}_0 + \frac{\mathbb{C}_1}{r} + \frac{\mathbb{C}_2}{r^2} + \mathcal{O}(r^{-3}), \tag{H.6}$$

with

$$2\mathbb{C}_0 = -V_{+3}\delta C, \tag{H.7}$$

$$2\mathbb{C}_1 = -V_{+2}\delta C + V_{+3}\delta([CC] - 2D), \tag{H.8}$$

$$2\mathbb{C}_2 = -V_{+1}\delta C + V_{+2}\delta([CC] - 2D) + \frac{1}{2}V_{+3}\delta(C^3 - 6E - 3C[CC] + 6[CD]). \tag{H.9}$$

Finally, the fourth and most intricate term is

$$\mathbb{D} = e^{2\beta}\gamma^{AB}\delta\Gamma^r_{AB} = r\mathbb{D}_{+1} + \mathbb{D}_0 + \frac{\mathbb{D}_1}{r} + \frac{\mathbb{D}_2}{r^2} + \mathcal{O}(r^{-3}), \tag{H.10}$$

with

$$\mathbb{D}_{+1} = q^{AB}\delta(q_{AB}V_{+3}) - 4\delta\beta_0 V_{+3}, \tag{H.11a}$$

$$\begin{aligned}
\mathbb{D}_0 = {} & q^{AB}\delta\left\{q_{AB}V_{+2} - \frac{1}{2}C_{AB}V_{+3} + \frac{1}{2}\partial_u q_{AB} + D_A U^0_B\right\} \\
& + \delta\beta_0(CV_{+3} - 4V_{+2} - \partial_u \ln q - 2D_A U^A_0) + V_{+3}\delta C,
\end{aligned} \tag{H.11b}$$

$$\begin{aligned}
\mathbb{D}_1 = {} & q^{AB}\delta\left\{q_{AB}V_{+1} - \frac{1}{2}C_{AB}V_{+2} + \frac{1}{2}\partial_u C_{AB} + D_A U^1_B + C_{AC}D_B U^C_0 + \frac{1}{2}U^C_0 D_C C_{AB}\right\} \\
& - C^{AB}\delta\left\{\frac{1}{2}\partial_u q_{AB} + D_A U^0_B\right\} \\
& - \delta\beta_0\left(4V_{+1} - CV_{+2} + ([CC] - 2D)V_{+3} + \partial_u C + 2D_A U^A_1 + U^A_0\partial_A C\right) \\
& + V_{+2}\delta C + \frac{1}{2}\delta V_{+3}([CC] - 2D) + V_{+3}\left(\gamma^{AB}_4\delta q_{AB} - \frac{1}{2}C^{AB}\delta C_{AB} - 4\delta\beta_2\right),
\end{aligned} \tag{H.11c}$$

$$\begin{aligned}
\mathbb{D}_2 = {} & q^{AB}\delta\left\{2M q_{AB} + \frac{1}{2}C_{AB}V_{+1} + \frac{1}{2}\partial_u D_{AB} - \frac{1}{2}E_{AB}V_{+3}\right\} \\
& + q^{AB}\delta\left\{D_A(U^2_B + \bar{U}^1_B + \bar{\bar{U}}^0_B) - \Gamma^C_{1AB}(U^1_C + \bar{U}^0_C) - \Gamma^C_{2AB}U^0_C\right\} \\
& - C^{AB}\delta\left\{q_{AB}V_{+1} + \frac{1}{2}C_{AB}V_{+2} + \frac{1}{2}\partial_u C_{AB} + D_A(U^1_B + \bar{U}^0_B) - \Gamma^C_{1AB}U^0_C\right\} \\
& + \gamma^{AB}_4\delta\left\{q_{AB}V_{+2} + \frac{1}{2}C_{AB}V_{+3} + \frac{1}{2}\partial_u q_{AB} + D_A U^0_B\right\} + \gamma^{AB}_5\delta(q_{AB}V_{+3}) \\
& - \delta\beta_0\left(8M - CV_{+1} - 2DV_{+2} - EV_{+3} + q^{AB}\partial_u D_{AB} + [CC]V_{+2} - C^{AB}\partial_u C_{AB}\right) \\
& - \delta\beta_0\left(\gamma^{AB}_4(C_{AB}V_{+3} + \partial_u q_{AB} + 2D_A U^0_B) + 2\gamma^{AB}_5 q_{AB}V_{+3}\right) \\
& - 2\delta\beta_0\left(D_A(U^A_2 + \bar{U}^A_0 - C^{AB}\bar{U}^0_B) + \frac{1}{2}(U^A_1 + \bar{U}^A_0)\partial_A C + (C^{AB}\Gamma^C_{1AB} - \Gamma^C_2)U^0_C\right) \\
& + \delta\beta_2(CV_{+3} - 4V_{+2} - \partial_u \ln q) - 4\delta\beta_3 V_{+3}.
\end{aligned} \tag{H.11d}$$

We then collect all these pieces and use the expansion (D.10) of the boundary volume to find

$$\Theta^r = \sqrt{q}\left(r^3\theta^r_3 + r^2\theta^r_2 + r\theta^r_1 + \theta^r_0\right), \tag{H.12}$$

with

$$\theta^r_3 = \delta\mathbb{A}_{+1} + \mathbb{D}_{+1}, \tag{H.13a}$$

$$\theta^r_2 = \delta\mathbb{A}_0 + \mathbb{D}_0 + \mathbb{C}_0 + [U_0\delta\mathbb{B}_0] + \frac{1}{2}C\theta^r_3, \tag{H.13b}$$

$$\theta^r_1 = \delta\mathbb{A}_1 + \mathbb{D}_1 + \mathbb{C}_1 + [U_0\delta\mathbb{B}_1] + [U_1\delta\mathbb{B}_0] + \frac{1}{2}C\theta^r_2 + \frac{1}{8}\left(4D - C^2 - 2[CC]\right)\theta^r_3, \tag{H.13c}$$

$$\theta^r_0 = \delta\mathbb{A}_2 + \mathbb{D}_2 + \mathbb{C}_2 + [U_0\delta\mathbb{B}_2] + [U_1\delta\mathbb{B}_1] + [U_2\delta\mathbb{B}_0] + \frac{1}{2}C\theta^r_1 + \frac{1}{8}\left(4D - C^2 - 2[CC]\right)\theta^r_2$$
$$+ \frac{1}{8}\left(4E + 3C[CC] - 4[CD] - 2CD - \frac{1}{2}C^3\right)\theta^r_3. \tag{H.13d}$$

Here we clearly see in each $\theta^r_n$ the terms involving $\theta^r_{m>n}$ which come from the relaxation of the determinant condition, or in other words the fact that the co-dimension 2 volume has a non-trivial expansion. It is immediate to see that these terms drop in the BS gauge.

These expressions are so far valid off-shell. The only assumptions we have made about the expansion is $\beta_1 = 0$ and the absence of logarithmic terms. It is now convenient to use the on-shell expressions derived in the partial Bondi gauge in section 2.2 in order to separate these components of the potential into sources pieces (S) which vanish when the boundary sources $(\beta_0, U^A_0)$ are set to zero, and fixed pieces (F) which remain even in the absence of boundary sources. For compactness and convenience we will however adopt a mixed notation where we still keep e.g. $U^A_1$ without its on-shell value since it is a pure source term. We also keep $\beta_2$ and $\beta_3$ unspecified in order to later be able to reach the BS or NU gauge, and sometimes keep writing $V^+_n$ instead of their explicit on-shell values.

We now decompose each coefficient in the expansion of the potential as $\theta_n^r = {}^{(\mathrm{F})}\theta_n^r + {}^{(\mathrm{S})}\theta_n^r$, keeping in mind that we are now using the solution space of section 2.2. For the terms which remain in the absence of boundary sources we find

$$
{}^{(\mathrm{F})}\theta_3^r = V_{+3}\,\delta\ln q, \tag{H.14a}
$$

$$
{}^{(\mathrm{F})}\theta_2^r = \left(2CV_{+3} - \partial_u\ln q - \partial_u\right)\delta\ln\sqrt{q} - \frac{1}{2}\delta q^{AB}\partial_u q_{AB} + \frac{1}{2}V_{+3}\left(q_{AB}\delta C^{AB} + 2\delta C\right), \tag{H.14b}
$$

$$
\begin{aligned}
{}^{(\mathrm{F})}\theta_1^r ={}& -\frac{1}{2}q^{AB}\delta\left\{q_{AB}\left[e^{2\beta_0}\left(R[q] + \frac{\Lambda}{2}\left(\frac{3}{4}[CC] - \frac{C^2}{3} - D\right)\right) + \frac{1}{2}(\partial_u\ln\sqrt{q} + 2\partial_u)C\right]\right\} \\
&+ \frac{1}{4}q^{AB}\delta\left\{C_{AB}(\partial_u\ln q - CV_{+3}) + 2\partial_u C_{AB}\right\} - \frac{1}{2}C^{AB}\delta\partial_u q_{AB} \\
&+ \frac{1}{4}(CV_{+3} - \partial_u\ln q + 2\partial_u)\delta C + V_{+3}\left(\gamma_4^{AB}\delta q_{AB} - \frac{1}{2}C^{AB}\delta C_{AB} + \frac{1}{4}\delta[CC]\right) \\
&+ \frac{1}{2}C^{(\mathrm{F})}\theta_2^r + \frac{1}{8}\left(4D - C^2 - 2[CC]\right){}^{(\mathrm{F})}\theta_3^r, \tag{H.14c}
\end{aligned}
$$

$$
\begin{aligned}
{}^{(\mathrm{F})}\theta_0^r ={}& 2(\delta\ln q + \delta)M + \frac{1}{2}q^{AB}\delta\left\{\partial_u D_{AB} - E_{AB}V_{+3} + D_A\left(e^{2\beta_0}\left(\partial_B C - D^C C_{BC}\right)\right)\right\} \\
&+ \frac{1}{4}q^{AB}\delta C_{AB}\left\{e^{2\beta_0}\left(R[q] + \frac{\Lambda}{2}\left(\frac{3}{4}[CC] - \frac{C^2}{3} - D\right)\right) + \frac{1}{2}(\partial_u\ln\sqrt{q} + 2\partial_u)C\right\} \\
&- \frac{1}{4}C^{AB}\delta\left\{C_{AB}(CV_{+3} - \partial_u\ln q) + 2\partial_u C_{AB}\right\} + \frac{1}{2}\left(\delta CV_{+1} - C\delta V_{+1}\right) \\
&+ \frac{1}{2}\gamma_4^{AB}\left(\delta\left\{q_{AB}(CV_{+3} - \partial_u\ln q)\right\} + \delta C_{AB}V_{+3} + \delta\partial_u q_{AB}\right) + \gamma_5^{AB}\delta q_{AB}V_{+3} \\
&+ \frac{1}{4}\partial_u\delta(2D - [CC] + 8\beta_2) + 2\delta\left\{\beta_2(\partial_u\ln q - CV_{+3})\right\} + \delta\beta_2(\partial_u\ln q - CV_{+3}) \\
&+ \frac{1}{4}(CV_{+3} - \partial_u\ln q)\delta([CC] - 2D) + \frac{1}{4}V_{+3}\delta(C^3 - 6E - 3C[CC] + 6[CD]) \\
&+ \frac{1}{2}e^{2\beta_0}\left(\partial^A C - D_B C^{AB}\right)\partial_A\delta\ln\sqrt{q} + \frac{1}{2}C^{(\mathrm{F})}\theta_1^r + \frac{1}{8}\left(4D - C^2 - 2[CC]\right){}^{(\mathrm{F})}\theta_2^r \\
&+ \frac{1}{8}\left(4E + 3C[CC] - 4[CD] - 2CD - \frac{1}{2}C^3\right){}^{(\mathrm{F})}\theta_3^r - 10\delta\beta_3 V_{+3}. \tag{H.14d}
\end{aligned}
$$

For the terms which drop when turning off the sources we find

$$^{(S)}\theta_3^r = 2\delta V_{+3},\tag{H.15a}$$

$$^{(S)}\theta_2^r = 2C\delta V_{+3} - \delta\left\{e^{-2\beta_0}[U_0U_1] - 2[U_0\partial\beta_0] - 2\partial_u\beta_0 + 3D_AU_0^A\right\} + [U_0\delta\mathbb{B}_0]$$
$$+ q^{AB}\delta\left(D_AU_B^0\right) - \left(D_AU_0^A\right)\delta\ln q - \delta\beta_0\left(CV_{+3} - \partial_u\ln q - 2D_AU_0^A\right),\tag{H.15b}$$

$$^{(S)}\theta_1^r = -\frac{1}{2}q^{AB}\delta\left\{q_{AB}\left[e^{2\beta_0}\left(4D_A\partial^A\beta_0 + 8(\partial^A\beta_0)(\partial_A\beta_0)\right) + \frac{1}{2}\left(D_AU_0^A + 2U_0^A\partial_A\right)C\right]\right\}$$
$$+ q^{AB}\delta\left\{D_AU_B^1 + C_{AC}D_BU_0^C + \frac{1}{2}U_0^CD_CC_{AB}\right\} - C^{AB}\delta\left(D_AU_B^0\right)$$
$$+ \frac{1}{8}\delta V_{+3}\left(3[CC] - 4D\right) - \frac{1}{2}\left(D_AU_0^A\right)\delta C + \frac{1}{2}q^{AB}\delta\left(C_{AB}D_CU_0^C\right)$$
$$- \delta\beta_0\left(4V_{+1} - CV_{+2} + \left([CC] - 2D\right)V_{+3} + \partial_uC + 2D_AU_1^A + U_0^A\partial_AC\right)$$
$$- \delta\left\{e^{-2\beta_0}\left([U_1U_1] + 2[U_0U_2] + [\bar{U}_0U_1]\right)\right\} + 2\delta[U_1\partial\beta_0]$$
$$+ [U_0\delta\mathbb{B}_1] + [U_1\delta\mathbb{B}_0] + \frac{1}{2}C^{(S)}\theta_2^r + \frac{1}{8}\left(4D - C^2 - 2[CC]\right)^{(S)}\theta_3^r,\tag{H.15c}$$

$$^{(S)}\theta_0^r = -q^{AB}\delta\left\{D_A\left(e^{2\beta_0}C_{BC}\partial^C\beta_0\right) - D_A\left(\bar{U}_B^1 + \bar{\bar{U}}_B^0\right) + \Gamma_{1AB}^C\left(U_C^1 + \bar{U}_C^0\right) + \Gamma_{2AB}^CU_C^0\right\}$$
$$+ \frac{1}{4}q^{AB}\delta C_{AB}\left\{e^{2\beta_0}\left(4D_A\partial^A\beta_0 + 8(\partial^A\beta_0)(\partial_A\beta_0)\right) + \frac{1}{2}\left(D_AU_0^A + 2U_0^A\partial_A\right)C\right\}$$
$$+ C^{AB}\delta\left\{\frac{1}{2}C_{AB}D_CU_0^C - D_A\left(U_B^1 + \bar{U}_B^0\right) + \Gamma_{1AB}^CU_C^0\right\}$$
$$+ \gamma_4^{AB}\left(\frac{1}{2}C_{AB}\delta V_{+3} - \delta\left(q_{AB}D_CU_0^C\right) + \delta\left(D_AU_B^0\right)\right) + \gamma_5^{AB}q_{AB}\delta V_{+3}$$
$$- \delta\beta_0\left(8M - CV_{+1} - 2DV_{+2} - EV_{+3} + q^{AB}\partial_uD_{AB} + [CC]V_{+2} - C^{AB}\partial_uC_{AB}\right)$$
$$- \delta\beta_0\left(\gamma_4^{AB}\left(C_{AB}V_{+3} + \partial_uq_{AB} + 2D_AU_B^0\right) + 2\gamma_5^{AB}q_{AB}V_{+3}\right)$$
$$- 2\delta\beta_0\left(D_A\left(U_2^A + \bar{\bar{U}}_0^A - C^{AB}\bar{U}_B^0\right) + \frac{1}{2}\left(U_1^A + \bar{U}_0^A\right)\partial_AC + \left(C^{AB}\Gamma_{1AB}^C - \Gamma_2^C\right)U_C^0\right)$$
$$- \delta\left\{e^{-2\beta_0}\left(3\left([U_0U_3] + [U_1U_2]\right) + [\bar{U}_1U_1] + 2[\bar{U}_0U_2] + [\bar{\bar{U}}_0U_1] - 2\beta_2[U_0U_1]\right)\right\}$$
$$+ 2\delta\left([U_0\partial\beta_2] + [U_2\partial\beta_0]\right) - 4\delta\left(\beta_2D_AU_0^A\right) + 2\delta\beta_2D_AU_0^A - \frac{1}{2}D_AU_0^A\delta\left([CC] - 2D\right) - 6\beta_3\delta V_{+3}$$
$$+ [U_0\delta\mathbb{B}_2] + [U_1\delta\mathbb{B}_1] + U_2^A\delta\left(e^{-2\beta_0}U_A^1\right) - e^{2\beta_0}C^{AB}\partial_A\beta_0\partial_B\delta\ln\sqrt{q} + \frac{1}{2}C^{(S)}\theta_1^r$$
$$+ \frac{1}{8}\left(4D - C^2 - 2[CC]\right)^{(S)}\theta_2^r + \frac{1}{8}\left(4E + 3C[CC] - 4[CD] - 2CD - \frac{1}{2}C^3\right)^{(S)}\theta_3^r.\tag{H.15d}$$

From these expressions it is then possible to reduce the potential to the BS or NU gauges. This

reduction is done by imposing either of the following two conditions:

$$\text{(BS)} \quad \leftarrow \quad C = 0, \quad D = \frac{1}{2}[CC], \quad E = [C\mathscr{D}], \tag{H.16a}$$

$$\text{(NU)} \quad \leftarrow \quad D = \frac{1}{4}[CC], \quad E = \frac{2}{3}\left([C\mathscr{D}] + \frac{1}{8}C(C^2 - 2[CC])\right), \tag{H.16b}$$

where in NU gauge the conditions on $D$ and $E$ are equivalent to $\beta_2 = 0$ and $\beta_3 = 0$ respectively.

In the main text we have focused on the potential $^{(\mathrm{F})}\theta_0$ in the absence of boundary sources, i.e. with $\beta_0 = 0 = U_0^A$. We have furthermore split the potential between a part which survives in BS gauge, and additional terms which are only present in NU gauge (involving essentially $C$). The divergent pieces are reported in (2.103) and (2.104). The constant piece $^{(\mathrm{F})}\theta_0$ is however still very lengthy so we report it here. In BS gauge it is given by

$$\begin{aligned}
^{(\mathrm{F})}\theta_0^r\big|_{\mathrm{BS}} = {}& 2(\delta \ln q + \delta)M - \frac{1}{2}q^{AB}\delta\left(\Lambda\mathscr{E}_{AB} + D_A D^C C_{BC}\right) \\
& + \frac{1}{4}q^{AB}\delta C_{AB}\left(R[q] + \frac{5\Lambda}{8}[CC]\right) + \frac{1}{2}\delta C^{AB}\partial_u C_{AB} - \frac{1}{8}\partial_u\left(C^{AB}\delta C_{AB}\right) \\
& + \frac{1}{2}\left(\frac{3}{2}C^{AB}\partial_u C_{AB} - D_A C^{AB}\partial_B\right)\delta \ln\sqrt{q} \\
& + \frac{1}{8}\left([CC]\delta \ln\sqrt{q} - 3C^{AB}\delta C_{AB}\right)\partial_u \ln\sqrt{q}.
\end{aligned} \tag{H.17}$$

In NU gauge we find instead

$$\begin{aligned}
^{(\mathrm{F})}\theta_0^r\big|_{\mathrm{NU}} = {}& ^{(\mathrm{F})}\theta_0^r\big|_{\mathrm{BS}} \\
& + \left(\frac{\Lambda}{12}C\gamma_4^{AB} - \frac{\Lambda}{12}(C^2 + [CC])C^{AB} + \frac{1}{2}CC^{AB}\partial_u \ln\sqrt{q}\right)\delta q_{AB} \\
& + \frac{1}{2}\left(\frac{37\Lambda}{72}C^3 - \frac{11\Lambda}{18}C[CC] + C\left(C\partial_u - R[q] - C^{AB}\partial_u q_{AB}\right) + \partial^A C\partial_A - \frac{1}{4}[CC]\partial_u \ln\sqrt{q}\right)\delta \ln\sqrt{q} \\
& - \left(\frac{\Lambda}{18}C + \frac{1}{32}\left(\partial_u \ln q + 2\partial_u\right)\right)\delta[CC] \\
& + \left(\frac{\Lambda}{6}C^2 - \frac{2\Lambda}{9}[CC] - \frac{1}{4}\partial_u C + \frac{7}{8}C\partial_u \ln\sqrt{q}\right)\delta C \\
& + \left(\frac{\Lambda}{6}CC^{AB} - \frac{\Lambda}{6}C^2 q^{AB} + \frac{7\Lambda}{96}[CC]q^{AB} + \frac{1}{8}q^{AB}\left(\partial_u \ln\sqrt{q} + 2\partial_u\right)C\right)\delta C_{AB} \\
& + \frac{1}{2}q^{AB}\delta\left(D_A\partial_B C\right) + \frac{1}{8}q^{AB}\delta\partial_u\left(CC_{AB}^{\mathrm{TF}}\right) - \frac{1}{2}\delta\left(CC^{AB}\partial_u q_{AB}\right) \\
& - \frac{1}{4}\delta\left(CR[q]\right) + \frac{1}{8}C\delta\left(\left(2\partial_u + \partial_u \ln\sqrt{q}\right)C\right) - \frac{1}{8}\partial_u\left(C^2\delta \ln\sqrt{q} - CC^{AB}\delta q_{AB}\right) \\
& + \frac{1}{8\sqrt{q}}\left([CC] + C^2\right)\delta\partial_u\sqrt{q} - \frac{\Lambda}{9\sqrt{q}}\delta\left(\sqrt{q}[C\mathscr{D}]\right) - \frac{1}{16}q^{AB}\delta\left(\partial_u q_{AB}[CC] + q_{AB}\partial_u[CC]\right) \\
& + \frac{1}{4\sqrt{q}}C\left(\partial_u\left(\sqrt{q}\,q^{AB}\delta C_{AB}\right) - \partial_u\delta\left(\sqrt{q}\,C\right) - \partial_u\left(\delta\sqrt{q}\,C\right)\right).
\end{aligned} \tag{H.18}$$

This expression can surely be simplified further, but we refrain from doing so here. One should just note that the additional terms which differentiate the BS and NU potential are not only proportional

to $C$. The reason for this is two-fold. First, we see by comparing (3.2) and (4.2) that the on-shell expression of the subleading components of the angular metric differ in BS and NU gauge. Second, we should recall that in BS gauge $\beta_{n>0} \neq 0$ while in NU gauge $\beta_{n>0} = 0$. Since the NU potential is written in (H.18) as the sum of the BS potential and extra terms, these terms need to remove the non-vanishing $\beta_{n>0} \neq 0$ which are present in the BS potential (in order to ensure that for the NU potential we have $\beta_{n>0} = 0$).

# I   Off-shell $\mathbf{E}^{\mathrm{TF}}_{AB}\big|_{\mathcal{O}(r^{-2})}$

We report here the off-shell expression for $\mathrm{E}^{\mathrm{TF}}_{AB}\big|^{\beta_0=0=U_0}_{\mathcal{O}(r^{-2})}$ . This is the expression from which the equations (2.19), (2.20), and (2.21) have been derived in the main text (upon further imposing $\Lambda = 0$ and going on-shell of $\mathfrak{B}^0_{AB} = 0$). The length of this expression also illustrates the clear advantage of the NP formalism over the use of the tensorial Einstein equations. We have

$$
\mathrm{E}^{\mathrm{TF}}_{AB}\big|^{\beta_0=0=U_0}_{\mathcal{O}(r^{-2})} = \frac{2\Lambda}{3} F^{\mathrm{TF}}_{l^2 AB}(\ln r)^2 - \mathrm{EOM}(E_{lAB})\ln r
$$

$$
+ \Lambda \left\{ \frac{1}{3}F^{\mathrm{TF}}_{l^2 AB} - \frac{5}{6}F^{\mathrm{TF}}_{lAB} + \frac{2}{3}F^{\mathrm{TF}}_{AB} - \frac{1}{4}C_{(AC}(E^{\mathrm{TF}}_l)^C_{B)} + \frac{1}{6}CE^{\mathrm{TF}}_{AB} \right.
$$

$$
+ \frac{1}{32}\left( \frac{4}{3}D[CC^{\mathrm{TF}}] + \frac{5}{6}C^2[CC] - \frac{1}{6}C^4 - [CC]^2 \right) q_{AB}
$$

$$
+ \frac{1}{18}C^{\mathrm{TF}}_{(AC}\mathfrak{L}^{CD}C^{\mathrm{TF}}_{DB)} - \frac{1}{9}[CC^{\mathrm{TF}}]\mathfrak{L}_{AB} + \frac{1}{8}C[C\mathfrak{L}]q_{AB} - \frac{1}{6}[D\mathfrak{L}]q_{AB} \bigg\}
$$

$$
+ \frac{1}{4}\left( 3E_l - 5E + \frac{1}{4}CD - \frac{1}{8}C^3 + \frac{3}{16}C[CC] \right) \mathfrak{B}^\Lambda_{AB} - \frac{1}{2}[E\mathfrak{B}^\Lambda]q_{AB}
$$

$$
- \frac{2}{3}\left\{ \partial_u \mathcal{E}_{AB} - \frac{1}{2}\left(D_{(A}\mathcal{P}_{B)}\right)^{\mathrm{TF}} - \frac{3}{2}C^{\mathrm{TF}}_{AB}\mathcal{M}\big|_{\Lambda=0} + \frac{3}{2}q_{AC}\epsilon^{CD}C^{\mathrm{TF}}_{DB}\widetilde{\mathcal{M}} \right.
$$

$$
+ \frac{1}{2}\left( \mathcal{E}_{AB} - 3E^{\mathrm{TF}}_{lAB} - \frac{3}{8}C\mathfrak{L}_{AB} - \frac{9}{4}C^{\mathrm{TF}}_{AB}D + \frac{15}{8}\mathfrak{L}_{(AC}(C^{\mathrm{TF}})^C_{B)} \right)\partial_u\ln\sqrt{q}
$$

$$
+ \frac{1}{2}\left( \partial_u C + \frac{3}{2}R[q] \right)\mathfrak{L}_{AB} + \frac{1}{8}C\partial_u\mathfrak{L}_{AB} + \frac{3}{4}(C^{\mathrm{TF}})^C_{(A}\partial_u\mathfrak{L}_{B)C}
$$

$$
+ \frac{9}{16}D(C^{\mathrm{TF}})^C_{(A}\partial_u q_{CB)} + \frac{7}{16}\mathfrak{L}_{(AC}C^{CD}\partial_u q_{DB)} - \frac{11}{8}(C^{\mathrm{TF}})^C_{(A}(\partial_u q_{CD})\mathfrak{L}^D_{B)}
$$

$$
+ \frac{3}{16}C^{\mathrm{TF}}_{(AC}\mathfrak{L}^{CD}\mathfrak{B}^0_{DB)} - \frac{5}{32}C\mathfrak{L}^C_{(A}\mathfrak{B}^0_{CB)} + \frac{1}{4}[\mathfrak{L}\mathfrak{B}^0]C^{\mathrm{TF}}_{AB}
$$

$$
+ \frac{21}{32}C_{(AC}C^{CD}_{\mathrm{TF}}C^{\mathrm{TF}}_{DE}(\mathfrak{B}^0)^E_{B)} - \frac{3}{32}[CC]C^{\mathrm{TF}}_{(AC}(\mathfrak{B}^0)^C_{B)}
$$

$$
\left. - \frac{41}{64}CC_{(AC}C^{CD}\mathfrak{B}^0_{DB)} + C[CC^{\mathrm{TF}}]\mathfrak{B}^0_{AB} + \frac{7}{16}C^2C^{\mathrm{TF}}_{(AC}(\mathfrak{B}^0)^C_{B)} - \frac{3}{16}C[C\mathfrak{B}^0]C^{\mathrm{TF}}_{AB} \right\}
$$

$$
- \frac{1}{3}\left( (2\partial_u + \partial_u\ln\sqrt{q})E^{\mathrm{TF}}_{lAB} + \frac{1}{2}\left(D_{(A}U^{l3}_{B)}\right)^{\mathrm{TF}} \right)
$$

$$
+ \left( \frac{1}{2}E^C_{l(A} - 4E^C_{(A} \right)\mathfrak{B}^0_{B)C} + 5\left( E - \frac{1}{4}E_l \right)\mathfrak{B}^0_{AB}, \tag{I.1}
$$

with $\mathcal{M}\big|_{\Lambda=0} = M + \frac{1}{16}\big(\partial_u + \partial_u \ln \sqrt{q}\big)\big(4D - [CC]\big)$ and

$$\mathrm{EOM}(E_{lAB}) = \big(2\partial_u + \partial_u \ln \sqrt{q}\big)E_{lAB}^{\mathrm{TF}} + \frac{1}{2}\big(D_{(A}U_{B)}^{l3}\big)^{\mathrm{TF}} - \frac{1}{2}E_l\mathfrak{B}_{AB}^0 \tag{I.2}$$
$$+ E_{l(AC}^{\mathrm{TF}}\mathfrak{B}_{B)}^{0C} + \frac{3}{4}E_l\mathfrak{B}_{AB}^\Lambda + \frac{1}{2}[E_l\mathfrak{B}^\Lambda]q_{AB}$$
$$+ \frac{\Lambda}{3}\left(5F_{l^2AB}^{\mathrm{TF}} - 2F_{lAB}^{\mathrm{TF}} - \frac{1}{2}CE_{lAB}^{\mathrm{TF}} + \frac{1}{8}\big([CC] - C^2\big)\mathfrak{L}_{AB} + \frac{1}{8}\big(C_{(AC}\mathfrak{L}^{CD}C_{DB)}\big)^{\mathrm{TF}}\right).$$

We note that the first term on the second line is always vanishing on-shell. This is because we have $\mathfrak{B}_{AB}^0 = 0$ in the flat case and $E_{lAB}^{\mathrm{TF}} = 0$ in (A)dS.

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
