# Peer review of "The partial Bondi gauge: Further enlarging the asymptotic structure of gravity"

_SciPost Physics_

## Round 1 · Referee Report · Anonymous (Referee 1) · 2022-7-8

Strengths

1- Structure and organisation of the paper

2- Unifying approach to the study of the spacetime asymptotics of General Relativity at null infinity

3- Relaxation of working assumptions commonly used in the literature

Weaknesses

1- Falling short of determining the set of physical asymptotic symmetries (understandably deferred to follow-up work)

Report

In this paper the authors study the asymptotic structure of General Relativity in a more general setup than is usually done. In particular they consider both vanishing and non-vanishing comsological constant, a polyhomogenous radial expansion and a partial gauge fixing which encompasses the common Bondi-Sachs and Newman-Unti gauges.

The authors start by defining the partial Bondi gauge where only three gauge fixing conditions are imposed, such that the radial coordinate is still completely arbitrary. Upon imposing a fourth gauge condition which (partially) determines the radial coordinate, one can reduce to the Bondi-Sachs or Newman-Unti gauge. They study the solution space with prescribed asymptotic falloffs and polyhomogeneous expansion for the angular components of the metric, the constraint equations on the various metric functions, the relation with the Newman-Penrose formalism, the symplectic potential and the residual diffeomorphisms. In particular, they find that the asymptotic symmetry algebra is a large extension of the BMS algebra. Of prime interest is to determine which subset of these residual diffs are genuine physical symmetries associated with nonzero canonical generators, a problem which is however left out to future work.

The authors subsenquently study the reduction of their findings to the Bondi-Sachs gauge where they recover previous results from the literature. They propose a relaxed Newman-Unti gauge where the standard condition $g_{ur}=-1$ is replaced by $\partial_r g_{ur}=0$, which they argue allows to uncover boundary Weyl rescalings. However Weyl rescalings were already obtained by Barnich and Lambert [127] in the historical gauge fixing $g_{ur}=-1$, such that the claim of the authors is misleading. In addition they relax the tracelessness condition on the shear tensor $C_{AB}$, which yields a new residual asymptotic diff shifting the trace of $C_{AB}$.

This work unifies many results in the literature and provides various extensions which are worth investigating. Although long and technical, the authors made substantial effort in structuring the paper in a readable way. I recommend it for publication.

Requested changes

1- The authors should consider giving proper credit to Barnich and Lambert [127] for uncovering the Weyl-BMS algebra in NU gauge, and correct their statement that the differential NU gauge condition is necessary to do so.

2- In the general discussion, I would recommend commenting on the asymptotic symmetries and conservation laws at spatial infinity. There are many reasons for this. Although charges at scri are generically not integrable nor conserved, one expects them to match onto conserved charges in the limit to spatial infinity (1704.06223, 1902.08200, 2110.04900, 2204.06571). If not, the physical meaning of these charges would have to be clarified. However conservation of the charges at spatial infinity cannot happen if the symmetry parameters are arbitrary functions of three coordinates (time and angular variables), in particular the time dependence must be constrained. This indeed agrees with the situation encountered for extended/generalized BMS at scri, but raises questions regarding the asymptotic symmetries discussed in this paper. Another interesting point is that symmetries which go beyond previously known BMS symmetries are known to exist at spatial infinity and some of the corresponding charges have been computed in 1106.4045. The spi-supertranslations contain two branches of symmetries: one maps to BMS supertranslations while the other have nonzero charges only if overleading log r divergences in the Weyl tensor are allowed! (see 1704.06223) These have usually been discarded without further reasons. In addition there exist also the less understood logarithmic supertranslations which also require log r terms at scri. One could imagine relating these quantities at $i^0$ with the ones studied at scri in this paper using the methodology of 2204.06571.

  • validity: high
  • significance: good
  • originality: good
  • clarity: high
  • formatting: excellent
  • grammar: excellent

Author:  Marc Geiller  on 2022-07-29  [id 2698]

(in reply to Report 2 on 2022-07-08)

  1. It is correct that the Weyl rescalings in NU gauge were already revealed and studied in the work [127] by Barnich and Lambert. We had already mentioned this point in the first version of the manuscript at the beginning of section 4.1. But as the referee correctly points out, our phrasing about the relationship between Weyl and $\beta_0$ was a bit ambiguous/confusing in the introduction of the manuscript. The correct statement is that the Weyl rescalings are indeed always present in NU gauge (since in this gauge the determinant condition is dropped). However, in the usual analysis where $\beta=0$ (i.e. with the algebraic NU gauge), the Weyl transformations $h$ are time-independent and related to the time translations $f$ as $h=-\partial_uf$. Since in this work we aim at relaxing completely the boundary metric and its time-dependency, we are looking for symmetry parameters which are arbitrary functions of $u$ and $x^A$. What we initially meant to say is that the differential NU gauge is indeed necessary in order to access the \textit{time-dependent} Weyl transformations. This can indeed be seen on equation (2.83b) or on (4.4), which show that setting $\beta_0$, or equivalently going to the algebraic NU gauge, does not allow for an independent symmetry generator $h$, but instead relates this latter to $\partial_uf$. We have added remarks and precisions are various locations in the manuscript to clarify this point, and to emphasize once again that, indeed, the Weyl transformations in NU gauge were revealed already in [127]. These are the locations of the changes: 1) end of the 1st paragraph of the part ``Gauge choices'' in section 1.2. 2) caption of table 1. 3) beginning of section 4.1.

  2. We thank the referee for the references and interesting comments regarding structure at spatial infinity. We have added a final bullet point in section 5 ``Perspectives'' to discuss these important directions for future work.

---

## Round 1 · Referee Report · Anonymous (Referee 2) · 2022-7-12

Report

The mail goal of this paper is to derive the solution space of Einstein’s gravity in generalized outgoing Bondi-type coordinates with enriched boundary structure and assuming polyhomogeneous radial expansion. Considering the current popularity of asymptotic symmetries and the current research endeavor being invested in, the numerous results compiled in the present manuscript are unquestionably timely and scientifically sound.

Using a partial gauge fixing in outgoing null coordinates encompassing both Bondi-Sachs and Newman-Unti coordinate systems, the general solution of Einstein’s equations near future null infinity is derived for any value of the cosmological constant. The residual gauge transformations acting on the solution space, as well as the variations they induce on the phase space parameters and their vector algebra are extensively computed. The latter consists of the enhancement of the Generalized BMS algebra (resp. Λ-BMS algebroid) by a boundary Weyl rescaling and a radial ``super-translation’’ when Λ = 0 (resp. Λ ≠ 0).

Important achievements of the paper include: - Analyzing the impact of logarithmic terms in the radial expansion on the structure of the solution space as well as precisely quantifying the violation of Sachs’ peeling property; - Discussing the solution space in the Newman-Penrose formalism, which allows to obtain the quantities that transform homogeneously under the extended asymptotic symmetries in a very natural and transparent way. The structuration of the equations of motion is also more luminous in this formalism (see e.g. Equations (2.71)-(2.72)). - Computing the presymplectic potential assuming weaker boundary conditions in (A)dS, which will allow to discuss various extensions (including the Weyl symmetry) of the Λ-BMS phase space in the future and compute the associated surface charges.

The authors offer a very detailed and well-commented analysis, which is pleasant to read despite of the technical aspect of the paper. It should be noticed that, even though it significantly lengthens the manuscript, a constant effort has been put on pedagogy, clarity of explanations and precision regarding involved mathematical hypotheses, which is also more than commendable. The presence of summative tables and intermediate summaries greatly helps in following the construction and the multiple choices of notations and conventions.

For all these reasons, I am happy to recommend this manuscript for publication in SciPost and wish this paper a great success as a technical milestone in the field.

Requested changes

Just two comments that could be addressed in the proofs:

1. Concerning Section 3.3. In [67] and [69], the flat limit was taken (always in Bondi gauge) as a parametric limit Λ -> 0 assuming that the C_AB field has a well-defined flat limit. Using that, it was found that the presymplectic structure has a pole in 1/Λ that must be regularized by playing with the corner term ambiguities of the formalism. It should be interesting to precise a bit the assertion ``the flat limit is well-defined’’ by stating clearly all the hypotheses undergoing the flat limit process considered here.
2. Concerning Equation (2.68). Traditionally, the Bondi mass loss formula comes from the evolution equation written as \partial_u M = -1/8 N_{AB} N^{AB} + total derivative, and the crucial point is that the first term is a quadratic form with strictly negative prefactor. Hence the local flux of the BMS supertranslation charge is negative semi-definite. When the Bondi mass aspect is replaced by the covariant mass aspect, reading from (2.68), the equation becomes something like \partial_u \mathcal M = 1/8 \partial_u C_{AB} N^{AB} (forgetting about refinement related to the boundary structures), which is no more a negative semi-definite quantity but would require an integration over the retarded time to reproduce the right form of the integrated BMS flux (which is purely outgoing as it should). It should be interesting to comment a bit more on that point.

  • validity: top
  • significance: high
  • originality: good
  • clarity: top
  • formatting: excellent
  • grammar: perfect

Author:  Marc Geiller  on 2022-07-29  [id 2697]

(in reply to Report 1 on 2022-07-12)
Category:
answer to question

  1. It is correct that the flat limit process differs slightly in our manuscript and in [67,69]. This difference is due to a corner term. Let us explain where this term comes from. In our computation of the symplectic potential, we have started form the onset in the Bondi gauge, and obtained in the case of the BS gauge the generic expression (3.4). By construction, this symplectic potential admits a flat limit with no divergencies in $\Lambda^{-1}$. In this sense, there are therefore no hypothesis in our derivation of the symplectic structure and of its flat limit. We have simply evaluated the Einstein--Hilbert symplectic potential on-shell to obtain (3.4), which has a well-defined flat limit. When going to AdS, we have to rewrite the shear in terms of $\partial_uq_{AB}$, which introduces factors of $\Lambda^{-1}$, and we have found that (3.4) can be rewritten identically as (3.8). Starting from (3.8), one can therefore work our way back to (3.4), and the flat limit is well-defined. Now, in order to understand the difference with [67,69], one should note that in these references the starting point is e.g. (4.4) of [69]. From there, the authors work their way back to (4.7), which contains a divergent corner term in the flat limit. This corner term is precisely that contained in (3.8) of our manuscript. Up to the change of notations, (4.4) of [69] is therefore only the first term in (3.8) of our manuscript (appart from the corner term the other terms are irrelevant for the present discussion). Since we have shown that our (3.8) is identically equal to (3.4), which admits a flat limit, it is normal to observe that (4.4) of [69], which differs from our (3.8) by a corner term, does not lead to a well-defined flat limit. This is the origin of the extra corner term introduced by the authors of [67,69]. In summary, this slight difference appears because the starting point of [67,69] is the Fefferman--Graham symplectic potential, which is then mapped to Bondi gauge via the diffeomorphism between the two gauges. Both procedures are however correct. We have added a comment below (3.8) of our manuscript to point out this slight difference of treatment of the flat limit.

  2. It is correct that (2.68) is not the Bondi mass loss formula in its usual form, since here it is written in terms of the covariant mass $\mathcal{M}$ instead of the Bondi mass $M$. The former has the property of being covariant under the action of the BMSW asymptotic symmetry group, while the latter satisfies the ``true'' mass-loss formula with a negative semi-definite news squared term on the rhs. We have added a paragraph below (2.68) to point this out and to explain the difference between the two formulas. It seems to us that, to some authors, the proper definition of the mass is still debated, precisely because of these two different properties.

Attachment:

Reply.pdf

---

## Editorial Decision

resubmitted